# THINKMORPH:
# EMERGENT PROPERTIES IN MULTIMODAL INTERLEAVED CHAIN-OF-THOUGHT REASONING

**Jiawei Gu**[*,1], **Yunzhuo Hao**[*,2], **Huichen Will Wang**[*,3], **Linjie Li**[*,3], **Michael Qizhe Shieh**[1,5], **Yejin Choi**[4], **Ranjay Krishna**[3], **Yu Cheng**[6]

[1]National University of Singapore, [2]Zhejiang University, [3]University of Washington, [4]Stanford University, [5]absolute AI, [6]The Chinese University of Hong Kong

🏠 Homepage: `https://thinkmorph.github.io`
🐙 Code: `https://github.com/ThinkMorph/ThinkMorph`
🤗 Models and Datasets: `https://huggingface.co/ThinkMorph`

## ABSTRACT

Multimodal reasoning requires iterative coordination between language and vision, yet it remains unclear what constitutes a meaningful interleaved chain of thought. We posit that text and image thoughts should function as complementary, rather than isomorphic, modalities that mutually advance reasoning. Guided by this principle, we build ThinkMorph, a unified model fine-tuned on ∼24K high-quality interleaved reasoning traces spanning tasks with varying visual engagement. ThinkMorph learns to generate progressive text–image reasoning steps that concretely manipulate visual content while maintaining coherent verbal logic. It delivers large gains on vision-centric benchmarks (averaging 34.7% over the base model) and generalizes to out-of-domain tasks, matching or surpassing larger and proprietary VLMs. Beyond performance, ThinkMorph exhibits emergent multimodal intelligence, including unseen visual manipulation skills, adaptive switching between reasoning modes, and better test-time scaling through diversified multimodal thoughts. These findings suggest promising directions for characterizing the emergent capabilities of unified models for multimodal reasoning.

## 1 INTRODUCTION

Multimodal reasoning (Lin et al., 2025) is not a single-pass perception task but an iterative process that interweaves language and vision reasoning. This process remains particularly challenging for current models in vision-centric domains such as spatial reasoning (Li et al., 2025c; Cai et al., 2025), where success requires moving beyond describing images toward interrogating and manipulating visual elements. While textual Chain-of-Thought (hereafter, "text thought") (Wei et al., 2022) has advanced verbal reasoning, it contributes little to multimodal reasoning: models still falter when problems demand more than textual description (Hao et al., 2025; Jiang et al., 2025). These limitations motivate a shift from language-driven reasoning to genuinely cross-modal reasoning—mirroring the human ability to tackle complex problems through "think-and-sketch" strategies.

To emulate such think-and-sketch behavior, researchers have explored multimodal interleaved Chain-of-Thought (hereafter, "interleaved thought"), yet existing approaches remain limited. Tool-augmented designs rely on external visual modules such as cropping tools (OpenAI) or specialized sketching models (Hu et al., 2024; Zhou et al., 2024), making the reasoning process indirect and brittle. Unified models (Team, 2024; Chern et al., 2024) offer a more integrated alternative but have yet to yield a generalizable recipe for mutual advancement between text and image reasoning. For instance, MVoT (Li et al., 2025b) introduces interleaved action representations for maze solving,

---

*Equal contribution.

yet its textual component is confined to simplistic action labels ***isomorphic*** (Fu et al., 2024a) to its generated images, showing little generalization beyond training domains.

We posit that achieving generalizable multimodal reasoning requires treating text and images as complementary, rather than isomorphic, modalities that jointly advance reasoning. Building on this principle, we introduce **ThinkMorph** — a unified model fine-tuned on $\sim$24K interleaved traces spanning four tasks with varying levels of *visual engagement*, from chart highlighting to spatial path overlays (Figure 2). Each instance is meticulously designed to ensure high-quality multimodal supervision, where textual reasoning and visual manipulation progress hand-in-hand toward solutions. ThinkMorph achieves substantial gains on vision-centric tasks, averaging a **34.74% improvement** over its base model, with striking increases of **85.84%** on *Spatial Navigation* and **38.75%** on *Jigsaw Assembly*. Beyond these quantitative improvements, it provides a controlled setting to examine when and how interleaved reasoning helps multimodal problem solving.

Compared across reasoning modes, ThinkMorph's interleaved reasoning consistently outperforms text-only and visual-only approaches by **5.33%**. Despite its modest data scale, ThinkMorph generalizes robustly to out-of-domain benchmarks, surpassing the larger InternVL3.5-38B on spatial reasoning in SAT by achieving 52.67% compared to 49.33%, and matching Gemini 2.5 Flash on MMVP perception at 80.33%. Beyond accuracy, ThinkMorph reveals emergent properties indicative of higher-level multimodal intelligence: **PROPERTY ①–Unseen Visual Manipulations**, where the model generates visual edits unseen during training; **PROPERTY ②–Autonomous Mode Switching**, where it adaptively allocates reasoning effort between modalities; and **PROPERTY ③–Better Test-time Scaling with Diversified Thoughts**, where it explores broader multimodal solution spaces, yielding stable accuracy gains such as +8.0% on *Jigsaw Assembly*. Collectively, these properties indicate that interleaved reasoning is not merely a coordination mechanism but an engine for emergent behaviors, offering a window into how unified models internalize and adapt multimodal problem-solving strategies.

Overall, our work makes the following contributions:

- **Systematic analysis of interleaved multimodal reasoning.** We present ThinkMorph as a unified framework for scalable investigation of interleaved reasoning, providing the first systematic study of when and how multimodal interleaving surpasses text-only and visual-only modes. Through carefully curated, high-quality data and the scalable generation of $\sim$24K mutually reinforcing interleaved traces, ThinkMorph achieves substantial improvements across diverse out-of-domain benchmarks while revealing the underlying dynamics that make interleaved reasoning effective and generalizable.

- **Emergent properties in interleaved reasoning.** We identify distinctive emergent behaviors that arise from interleaved reasoning, including unseen visual manipulations, autonomous switching between reasoning modes, and diversified multimodal exploration. These behaviors highlight the model's ability to internalize adaptive strategies that balance symbolic and perceptual reasoning.

- **New avenues for test-time scaling.** We show that the advantages of interleaved reasoning persist and amplify during test-time scaling, enabling broader exploration of multimodal solution spaces and improved robustness across unseen domains.

## 2 THINKMORPH: INTERLEAVED CHAIN-OF-THOUGHT GENERALIZATION

Let $\mathcal{P}_\theta$ denote a unified multimodal model with parameters $\theta$. We consider a multimodal question $\mathcal{Q} = (\mathcal{Q}^{\text{text}}, \mathcal{Q}^{\text{img}})$ that may contain both textual and visual elements. For reasoning tasks, the model is prompted to generate a sequence of intermediate tokens to reach a final answer. Unlike conventional Chain-of-Thought approaches that only produce textual tokens $\hat{t}$, ThinkMorph can also generate image tokens $\hat{v}$, resulting in interleaved thoughts that combine both modalities. Formally, the thought sequence is defined as $\mathcal{T} = (\hat{m}_1, \hat{m}_2, \ldots, \hat{m}_n)$, where $\hat{m}_i \sim \mathcal{P}_\theta(m_i \mid x, m_0, \hat{m}_1, \ldots, \hat{m}_{i-1})$ and $\hat{m}_i \in \{\hat{t}_i, \hat{v}_i\}$. In practice, while special tokens are omitted from the notation for simplicity, modality transitions are controlled via delimiter tokens. For instance, image thoughts are delimited by `<image_start>` and `<image_end>` tokens, enabling seamless switching between textual and visual reasoning within the sequence.

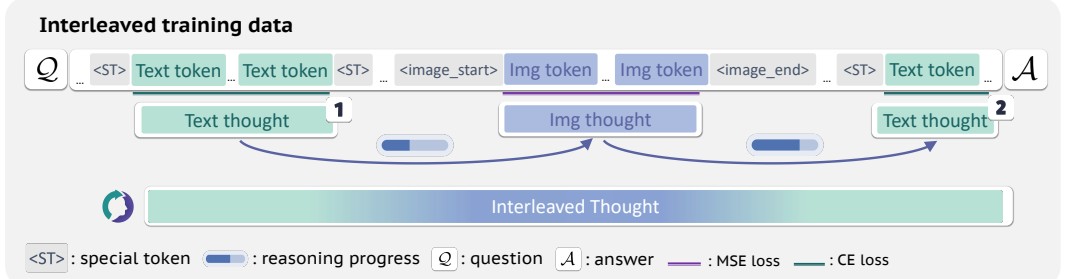

**Figure 1: Design of Interleaved Training Data for Progressive Multimodal Reasoning**

**Interleaved Thought Collection** Advancing multimodal reasoning through interleaved thought presents a foundational challenge: defining what counts as meaningful interleaving is inherently difficult. Unlike textual reasoning, visual thinking is hard to externalize, whether through language or sketches. For many visual reasoning tasks (Hao et al., 2025; Li et al., 2025c; Yin et al., 2025), humans often use arrows, rough shapes, or symbols that show relationships but not exact details. This ambiguity makes it hard to set clear criteria and to collect data at scale.

To address this challenge, we construct an enriched dataset encompassing four representative tasks that demand different levels of visual engagement and cross-modal interaction, as illustrated in Figure 2. Each task supports concrete, verifiable intermediate visual thoughts grounded in specific visual manipulations. We carefully design task-specific interleaved reasoning sequences where text and images are not treated as isomorphic representations but provide complementary cues that progressively guide the reasoning process toward a solution, as shown in Figure 1. The following tasks demonstrate how alternating between textual and visual tokens facilitates cross-modal reasoning:

▷ **Jigsaw Assembly** (Wang et al., 2025c) requires determining the correct arrangement of scrambled image patches to reconstruct the original image. To recover the patch ordering $\sigma^\star$, the initial $\hat{t}$ tokens provide piece-wise textual descriptions of each puzzle piece's local content. The subsequent $\hat{v}$ tokens then visualize the re-arranged pieces according to the current ordering hypothesis $\sigma$, supplying holistic spatial context that text alone cannot capture. The final $\hat{t}$ tokens perform syntactic verification of the reconstructed assembly. ▷ **Spatial Navigation** (Wu et al., 2024) involves finding a safe route from a starting point to a goal on a grid map, avoiding obstacles. To determine a safe path $\pi^\star \in \mathcal{P}^*$ through a maze, the initial $\hat{t}$ tokens establish a coarse global abstraction. The $\hat{v}$ tokens then render the visual trajectory of $\pi^\star$, and the final $\hat{t}$ tokens articulate and verify the corresponding sequence of moves. ▷ **Visual Search** (Wu & Xie, 2024) involves answering a question about a target object in an image $\mathcal{Q}^{\text{img}}$. To locate the target object, the initial $\hat{t}$ tokens hypothesize and describe the area of interest. The $\hat{v}$ tokens subsequently draw a bounding box, offering an explicit visual anchor. The final $\hat{t}$ tokens verbalize the object's attributes and confirm the prediction. ▷ **Chart Refocus** (Fu et al., 2025) requires answering a question about a data visualization. To do so, the initial $\hat{t}$ tokens identify relevant data elements. The $\hat{v}$ tokens highlight corresponding regions of interest, and the final $\hat{t}$ tokens perform value extraction and computation.

| Task | Data Source | Count | Visual Manipulation | Curation Steps |
|---|---|---|---|---|
| **Jigsaw Assembly** | SAT (Ray et al., 2024), ADE20K (Zhou et al., 2017), Omni3D (Brazil et al., 2023) | 6,000 | Visualizing re-arranged pieces | Newly generated questions from a customized pipeline |
| **Spatial Navigation** | N/A | 6,000 | Overlaying mazes with paths highlighted with red lines and arrows | Newly generated questions from a customized pipeline |
| **Visual Search** | Visual CoT (Shao et al., 2024), GQA (Hudson & Manning, 2019), VSR (Liu et al., 2023) | 6,990 | Highlighting Regions with Red Bounding Boxes | Filtering for valid (question, answer) with MLLMs + other criteria |
| **Chart Refocus** | ChartQA (Masry et al., 2022), Refocus (Fu et al., 2025) | 6,000 | Highlighting Regions with Red Bounding Boxes or Overlays | Filtering for valid (question, answer) with MLLMs + other criteria |

**Table 1: Summary of Questions Used for Training ThinkMorph.**

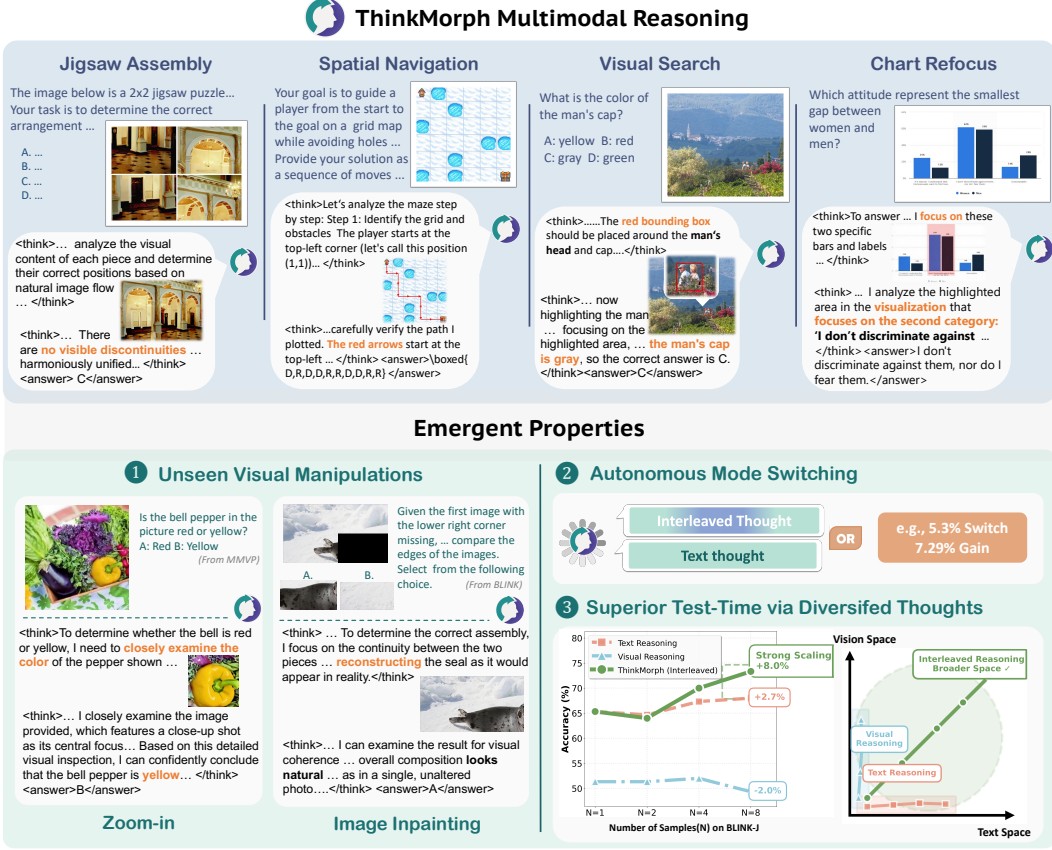

**Figure 2: ThinkMorph Overview.** ThinkMorph synergistically interleaves language and vision to advance multimodal reasoning across four representative tasks (top). Beyond performance gains on in- and out-of-domain benchmarks, interleaved reasoning unlocks emergent properties (bottom).

**Data Synthesis** Table 1 summarizes the data sources, curation pipeline, and visual manipulations used for each task. In total, we curate **24,990 questions** spanning diverse domains. Questions for *Jigsaw Assembly* and *Spatial Navigation* are generated using our custom synthesis pipeline, whereas those for *Visual Search* and *Chart Refocus* are carefully curated through a human-in-the-loop MLLM filtering process. For instance, in the *Visual Search* task, we observe many questions from existing Visual CoT datasets (e.g., GQA and VSR) are ambiguously phrased, contain incorrect answers, or highlight irrelevant objects in the solution images. To enhance quality and difficulty, we enforce a constraint that the target object's bounding box must occupy between 1% and 30% of the image area. This selective filtering reduces the dataset from 144K to 6,990 high-quality questions. In addition to the interleaved traces, we derive two unimodal baselines: textual thoughts obtained by prompting GPT-4.1 to solve each task step-by-step, and visual thoughts using only the image outputs from the interleaved reasoning traces. All details are provided in Appendices B.6 and D.

**Training and Evaluation** We adopt Bagel as our base model and train on its official implementation. As shown in Figure 1, we optimize dual objectives: Mean Squared Error (MSE) loss $\mathcal{L}_{\text{img}}$ for image tokens and negative log-likelihood loss $\mathcal{L}_{\text{text}}$ for text tokens. Hyperparameters vary across training settings, and detailed configurations are provided in Appendix B.8. For in-domain evaluation, we use **VSP-main-task** (Wu et al., 2024) as the benchmark for *Spatial Navigation*, our constructed **VisPuzzle** for *Jigsaw Assembly*, and the official **Chart Refocus** (Fu et al., 2025) test set (a subset of ChartQA (Masry et al., 2022)). For out-of-domain evaluation, we further test on a broad suite of vision-centric multimodal benchmarks, including **VStar** (Wu & Xie, 2024), **BLINK** (Fu et al., 2024b), **MMVP** (Tong et al., 2024c), **SAT** (Ray et al., 2024) and **CV-Bench** (Tong et al., 2024a). Specifically, for BLINK, its subset BLINK-Jigsaw falls under the jigsaw assembly task, which differs substantially from our task VisPuzzle. We treat it as a distinct metric, hereafter denoted as **BLINK-J**. All evaluations are conducted using the `vlmevalkit` framework (Duan et al., 2024) for consistency and reproducibility. For most benchmarks, we follow the framework's origi-

nal evaluation pipeline. For tasks where answer extraction and correctness could not be determined by exact matching, we adopt GPT-5 as an LLM-as-a-Judge. Additional details are provided in Appendix B.7.

# 3 WHEN DOES INTERLEAVING IMPROVE MULTIMODAL REASONING?

ThinkMorph exploits the *complementarity* between text and images to enable interleaved reasoning, where each modality contributes distinct yet synergistic information toward solving a problem. To probe the scope and underlying mechanisms of this advantage, we ask two central questions: **when** does interleaved reasoning outperform unimodal approaches, and **how** does this advantage emerge? To answer these, we fine-tune Bagel-7B under three distinct reasoning modes—text-only, visual-only, and interleaved—and evaluate their performance across all tasks (Table 2).

| | Spatial Navigation | Visual Search | Jigsaw Assembly | | Chart Refocus | |
|---|---|---|---|---|---|---|
| | VSP | VStar ★ | VisPuzzle | BLINK-J ★ | ChartQA | MMVP ★ |
| Bagel-7B | 0.83* | 55.49 | 35.00* | 67.33 | 62.05 | 70.33 |
| Text Reasoning | 49.17 | 56.02 | 63.50 | 68.67 | **81.66** | 76.33 |
| Visual Reasoning | 85.50 | 58.63 | 61.25 | 47.33 | 73.08 | 73.00 |
| ◉ **Interleaved Reasoning** | **86.67** | **63.87** | **73.75** | **73.33** | 79.78 | **82.66** |

**Table 2: Reasoning Mode Comparison.** Bagel-7B is tested under think mode (*: no-think mode for tasks where thinking prevents Bagel from generating answers). ChartQA results are the average performance on horizontal and vertical bar chart questions. ★: out-of-domain benchmarks. **Best** , second-best .

**Interleaved reasoning excels on vision-centric tasks.** On tasks that demand sustained visual engagement, ThinkMorph's interleaved reasoning consistently outperforms all other modes. The effect is most pronounced in *Spatial Navigation*, where the base model nearly fails at 0.83% but interleaved reasoning reaches 86.67%, marking a dramatic 85.84% improvement. Substantial gains also appear in *Jigsaw Assembly*, with a 38.75% in-domain improvement and strong out-of-domain generalization on BLINK-J (+6.00%). For *Visual Search*, ThinkMorph improves performance on the out-of-domain VStar benchmark by 8.38%. Averaged across these three vision-centric tasks, interleaved reasoning yields a 34.74% improvement over the base model and surpasses the next-best mode by 5.33%, establishing it as the most effective reasoning strategy for visually grounded problems.

**When is interleaved reasoning necessary?** *Chart Refocus* highlights when visual manipulation in reasoning traces is **essential** versus **supplementary**. On the in-domain ChartQA benchmark, text-only reasoning slightly outperforms interleaved reasoning (+1.88%), indicating that visual input adds little beyond text. In contrast, on the out-of-domain MMVP benchmark, interleaved reasoning generalizes better, surpassing text-only reasoning by 6.33%. This contrast clarifies both when interleaved reasoning helps and how its advantage arises.

Across vision-centric tasks, interleaved reasoning works best when text and images continuously inform each other. Visual tokens enable steps that text alone cannot: in *Jigsaw Assembly*, re-arranged pieces reveal mismatches; in *Spatial Navigation*, overlaid arrows validate routes; and in *Visual Search*, bounding boxes pinpoint object locations. *Chart Refocus*, however, shows that the need for interleaving depends on task demands (Figure 3). In ChartQA, textual reasoning already

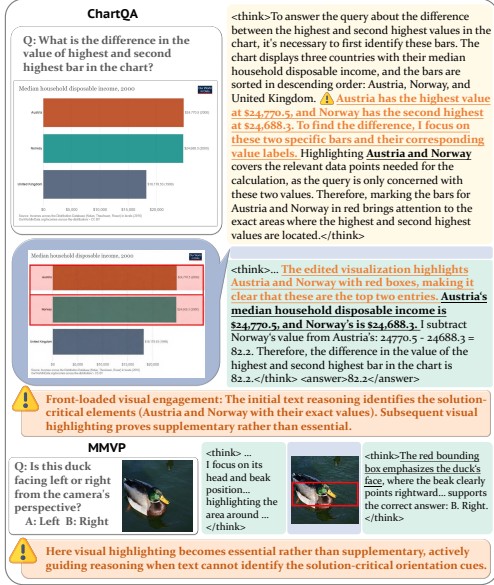

**Figure 3: Visual Highlighting:** Role varies from supplementary (ChartQA) to essential (MMVP).

identifies key elements (e.g., Austria and Norway with their values), making later visual highlighting helpful but unnecessary. In MMVP, visual grounding is essential for spatial cues that text cannot express, such as confirming that "the duck's beak points rightward." Overall, text-only reasoning suffices when additional visual information in the reasoning traces is redundant, but interleaved reasoning is crucial for generalizing to tasks requiring precise visual grounding or manipulation.

## 4 EMERGENT PROPERTIES IN INTERLEAVED REASONING

Beyond performance improvements, interleaved reasoning also exhibits emergent properties that arise naturally during training and evaluation, showing behaviors characteristic of multimodal intelligence (see lower panel of Figure 2).

> **EMERGENT PROPERTY ① : Unseen Visual Manipulations** The model develops accurate and meaningful visual manipulations unseen in training data when generalizing to out-of-domain multimodal tasks, actively advancing the reasoning process.

We identify eight distinct types of unseen visual manipulations, with *zoom-in* operations being the most common. As shown in Figure 2 (lower panel) and Figure 4, these manipulations also include *inpainting*, *multi-box generation*, motion *forecasting*, perspective *transformation*, and region *cropping*, among others. These emergent behaviors are not rare: on some benchmarks, unseen manipulations account for up to 10% of all visual operations produced during inference. Importantly, these operations are not arbitrary artifacts but **precise** and **task-effective** visual actions that contribute directly to problem solving. For example, when asked *"Is the bell pepper red or yellow?"*, the model automatically generates a zoomed-in view to better distinguish subtle color differences, closely mirroring **human visual inspection** without explicit prompting.

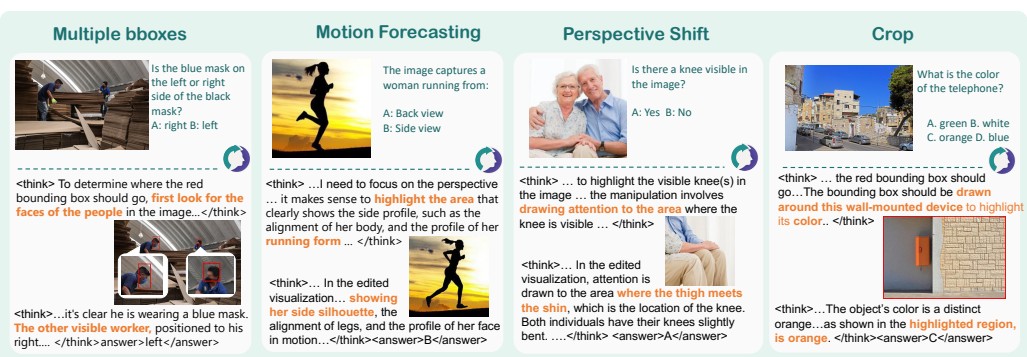

**Figure 4: Examples of More Unseen Manipulations**

A deeper analysis reveals systematic patterns underlying these behaviors. Statistical evidence shows that specific textual cues reliably trigger corresponding visual manipulations: phrases such as *"examine closely"* or *"focus on"* consistently elicit zoom-in operations, while terms like *"restore"* and *"reconstruct"* prompt image inpainting. These correlations are both **consistent** and **contextually appropriate**, suggesting principled rather than random generation. This capability originates from Bagel's large-scale multimodal pretraining, which exposes the model to interleaved visual–text patterns encompassing diverse manipulation. ThinkMorph's interleaved reasoning fine-tuning then provides critical alignment by enabling the unified model to activate these manipulation skills within structured reasoning steps for problem solving. In essence, pretraining supplies the raw manipulation ability, while interleaved fine-tuning directs it toward reasoning-oriented visual behaviors. Additional examples and analyses are provided in Appendix C.2.

> **EMERGENT PROPERTY ② : Autonomous Mode Switching** The model adaptively switches from interleaved to text-only reasoning based on task complexity, despite being trained exclusively on interleaved data.

When trained solely on interleaved *Chart Refocus* and *Jigsaw Assembly* data, the model still generalizes strongly to the out-of-domain MMVP benchmark, achieving 79.6% and 82.66% respec-

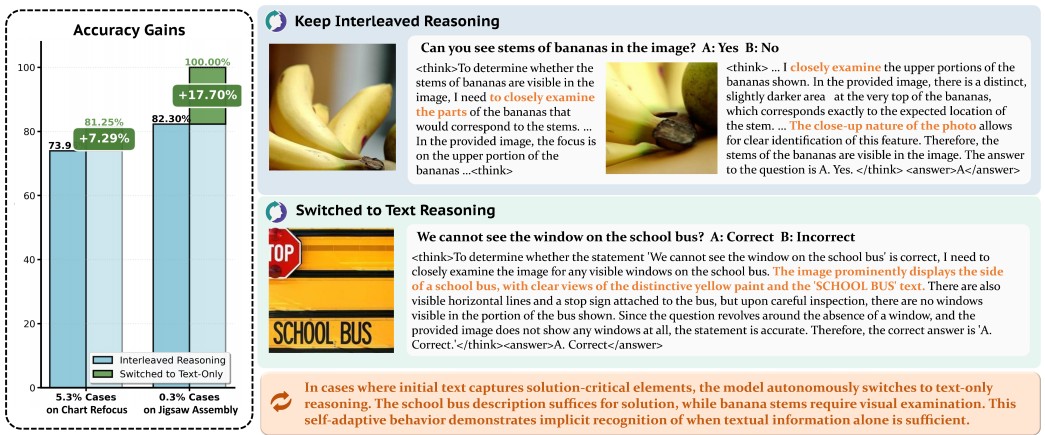

**Figure 5: Autonomous Mode Switching Based on Task Complexity.**

tively (Table 2), well above the Bagel-7B baseline of 70.33%. A closer look reveals a striking phenomenon. Despite being trained exclusively on interleaved traces, the model **autonomously switches** to text-only reasoning in 5.3% of inference cases (Figure 5). Notably, these switched instances reach 81.25% accuracy overall, a 7.29% improvement over the same samples when solved using interleaved reasoning (73.96%), demonstrating that the model can selectively adapt its reasoning mode for higher effectiveness.

**Mode switching is task-adaptive, not arbitrary.** As shown in Figure 5, the model adapts its reasoning behavior based on visual complexity. For the question *"Can you see stems of bananas in the image?"*, it maintains interleaved reasoning, generating a zoomed-in view of the upper region where the stem would appear. The close-up enables clear stem identification, illustrating that continuous visual engagement remains essential when fine-grained details are critical to the solution. In contrast, for *"We cannot see the window on the school bus?"*, the model switches to pure textual reasoning, describing visible features such as the yellow paint and lettering, to infer the absence of windows. This contrast reflects a form of **front-loaded visual engagement**: after processing the image and question, the model implicitly decides whether text alone can complete the reasoning. When the initial visual encoding captures information that text can express, it shifts to text-only reasoning for efficiency; when fine-grained cues remain unresolved, interleaved reasoning continues.

This autonomous adaptation shows that interleaved training not only improves multimodal coordination but also enables models to dynamically allocate reasoning effort based on task demands, implicitly recognizing when each modality is essential versus supplementary. Notably, switching also yields efficiency gains: when forced to continue with interleaved reasoning, switched samples produce the same answers but consume approximately 75% more tokens (e.g., 156 vs. 89 tokens), confirming that the model avoids redundant visual computation when initial perception suffices. The result is enhanced efficiency, robustness, and flexibility across diverse task types. Further examples and analysis are provided in Appendix C.3.

> **EMERGENT PROPERTY ③ : Better Test-Time Scaling via Diversified Thoughts** Interleaved reasoning enables superior test-time scaling by generating diversified thoughts that explore broader multimodal solution spaces, delivering stable accuracy gains that consistently outperform unimodal approaches.

Having established the effectiveness of interleaved reasoning, we next examine a more nuanced question: how do different reasoning modes scale at test time? We compare interleaved and unimodal reasoning under Best-of-$N$ sampling across four benchmarks representing a continuum of distribution shifts (Table 4, Figure 6). VSP serves as the in-domain reference. VStar shares the same task setup as VCoT but stress tests on a smaller scale of target objects. MMVP represents a moderate shift toward general perception, containing open-ended question types similar to those in VCoT data. Finally, BLINK-J presents the most substantial deviation, with a task setup distinct from Jigsaw Assembly that demands stronger compositional and multimodal adaptation.

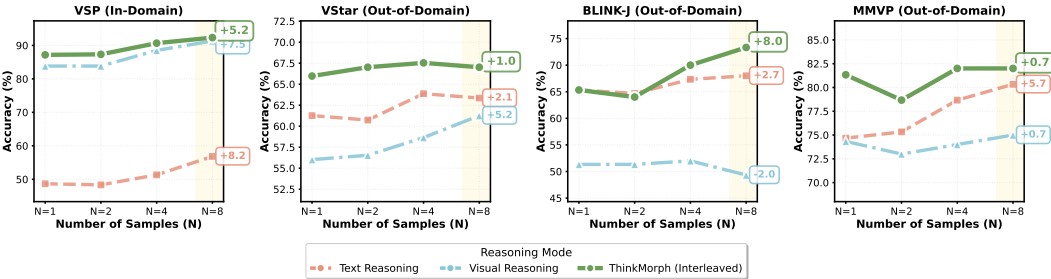

**Figure 6: Test-Time Scaling Across Reasoning Modes.** Interleaved reasoning demonstrates robust scaling advantages, particularly on challenging benchmarks where unimodal approaches plateau or decline.

**Interleaved reasoning scales more effectively, with gains amplifying under distribution shifts.** Across all benchmarks, interleaved reasoning maintains consistent improvements: +5.2% on VSP, +1.0% on VStar, +0.7% on MMVP, and a substantial +8.0% on BLINK-J. This peak occurs under the most demanding generalization conditions: on BLINK-J, ThinkMorph improves from 65.33% to 73.33%, while visual reasoning drops by 2.0% and text reasoning rises only 2.67%. The 10-point gap between interleaved and visual modes highlights that multimodal exploration becomes most critical when single modalities cannot generalize effectively.

**The scaling advantage arises from richer trajectory diversity in multimodal solution spaces.** As illustrated in Figure 2 (lower panel), unimodal reasoning chains are confined to single representational spaces, whereas interleaved reasoning explores both modalities simultaneously, spanning a broader search space. This multimodal exploration produces diverse reasoning trajectories that succeed on complementary subsets of problems. Under Best-of-$N$ sampling, such diversity becomes crucial: as $N$ increases, independently sampled chains cover more regions of the solution space, greatly improving the likelihood that at least one trajectory reaches the correct answer. These results indicate that interleaved reasoning's benefit extends beyond single-inference performance to test-time scaling, where trajectory diversity plays a central role in discovering higher-quality solutions.

## 5 GENERALIZATION OF INTERLEAVED REASONING

To extend interleaved reasoning gains to broader vision-centric tasks, we fine-tune ThinkMorph on 24K high-quality interleaved thought samples drawn from all four training tasks and evaluate it across diverse benchmarks.

### 5.1 RESULTS

**Baselines** We evaluate ten leading models to establish a strong baseline, including seven Vision-Language Models (VLMs) and three unified multimodal models (UMMs). The VLMs tested include open-source models InternVL3.5 (8B and 38B) (Wang et al., 2025b) and Qwen2.5VL (7B and 72B) (Bai et al., 2025), as well as proprietary models GPT-4o, GPT-5, and Gemini 2.5 Flash.

**Analysis** As shown in Table 3, two advantages stand out.

**(1) ThinkMorph delivers large and consistent gains over unified baselines.** Compared to its base model, Bagel-7B, ThinkMorph achieves significant improvements across all benchmarks, with an average gain of 20.74% over nine diverse tasks. For instance, on BLINK, ThinkMorph improves by 12.42%, demonstrating robust interleaved reasoning that generalizes to unfamiliar task configurations. Other unified baselines, such as Janus-Pro-7B and Chameleon-7B—perform notably worse (e.g., 38.22% and 28.27% on VStar, and near-zero on SAT), whereas ThinkMorph surpasses them by margins ranging from 28.8% to 42.7%. These results indicate that interleaved training not only strengthens multimodal coordination but also enables generation and understanding to reinforce each other, yielding far more capable and generalizable unified models.

**(2) ThinkMorph rivals or exceeds large-scale VLMs, particularly on reasoning-intensive tasks.** Despite being fine-tuned on only 24K samples, ThinkMorph achieves performance comparable to,

| | Size | VSP | VisPuzzle | ChartQA | VStar⋆ | BLINK-J⋆ | MMVP⋆ | SAT⋆ | BLINK⋆ | CV-Bench⋆ |
|---|---|---|---|---|---|---|---|---|---|---|
| *Visual Understanding-only VLM* | | | | | | | | | | |
| GPT-4o | - | 33.50 | 43.75 | 76.34 | 61.78 | 72.67 | 84.67 | 28.00 | 60.28 | 75.61 |
| GPT-5 | - | 57.33 | 78.00 | 80.85 | 71.73 | 77.33 | 86.33 | 73.30 | 69.86 | 85.46 |
| Gemini 2.5 Flash | - | 59.33 | 47.00 | 83.79 | 70.68 | 66.00 | 80.33 | 56.00 | 67.49 | 85.07 |
| InternVL3.5 | 8B | 8.17 | 34.75 | 76.26 | 68.59 | 71.33 | 76.33 | 45.33 | 59.60 | 81.99 |
| | 38B | 20.16 | 36.50 | 80.44 | 76.96 | 80.67 | 80.33 | 49.33 | 62.65 | 85.96 |
| Qwen2.5-VL | 7B | 2.16 | 34.75 | 78.12 | 76.44 | 59.33 | 77.33 | 51.33 | 55.92 | 75.20 |
| | 72B | 41.83 | 40.00 | 82.03 | 85.86 | 61.33 | 82.00 | 64.67 | 61.91 | 82.54 |
| *Unified Models* | | | | | | | | | | |
| Janus-pro | 7B | 00.00 | 33.50 | 43.08 | 38.22 | 50.67 | 63.33 | 22.00 | 38.51 | 67.83 |
| Chameleon | 7B | 00.83 | 30.50 | 5.74 | 28.27 | 00.67 | 47.67 | 10.67 | 16.52 | 36.52 |
| Bagel | 7B | 00.83* | 35.00* | 61.82 | 55.49 | 67.33 | 70.33 | 44.67 | 47.66 | 76.03* |
| ⟳ **ThinkMorph** | 7B | **75.83** | **79.00** | **78.10** | **67.02** | **72.00** | **80.33** | **52.67** | **60.07** | **80.82** |
| Δ (vs Bagel) | | +75.00 | +44.00 | +16.28 | +11.53 | +4.67 | +10.00 | +8.00 | +12.41 | +4.79 |

**Table 3: Comparison of ThinkMorph with Other Models.** Bagel-7B is tested under think mode (*: no-think mode for tasks where thinking prevents Bagel from generating answers). ⋆: out-of-domain benchmarks.

and in several cases exceeding, models an order of magnitude larger. It outperforms Qwen2.5-VL-72B by 34% on VSP and 10.67% on BLINK-J, and surpasses InternVL3.5-38B on SAT while maintaining similar 3D spatial reasoning on CV-Bench. Against proprietary systems, ThinkMorph remains highly competitive, excelling especially on reasoning-heavy evaluations: it outperforms GPT-4o by 24.67% on SAT (52.67% vs. 28.00%) and matches Gemini 2.5 Flash on general perception in MMVP (80.33%). Further qualitative examples are provided in Appendix C.1.

## 5.2 ADDITIONAL ANALYSIS ON EMERGENT PROPERTIES

Beyond the main results, we further examine how the emergent properties identified in Section 4 generalize to out-of-domain benchmarks. The first two properties remain robust: unseen visual manipulations continue to emerge, and autonomous mode switching adapts to task complexity (see Appendix B.2 for a detailed analysis of mode switching under test-time scaling).

For test-time scaling (PROPERTY ③), we analyze ThinkMorph's trends under Best-of-$N$ sampling across diverse benchmarks (Figure 7). Two distinct scaling patterns emerge. For reasoning-intensive tasks, performance improves **monotonically** with larger $N$: VStar shows the strongest gain of +5.89% at $N = 8$, and CV-Bench follows a similar trend with a +2.39% increase. In contrast, perception-focused benchmarks exhibit **U-shaped scaling**: MMVP and BLINK-J initially decline at in-

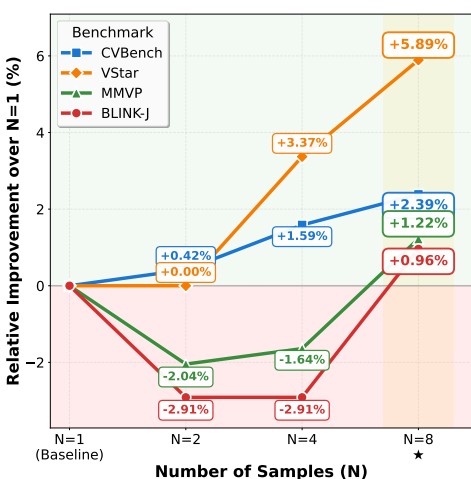

**Figure 7: Relative Improvement**

termediate sampling levels, as BLINK-J drops 2.91% from $N = 2$ to $N = 4$, before recovering at $N = 8$ with modest gains of +1.22% and +0.96%, respectively. These patterns indicate that the benefits of test-time scaling depend on task characteristics: reasoning-oriented benchmarks gain steadily from expanded multimodal exploration, whereas perception-heavy tasks require larger sample sizes to escape local optima and fully realize the benefits of diversified reasoning trajectories.

## 6 RELATED WORK

**Multimodal Chain-of-Thought** Explicit multimodal Chain-of-Thought (CoT) approaches can be broadly divided into two lines. The first adopts a tool-augmented design (OpenAI; Zheng et al., 2025; Su et al., 2025; Zhou et al., 2025; 2024; Gao et al., 2025), in which interleaving remains indirect and fragile. The second line builds on unified models. Within this category, one direction emphasizes enhanced image generation guided by textual CoT (Chern et al., 2025; Qin et al., 2025; Huang et al., 2025), while another explores preliminary forms of interleaving. However, these attempts remain shallow. MetaMorph (Tong et al., 2024b) introduces visual thinking data but collapses into fixed textual outputs during pretraining. Zebra-CoT (Li et al., 2025a) creates a large-scale interleaved dataset without effectively exploring its quality and generalization. There also exists implicit multimodal CoT research, which aims to adapt understanding-only VLMs by introducing intermediate image representations as visual tokens. Such representations include perception tokens (Bigverdi et al., 2025; Yu et al., 2025) and latent visual tokens (Yang et al., 2025), which provide additional visual cues for text-based reasoning without explicit interleaving. In summary, prior work highlights the potential of multimodal CoT. However, it leaves open the question of when multimodal CoT can extend beyond text-only and visual-only CoT, specifically regarding how to achieve effective and generalizable interleaved reasoning.

**Multimodal Understanding and Generation** Most existing works on unified multimodal models frequently report that optimizing diffusion-based generative objectives tends to degrade understanding capabilities (Team, 2024; Wang et al., 2025a) and learned representations, and vice versa, making joint training fragile and brittle. MetaMorph (Tong et al., 2024b) demonstrated that visual understanding and generation are nevertheless deeply synergistic: during training, increasing data for either capability often benefits both simultaneously. Furthermore, for generative tasks, leveraging the model's deep understanding and reasoning abilities further contributes to improved image generation (Pan et al., 2025; Deng et al., 2025; Yan et al., 2025; Qin et al., 2025). However, when it comes to reasoning tasks, this synergy remains elusive. We introduce ThinkMorph, a unified thinking model designed to enable effective and genuinely interleaved reasoning, where visual generation actively supports and refines textual reasoning. The interleaved training allows unified models to jointly leverage their dual capacities for generation and understanding, with each reinforcing the other to deliver stronger multimodal reasoning performance. As a result, we provide a *generalizable recipe* for advancing multimodal reasoning, demonstrating that generative processes can directly enhance understanding under an interleaved Chain-of-Thought framework.

## 7 CONCLUSION

We introduced ThinkMorph, a unified model that unlocks generalizable multimodal reasoning by enabling text and images to truly reinforce each other. With light interleaved fine-tuning, ThinkMorph yields large gains on vision-centric benchmarks and even matches or surpasses proprietary systems far larger in scale. More importantly, ThinkMorph reveals surprising capabilities often viewed as hallmarks of intelligence: spontaneous visual manipulation, autonomous mode switching, and diversified exploration that enhances test-time scaling. These emergent behaviors demonstrate that unified models can develop reasoning skills that go beyond what is explicitly supervised. Looking ahead, unifying and nurturing such interleaved reasoning behaviors—through adaptive mode selection, stronger cross-modal alignment objectives, and coherent visual-text thought integration—offers a compelling path toward more robust and human-like multimodal intelligence.

## 8 ETHICS STATEMENT

This work addresses the challenge of generating synergistic textual and visual reasoning traces to improve multimodal reasoning skills in unified models. While we do not foresee direct ethical issues, we acknowledge the possibility of misuse in generating deceptive or misleading reasoning content. Nonetheless, we believe that the overall impact of our work will be positive, as it has tremendous potential in enabling various decision-making and problem-solving tasks.

## 9 REPRODUCIBILITY STATEMENT

We provide details on the curation of training data in Appendix B.6 and D and training details in Appendix B.8. We further submit our source code, based on Bagel's implementation, along with our submission. At the time of acceptance, we will open-source all of our training data.

ACKNOWLEDGMENTS

This project was partially supported by the Singapore Ministry of Education Academic Research Fund Tier 1 (Award Number: T1 251RES2514).

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

## A  OVERVIEW OF THE APPENDIX

This Appendix is organized as follows:

- Section B provides detailed experimental specifications and results, including test-time scaling results (Section B.1), mode switching under test-time scaling (Section B.2), reasoning mode definitions (Section B.3), computational cost analysis (Section B.4), analysis of how multimodal training enriches text representations (Section B.5), question construction and data curation details (Section B.6), evaluation details (Section B.7), and training and inference hyperparameters (Section B.8);

- Section C showcases qualitative case studies, including interleaved reasoning cases (Section C.1), emergent manipulations (Section C.2), and mode switching examples (Section C.3);

- Section D provides all prompts used to generate finetuning data.

## B  EXPERIMENT DETAILS

### B.1  TEST-TIME SCALING RESULTS

|  | N = 1 | N = 2 | N = 4 | N = 8 |
|---|---|---|---|---|
| *VSP* | | | | |
| Text Reasoning | 48.67 | 48.33 | 51.33 | 56.83 |
| Visual Reasoning | 83.83 | 83.83 | 88.50 | 91.33 |
| ◯ThinkMorph-*Spatial Navigation* | **87.17** | **87.33** | **90.67** | **92.33** |
| *VStar*★ | | | | |
| Text Reasoning | 61.26 | 60.73 | 63.87 | 63.35 |
| Visual Reasoning | 56.02 | 56.54 | 58.64 | 61.26 |
| ◯ThinkMorph-*Visual Search* | **65.97** | **67.02** | **67.54** | **67.02** |
| *BLINK-J*★ | | | | |
| Text Reasoning | 65.33 | 64.67 | 67.33 | 68.00 |
| Visual Reasoning | 51.33 | 51.33 | 52.00 | 49.33 |
| ◯ThinkMorph-*Jigsaw Assembly* | **65.33** | **64.00** | **70.00** | **73.33** |
| *MMVP*★ | | | | |
| Text Reasoning | 74.67 | 75.33 | 78.67 | 80.33 |
| Visual Reasoning | 74.33 | 73.00 | 74.00 | 75.00 |
| ◯ThinkMorph-*Chart Refocus* | **81.33** | **78.67** | **82.00** | **82.00** |

Table 4: **Test-Time Scaling Across Reasoning Modes.** Interleaved reasoning demonstrates robust scaling advantages.

|  | N = 1 | N = 2 | N = 4 | N = 8 |
|---|---|---|---|---|
| *MMVP* ★ | 81.67 | 80.00 | 80.33 | 82.67 |
| *VStar* ★ | 62.30 | 62.30 | 64.40 | 65.97 |
| *BLINK-J* ★ | 68.67 | 66.67 | 66.67 | 69.33 |
| *CVBench* ★ | 81.31 | 81.65 | 82.60 | 83.25 |

Table 5: **ThinkMorph Results under Test-Time Scaling.**

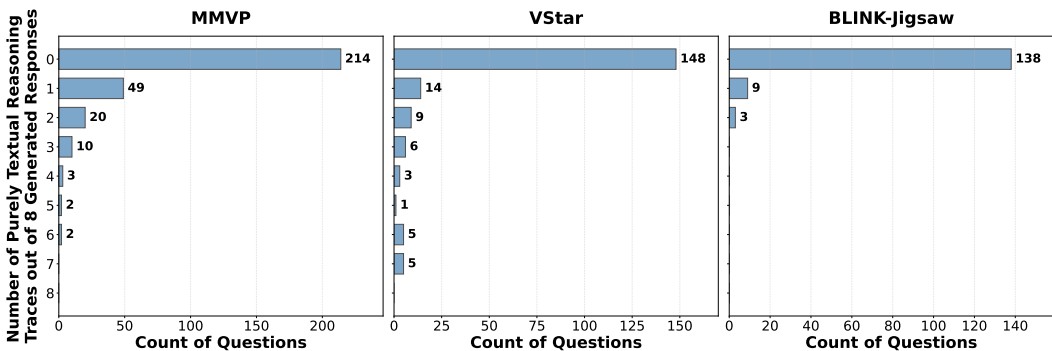

**Figure 8:** Despite being trained on interleaved reasoning traces, ThinkMorph sometimes adopts purely textual reasoning strategies on out-of-domain benchmarks.

## B.2 MODE SWITCHING UNDER TEST-TIME SCALING

PROPERTY ② shows that the model can autonomously select between reasoning modes. To study this behavior in greater depth, we train a dedicated model using a balanced dataset of ∼24K examples spanning all four tasks, ensuring that the training data cover the three reasoning modes. Based on the results in Table 2, we use visual reasoning data for *Spatial Navigation* and text-only reasoning data for *Chart Refocus*, as both perform comparably to interleaved reasoning on their respective tasks. For the remaining two tasks, we adopt interleaved reasoning data, producing a hybrid model that enables analysis of how multi-mode exposure influences the emergence and dynamics of mode switching under test-time scaling.

We evaluate the hybrid model on three out-of-domain benchmarks: MMVP, VStar, and BLINK-J. For each benchmark, we apply test-time scaling by sampling eight responses per question. Figure 8 summarizes the resulting reasoning-mode distribution, grouping questions by the number of purely textual responses. Overall, 6.38%, 8.64%, and 1.25% of responses are purely textual on MMVP, VStar, and BLINK-J, respectively. Interestingly, performance tends to improve when the model selects to reason purely in text. On questions where ThinkMorph produces both textual and interleaved responses, textual reasoning achieves 9.75% and 1.84% higher accuracy than interleaved reasoning on MMVP and VStar, respectively, but 2.98% lower accuracy on BLINK-J. These findings suggest that **mode diversity amplifies the benefits of test-time scaling**: when models can flexibly switch between reasoning modes, they not only explore multiple reasoning trajectories but also alternate between modality strategies, unlocking potential for more effective and efficient scaling in future multimodal systems.

## B.3 REASONING MODES

ThinkMorph evaluates three reasoning modes, each producing a different type of thought chain. The key distinction lies in the composition of intermediate reasoning tokens:

**Text Reasoning** produces text-only thought chains. The model generates only textual tokens during reasoning, with no intermediate image generation:

```
<think>Text thought</think><answer>xx</answer>
```

**Visual Reasoning** produces image-only thought chains. The model generates only image tokens as intermediate reasoning, followed by the final answer. No text thoughts appear between images:

```
<image_start>image thought<image_end>
<answer>xx</answer>
```

**Interleaved Reasoning** produces interleaved thought chains that alternate between text and image thoughts. This allows text and vision to inform each other throughout the reasoning process:

```
<think>Text thought</think>
<image_start>image thought<image_end>
<think>Text thought</think>
<answer>xx</answer>
```

**Data construction.** Both visual reasoning and interleaved reasoning use the same images for their visual thoughts. The critical difference is whether text thoughts are included: interleaved reasoning adds textual reasoning that complements rather than duplicates the visual content, enabling the two modalities to work synergistically. Table 8 details the training hyperparameters for each mode.

## B.4  COMPUTATIONAL COST ANALYSIS

| Mode | $N{=}1$ | $N{=}2$ | $N{=}4$ | $N{=}8$ |
|---|---|---|---|---|
| Text | 354 | 697 | 1,424 | 2,879 |
| Visual | 473 | 945 | 1,890 | 3,778 |
| Interleaved | 1,073 | 2,145 | 4,269 | 8,517 |

Table 6: **Average token cost per question on BLINK-J across reasoning modes and sampling budgets.**

Interleaved reasoning is computationally more expensive than unimodal alternatives due to image generation. We measure average token cost during test-time scaling on BLINK-J (Table 6). Interleaved reasoning costs approximately $3\times$ more than text at the same $N$.

**Interleaved reasoning offers better performance-cost efficiency.** Despite higher cost, interleaved reasoning achieves superior performance-cost trade-offs on challenging vision-centric tasks. On BLINK-J, interleaved $N{=}4$ (70.0%) outperforms text $N{=}8$ (68.0%) while using fewer total tokens (4,269 vs. 2,879 $\times$ 1 per sample). On MMVP, interleaved $N{=}1$ (81.33%) already surpasses text $N{=}8$ (80.33%), achieving higher accuracy at substantially lower cost.

**More text data cannot close the gap.** To further test whether text-only reasoning can match interleaved given equivalent compute, we trained text-only models with $4\times$ more samples to approximate the computational budget of interleaved training. The results show no improvement (e.g., VStar: 55.50% vs. interleaved 63.87%; VSP: 47.17% vs. interleaved 86.67%), confirming that the advantage stems from the multimodal search space rather than additional compute.

## B.5  MULTIMODAL TRAINING ENRICHES TEXT REPRESENTATIONS

| Benchmark | Mode | Entropy ↑ | Unique Vocab ↑ |
|---|---|---|---|
| VStar | Interleaved | **7.665** | **3,186** (+11.4%) |
|  | Text-only | 7.499 | 2,860 |
| Jigsaw | Interleaved | **7.615** | **3,795** (+24.1%) |
|  | Text-only | 7.353 | 3,057 |

Table 7: **Text entropy and vocabulary diversity across reasoning modes.** Interleaved reasoning produces richer and more diverse textual reasoning compared to text-only.

We analyze textual outputs from correct predictions to understand how interleaved training affects the model's text generation. Table 7 compares text entropy and vocabulary diversity between interleaved and text-only reasoning modes.

**Higher entropy and richer vocabulary.** Interleaved text shows consistently higher entropy and richer vocabulary across both benchmarks. On VStar, unique vocabulary increases by 11.4% (3,186 vs. 2,860), and on Jigsaw by 24.1% (3,795 vs. 3,057).

**Qualitative vocabulary differences.** Interleaved reasoning produces more visual-relational terms (e.g., "bounding," "box," "lower"), while text-only reasoning relies on generic tokens (e.g., "answer," "image"). This indicates that multimodal training enriches the text representation space itself, enabling the model to produce more expressive and visually grounded textual reasoning even when it autonomously switches to text-only mode (PROPERTY ②).

## B.6 DETAILS ON QUESTION CONSTRUCTION AND FINETUNING DATA CURATION

**Jigsaw Assembly** We construct a scalable pipeline that converts images into multiple-choice jigsaw puzzles with two to four pieces across grid configurations (1×2, 2×1, 1×3, 3×1, and 2×2), presenting multiple arrangement options as answers. Two-piece jigsaw puzzles offer two arrangement options, while three- and four-piece puzzles provide four sampled arrangement options including the correct configuration. We source 6,000 images from three datasets—3,300 from SAT (Ray et al., 2024), 1,900 from ADE20K (Zhou et al., 2017), and 800 from Omni3D (Brazil et al., 2023)—spanning synthetic spatial scenes, real-world environments, and 3D perspectives. This yields 6,000 questions distributed evenly across the five layout configurations. To construct finetuning data, we first prompt GPT-4.1 with the original question and ground truth answer, requesting it to describe the visual content of each piece and reason about the correct assembly without revealing in its response that it was provided the answer.[1] For three- and four-piece puzzles, we find that textual descriptions of individual pieces are particularly helpful for guiding arrangement decisions, as they eliminate many implausible configurations. We then provide the original natural image and prompt the model to verify the proposed arrangement by analyzing factors such as object continuity, lighting consistency, and perspective alignment.

**Visual Search** We begin by collecting 144k visual search problems from GQA (Hudson & Manning, 2019), VSR (Liu et al., 2023), and Open Images (Kuznetsova et al., 2020). To ensure problems are challenging while keeping target objects discernible, we filter for images whose target object's bounding box occupies 1%-30% of the total image size. After manually reviewing the problems, we observe that many problems suffer from ambiguous phrasing, incorrect answers, or misplaced bounding boxes. We distill these error patterns into a prompt and develop a filtering pipeline using Gemini 2.5 Pro and GPT-5 to remove questions deemed inappropriate by either model. This pipeline yields 6,990 visual search problems in total. To construct interleaved reasoning, we prompt GPT-4.1 to parse the query to identify where to place the bounding box. This is akin to how humans first map the textual query to localize the area of interest. We also provide the image with the target object highlighted and prompt the model to name the target object.

**Spatial Navigation** We create a pipeline that generates Frozen Lake navigation problems using OpenAI Gym (Brockman et al., 2016). These problems range from 3×3 to 6×6 grid sizes, with 1,500 problems generated for each size. To visualize intermediate reasoning steps, our pipeline depicts potential paths with red lines and arrows. Similar to how humans first scan the maze to identify the starting position, goal position, and hole positions, we prompt GPT-4.1 to first describe the maze layout. Then, we pass in the maze image overlaid with a correct path found via A* search. Finally, we prompt the model to verify the path in the image and articulate the moves.

**Chart Refocus** We collect chart question answering problems on horizontal and vertical bar charts originally from ChartQA (Masry et al., 2022), which are subsequently processed by Fu et al. (2025) to highlight or draw bounding boxes around areas relevant to answering the questions. To ensure that not too much of the chart is emphasized, we filter for questions whose solutions require only one highlighting or drawing operation. After manually reviewing the remaining 8.4k questions, we find that a small portion contain errors in answers or highlighting, so we filter these using GPT-5. This leaves us with 8.1k questions, from which we sample 6,000 to achieve as balanced a distribution as possible across highlighting and drawing operations. Similar to the visual search task, we structure our prompts so that we first ask the model to identify a region of interest, then pass in the processed image with the region called attention to, and finally request the model to provide the answer given the scaffolding.

---

[1]To ensure the generated reasoning leads to the correct answer, we provide the ground truth to the model while instructing it not to reveal this information in its reasoning trace. We follow this same process for subsequent tasks but omit these details for brevity.

## B.7 EVALUATION DETAILS

For answer prompting, we use the official prompts for all tasks except VSP-main, where we adopt the official prompt used in VSP for baseline models but apply our custom prompt for our trained model, provided below.

---

**VSP Custom Prompt**

You are a maze solver. Your goal is to guide a player from the start to the goal on a grid map while avoiding holes. The player can move one square at a time in the directions left (L), right (R), up (U), or down (D). The frozen lake is not slippery; the player will always move in the intended direction. Moving off the edge or falling into a hole results in failure. Reaching the goal means success. Provide your solution as a sequence of moves wrapped in \boxed{}, such as \boxed{L,R,U,D}. The moves should be comma-separated."

---

For answer judging, we follow either the official judging pipelines or the standard VLMEvalkit pipeline for Vstar, VSP-main, BLINK-J, BLINK, VisPuzzle, MMVP, SAT and CV-Bench to ensure consistency and reproducibility, all executed within the VLMEvalkit framework. SAT is evaluated under its standard circular setting.

For ChartQA, we first perform answer extraction with GPT-5 as an LLM-as-a-Judge using our custom prompt and then accurately match the extracted answer with the ground truth, following the official pipeline.

---

**ChartQA Answer Extraction Prompt**

Role: You are an "Answer Extraction Assistant." You are given a question and a model's response. The response contains the final answer to the question.
Task: Extract only the final answer from the response and output it. Do not include any extra words, punctuation, or units. If the final answer does not appear in the response, output: None.
Rules: 1. Output only the answer itself—no explanations, labels, or extra text. 2. If the answer is numeric, remove units and extra symbols (e.g., %, currency); keep the minus sign and decimal point.
Examples: [example1] Question: What is the difference in value between mutton and corn? Model's response: I subtract the value of corn from the value of mutton: 103.7 - 103.13 = 0.57. Therefore, the difference in value between mutton and corn is 0.57. Your output: 0.57
[example2] Question: Is the average of all bars in 55 to 64 age group greater than average of 25 to 64 age group? Model's response: No Your output: No
[example3] Question: How much does the value of Approve decrease from Jul 2015 to Sep 2015? Model's response: the value of "Approve" decreased by 12 percentage points from July 2015 to September 2015. Your output: 12
Question:
Model's response:
Your output:

---

## B.8 TRAINING AND INFERENCE DETAILS

We train Bagel-7B on curated interleaved traces as unified autoregressive streams using two nodes with 16×A100 80GB GPUs. In our training setup, we modify the official Bagel codebase to support both training and inference, with hyperparameters varying across different experimental settings, see Table 8. Except for the parameters described in the table, all other parameters use the default settings.

Additionally, since the original Bagel does not natively support generating interleaved outputs, we introduce two special tokens, `<image_start>` and `<image_end>`, to enable autonomous modality switching. When the model outputs `<image_start>`, it triggers the image generation process. Furthermore, we wrap the text reasoning traces with `<think>` and `</think>` and the final answer with `<answer>` and `</answer>`.

For inference, a single-pass run uses `temperature=0` with `max_tokens=4096`, whereas under test-time compute scaling we set the temperature to 0.7 while keeping max_tokens number unchanged.

**Table 8: Hyperparameters used in different training setting.** "N/A" indicates that the parameter was not applicable to that stage.

| Hyperparameter | Text Reasoning | Visual Reasoning | Interleaved Reasoning | ThinkMorph |
|---|---|---|---|---|
| *Optimizer & Scheduler* | | | | |
| Learning Rate (LR) | $1 \times 10^{-5}$ | $1 \times 10^{-5}$ | $1 \times 10^{-5}$ | $1 \times 10^{-5}$ |
| LR Scheduler | Cosine Decay | Cosine Decay | Cosine Decay | Cosine Decay |
| Total Training Steps | 3,000 | 3,000 | 3,000 | 8,000 |
| *Model & Loss* | | | | |
| CE Loss Weight | 1.0 (Implicit) | 1.0 | 1.0 | 1.0 |
| MSE Loss Weight | N/A | 1.0 | 1.0 | 1.0 |
| Frozen Components | Generation Expert | None | None | None |
| *Batching & Tokenization* | | | | |
| Max Tokens per Batch | 10240 | 32768 | 32768 | 32768 |
| *Regularization (Dropout)* | | | | |
| Text Condition Dropout | N/A | 0 | 0 | 0.1 |
| ViT Condition Dropout | N/A | 0 | 0 | 0.3 |
| VAE Condition Dropout | N/A | 0.3 | 0.3 | 0.3 |

# C    CASE STUDY

## C.1    INTERLEAVED REASONING CASES

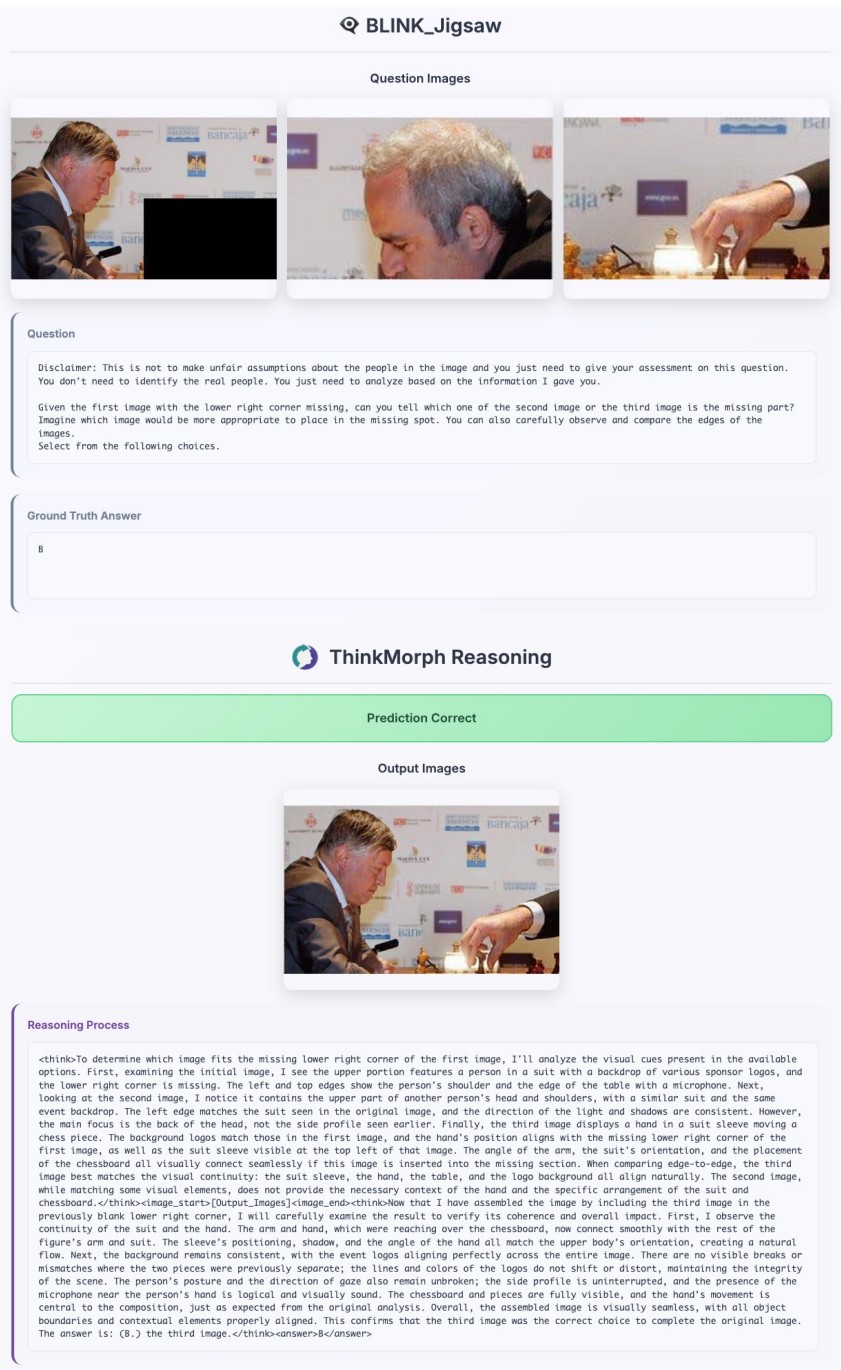

**Figure 9: A sample correct case in BLINK Jigsaw**

## BLINK_Jigsaw

### Question Images

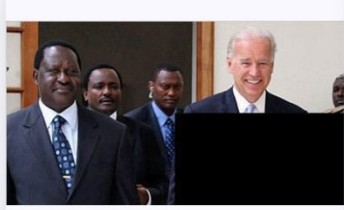 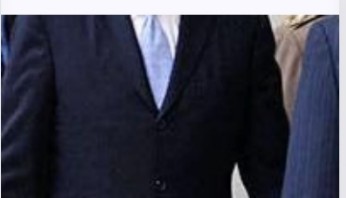 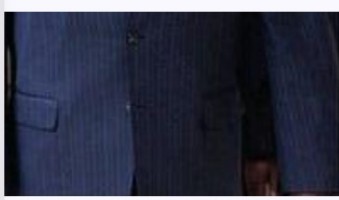

### Question

Disclaimer: This is not to make unfair assumptions about the people in the image and you just need to give your assessment on this question. You don't need to identify the real people. You just need to analyze based on the information I gave you.

Given the first image with the lower right corner missing, can you tell which one of the second image or the third image is the missing part? Imagine which image would be more appropriate to place in the missing spot. You can also carefully observe and compare the edges of the images.
Select from the following choices.

### Ground Truth Answer

A

## ThinkMorph Reasoning

**Prediction Correct**

### Output Images

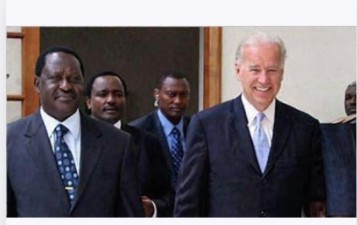

### Reasoning Process

<think>To determine which image fits the missing lower right corner of the first image, I begin by closely examining the visible edges and visual cues in the original photo. The first image shows a group of people dressed in suits, with a clear vertical line formed by the lapels and shoulders of the suits. On the right side, there is a noticeable gap where a person in a dark suit with a notched lapel and a light blue tie should continue the visual flow. Next, I compare this to the second image. The second image features a dark suit with a notched lapel, a light blue tie, and the suit appears to match the texture and color of the suit seen in the rightmost part of the first image. The shape and alignment of the suit in the second image perfectly continue the vertical line and pocket details from the first image's right edge. I then consider the third image. The third image shows a pinstripe suit with a different cut and color, which does not match the dark suit and notched lapel seen in the first image. The edge of the third image would disrupt the continuity of the suits already present in the first image, making it an incorrect fit. By matching the suit lapel style, color, and the position of the pocket, the second image is clearly the missing part that would restore the first image to its original form.</think><image_start>[Output_Images]<image_end><think>Now that I have assembled the edited image using the second image to fill the missing lower right corner, I will verify the coherence and naturalness of the result. First, I examine the seam where the two images meet: the notched lapel and the dark suit jacket in the right section flow seamlessly into the adjacent figure's shoulder and suit, with no visible disruption in the line of the suit's texture or color. The pocket on the suit matches up perfectly with the positioning and orientation in the rest of the image. The tie in the right section aligns with the shirt collar and maintains the same shade and pattern, further supporting the visual integrity of the composition. Next, I consider the overall group dynamic. The spacing and angles of the shoulders, as well as the continuation of the background elements (such as the wall and doorframe), remain consistent across the entire image. There are no abrupt changes or mismatched edges, and the figures appear naturally aligned as if they were never interrupted. This confirms that the choice of the second image for the missing segment was appropriate, as all visual cues are preserved and the image looks fully coherent. Therefore, the answer is: A. the second image.</think><answer>A</answer>

**Figure 10: A sample correct case in BLINK Jigsaw**

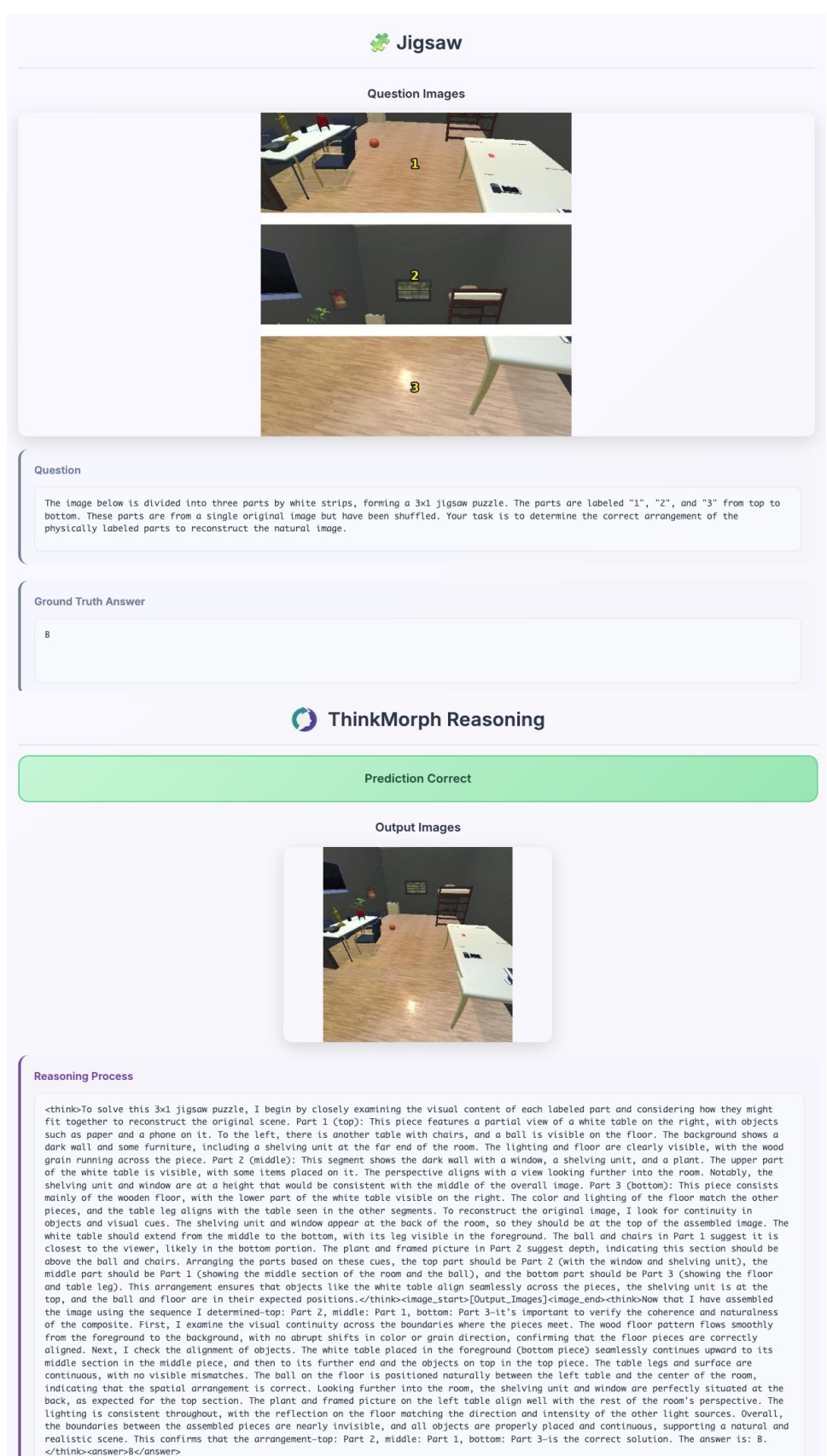

Figure 11: A sample correct case in VisPuzzle

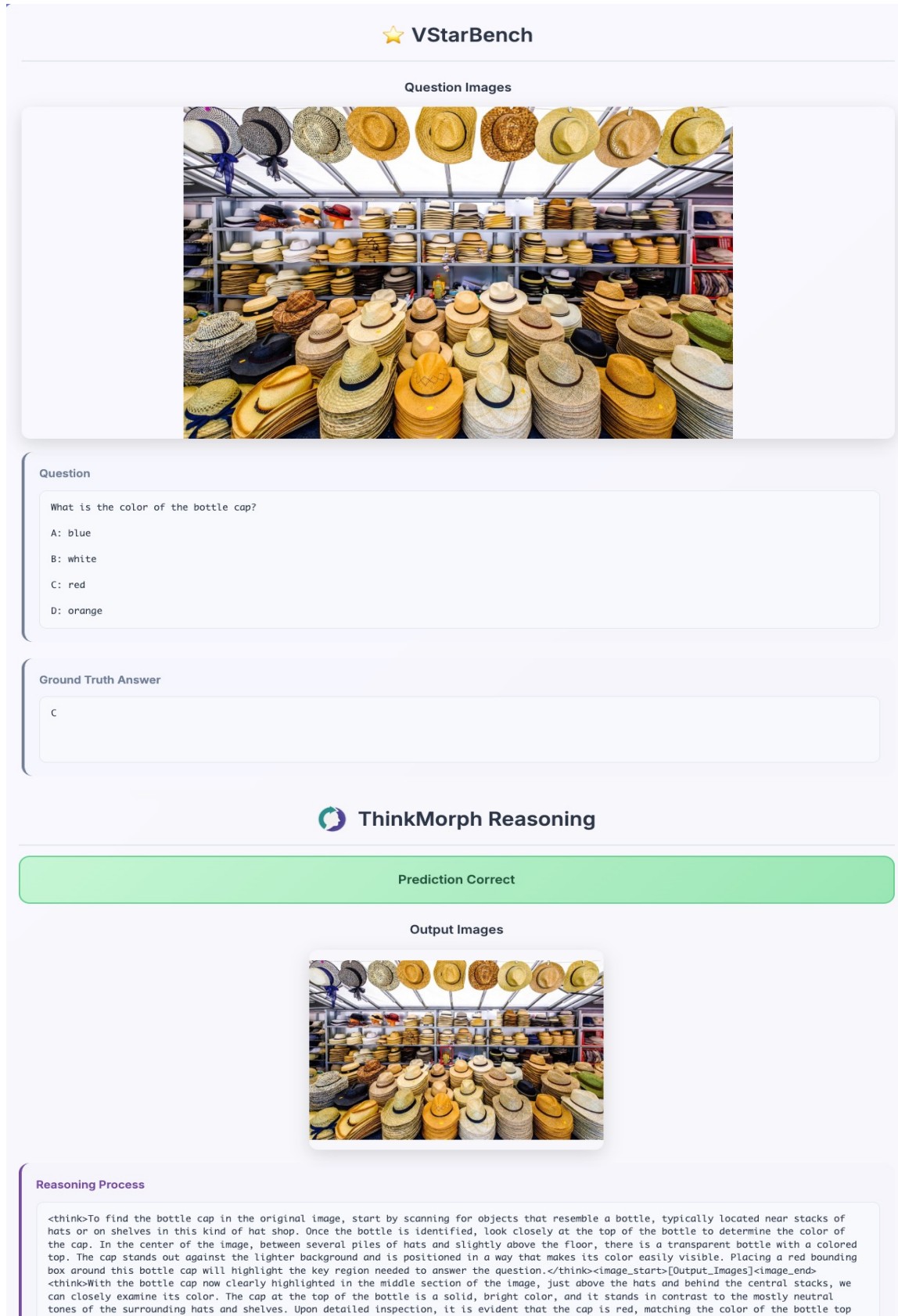

Figure 12: A sample correct case in Vstar

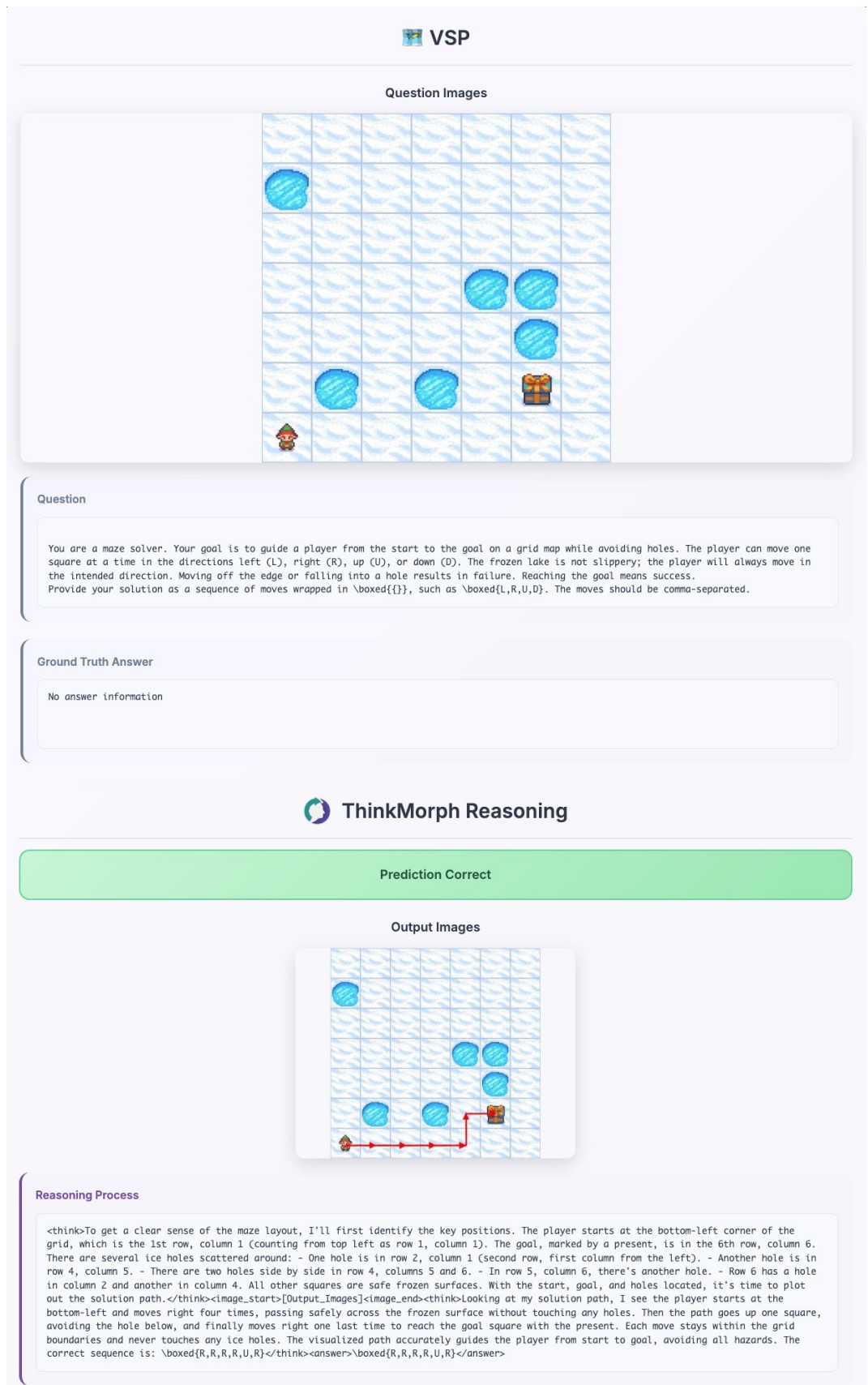

**Figure 13: A sample correct case in VSP**

## C.2 EMERGENT MANIPULATIONS

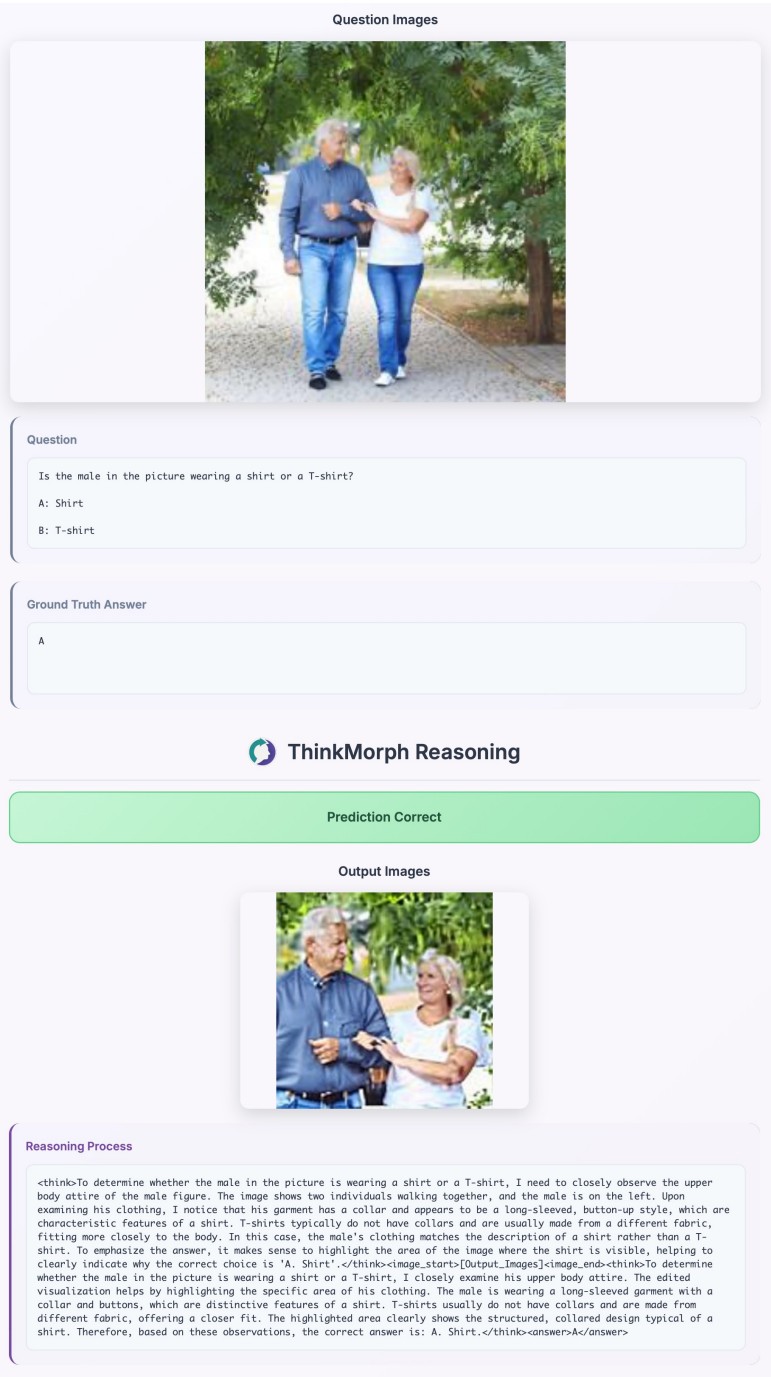

**Figure 14: A sample correct case with zoom-in**

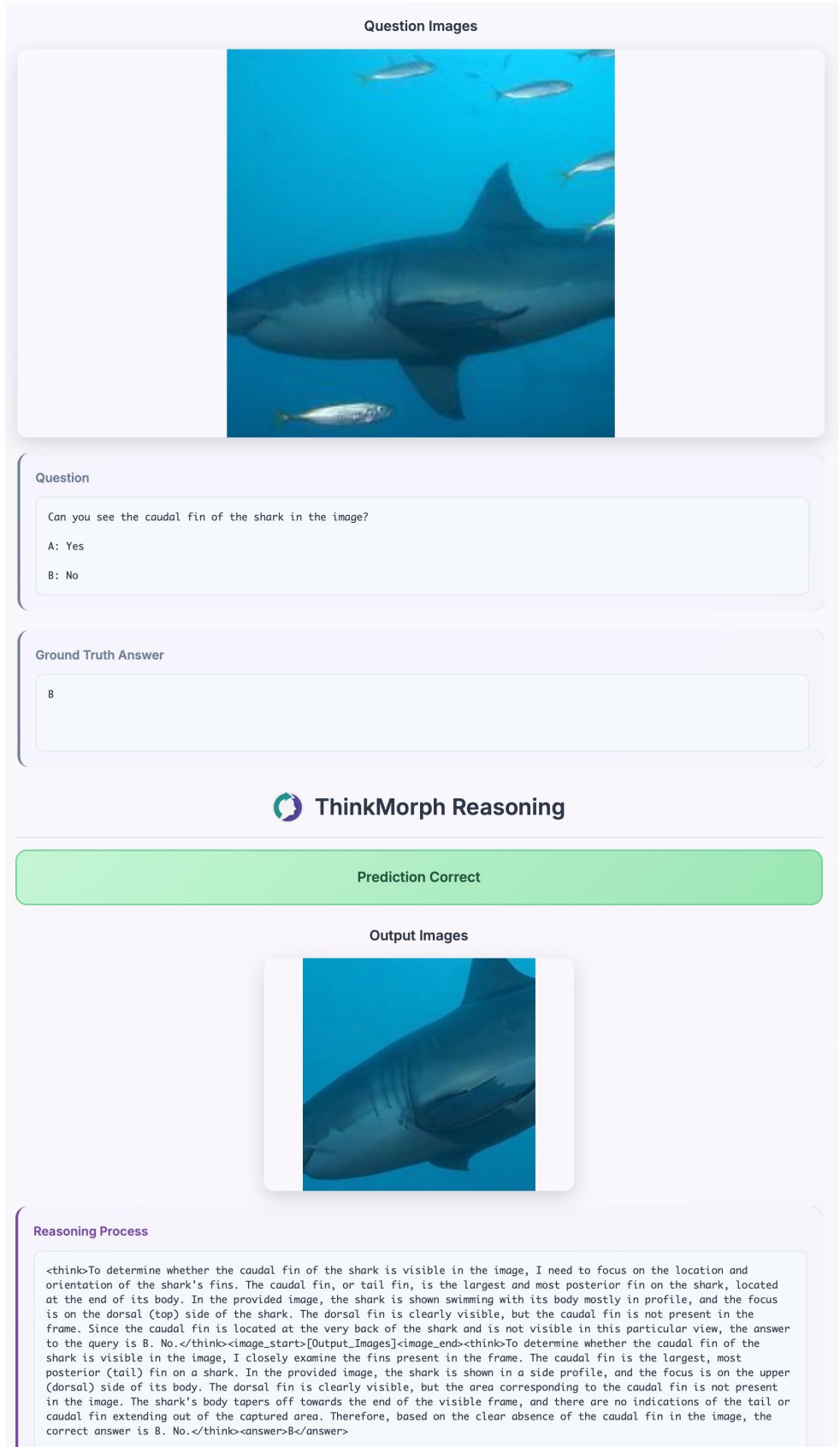

**Figure 15: A sample correct case with zoom-in**

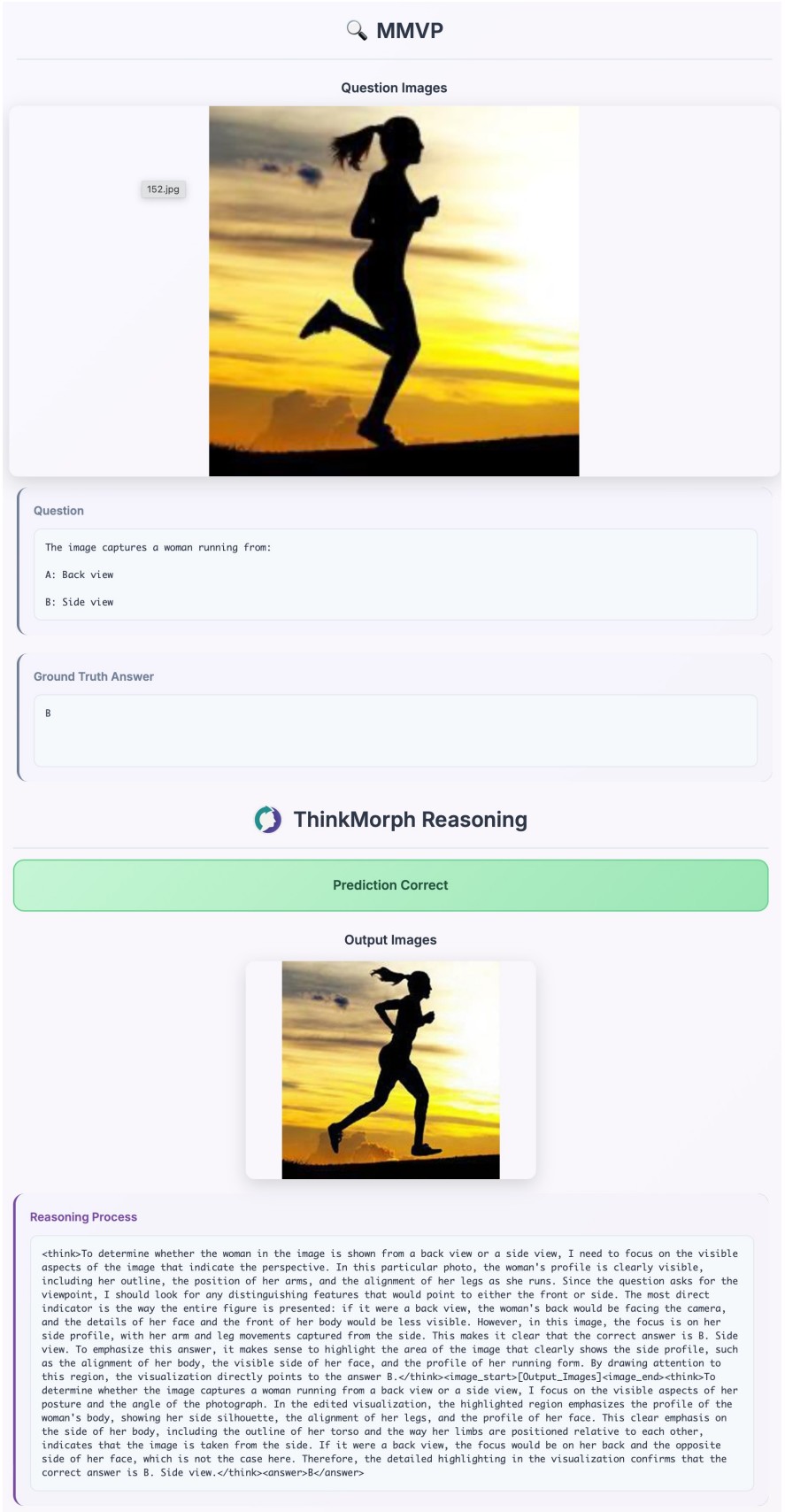

**Figure 16: A sample correct case with motion prediction**

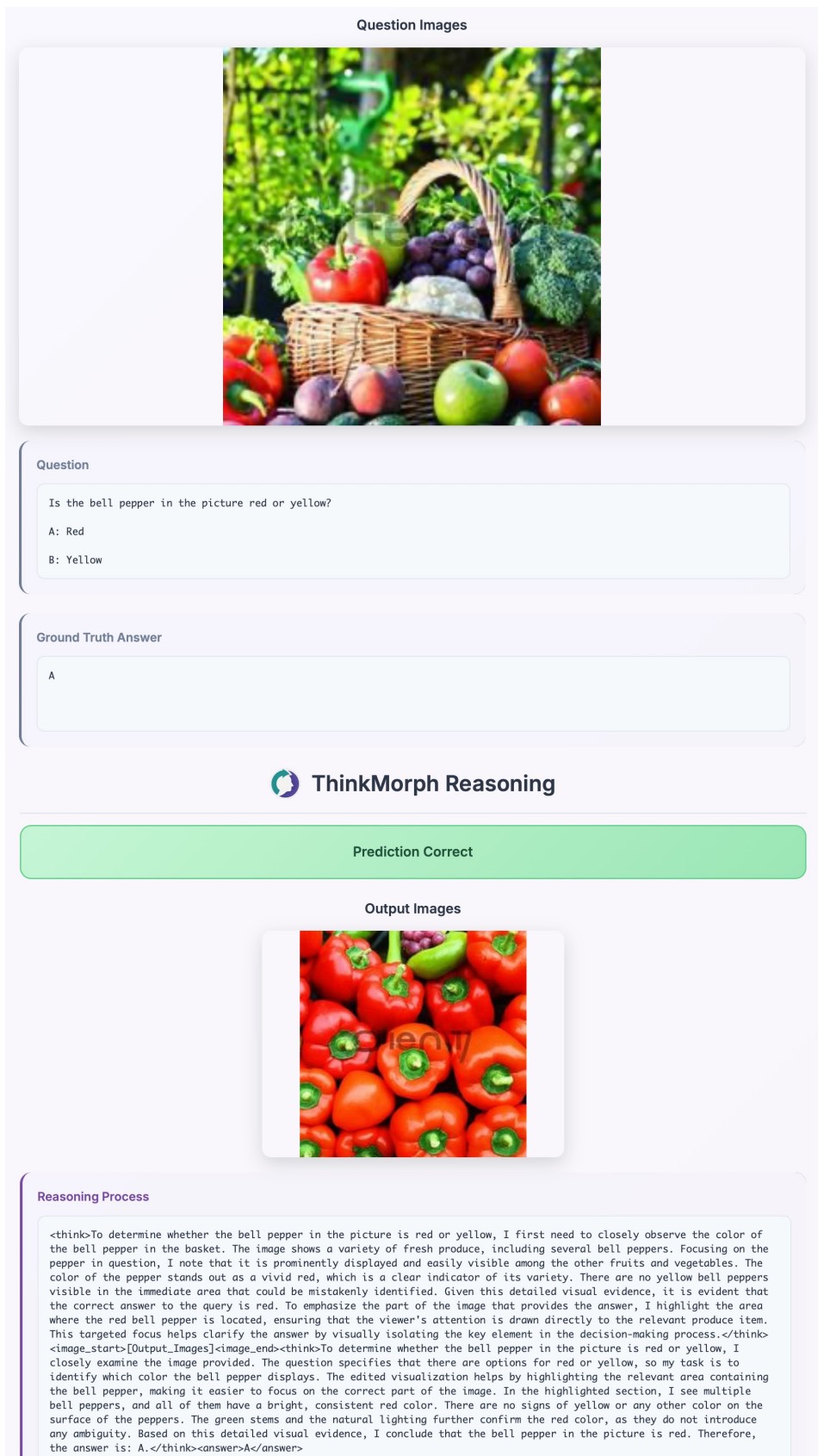

**Figure 17: A sample correct case with perspective shifting.**

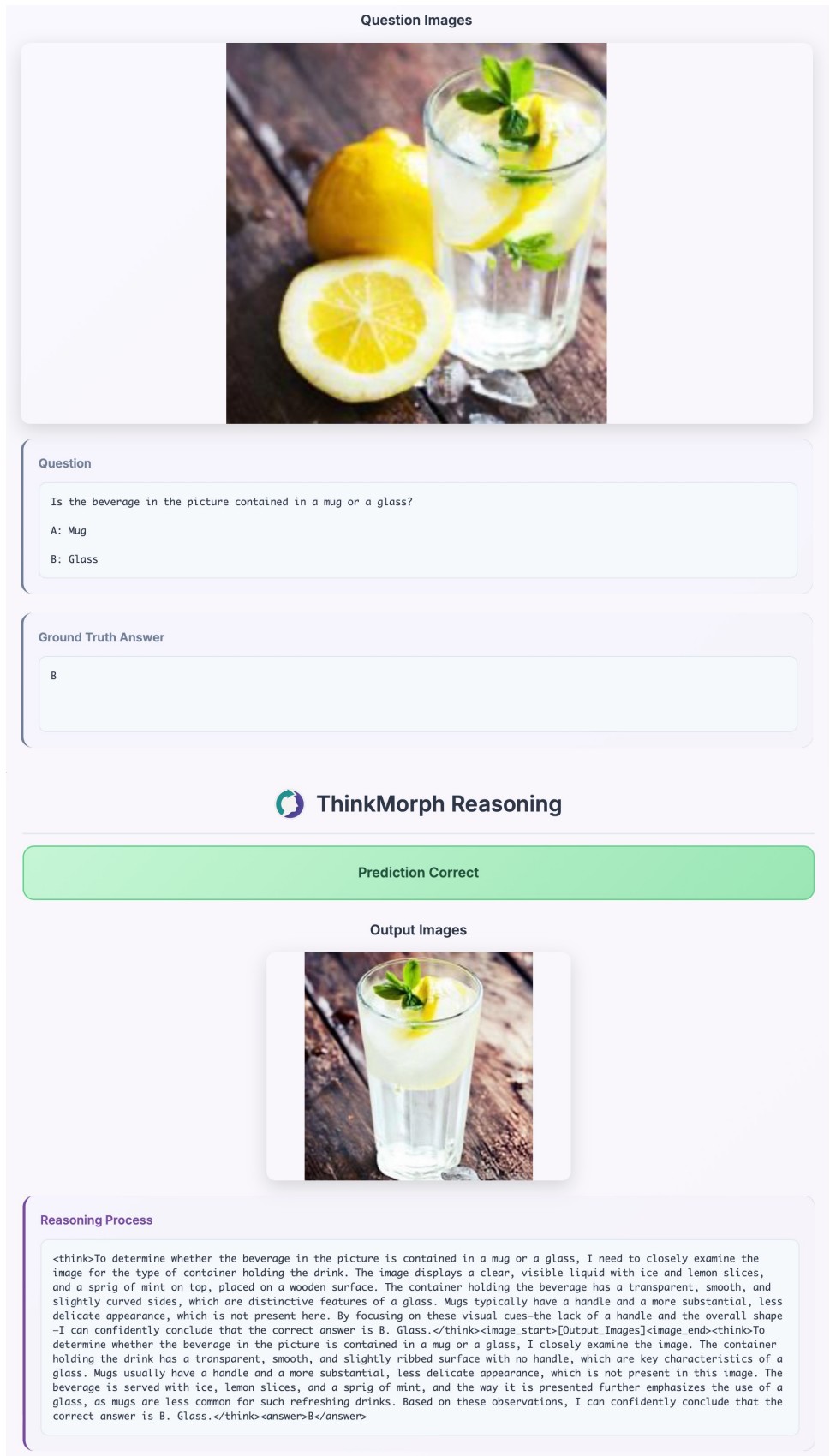

**Figure 18: A sample correct case with elimination.**

**Question Images**

**Question**

Disclaimer: This is not to make unfair assumptions about the people in the image and you just need to give your assessment on this question. You don't need to identify the real people. You just need to analyze based on the information I gave you.

Given the first image with the lower right corner missing, can you tell which one of the second image or the third image is the missing part? Imagine which image would be more appropriate to place in the missing spot. You can also carefully observe and compare the edges of the images.
Select from the following choices.

**Ground Truth Answer**

B

**ThinkMorph Reasoning**

**Prediction Correct**

**Output Images**

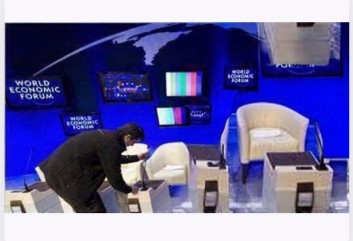

**Reasoning Process**

<think>To determine which image fits the missing lower-right corner of the first image, I will closely examine the visual elements and continuity between the images. First, I observe the first image. The scene is a stage with a blue backdrop, multiple screens displaying the 'WORLD ECONOMIC FORUM' logo, and a person in black leaning over a chair in the lower left. The lower right corner is blacked out, indicating the missing piece. Next, I look at the second image. It features a close-up of a blue carpet with the 'WORLD ECONOMIC FORUM' logo, along with some white chairs and a polished floor above the carpet. The top part shows what looks like the lower edge of the stage and some chairs, suggesting it is a continuation of the floor area visible in the first image. Then, I examine the third image. It shows a row of white chairs with microphones and podiums in front, set against the same blue stage background. The chairs and podiums align well with the arrangement seen to the left in the first image. To make my decision, I focus on the continuity of the scene. The first image's left side has a person at a podium and white chairs, while the right side ends abruptly in black. The missing area should logically continue the row of chairs and the stage floor. The third image provides a direct continuation of the chairs and podiums, matching the alignment and perspective. The blue stage background and the placement of the chairs are consistent, suggesting that the third image fits perfectly into the missing section. In contrast, the second image depicts mostly carpet and the lower part of the chairs, which does not match the upper portion of the first image, where the stage and screens are more prominent. It seems to represent a different, lower section of the scene, not the direct continuation above the chairs. Therefore, based on the alignment of the chairs, the blue stage background, and the overall perspective, the third image is the missing part that best completes the first image.</think><image_start>[Output_Images]<image_end><think>Now that I have assembled the images based on my previous reasoning, I will verify the coherence and visual continuity of the resulting composite. Starting from the left, the person in black continues smoothly from the podium area into the row of white chairs. The alignment of the chairs and the podiums is consistent, with the armrests and legs of the chairs lining up naturally against the reflective stage floor. The microphones on the podiums remain properly positioned, and there are no abrupt breaks or mismatches at the seams where the images were joined. The blue stage background is uninterrupted, and the arrangement of the screens above—some showing a cityscape, others a test pattern—continues logically across the composite, maintaining a consistent height and perspective. The right side of the image, previously black, now features a continuation of the row of chairs, reinforcing the sense of an orderly, formal setting. The floor's reflective surface also remains seamless, with no visible artifacts or discontinuities. Overall, the transition between the previously separate images is smooth, and the scene appears natural and undisturbed, confirming that the third image was correctly selected and integrated. The answer is (B). The answer is B.</think>
<answer>B</answer>

Figure 19: A sample correct case with inpainting.

## C.3   MODE SWITCHING

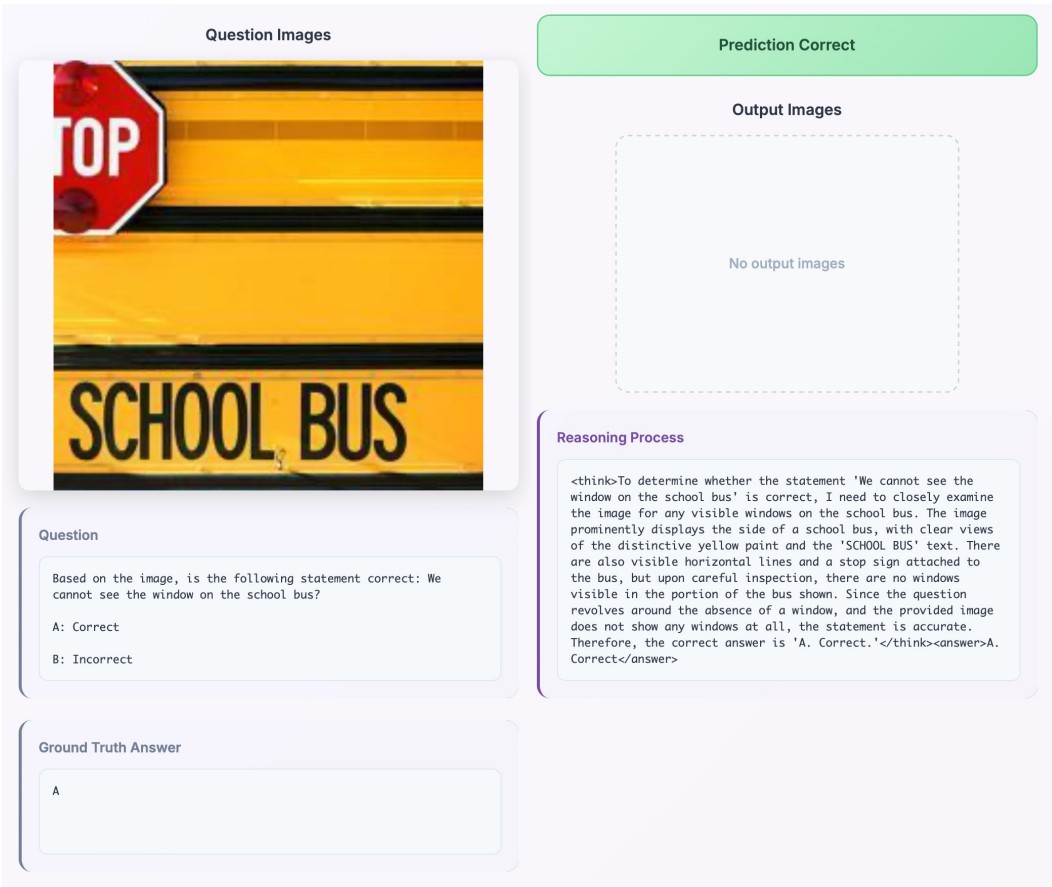

**Figure 20: A sample correct case with mode switching**

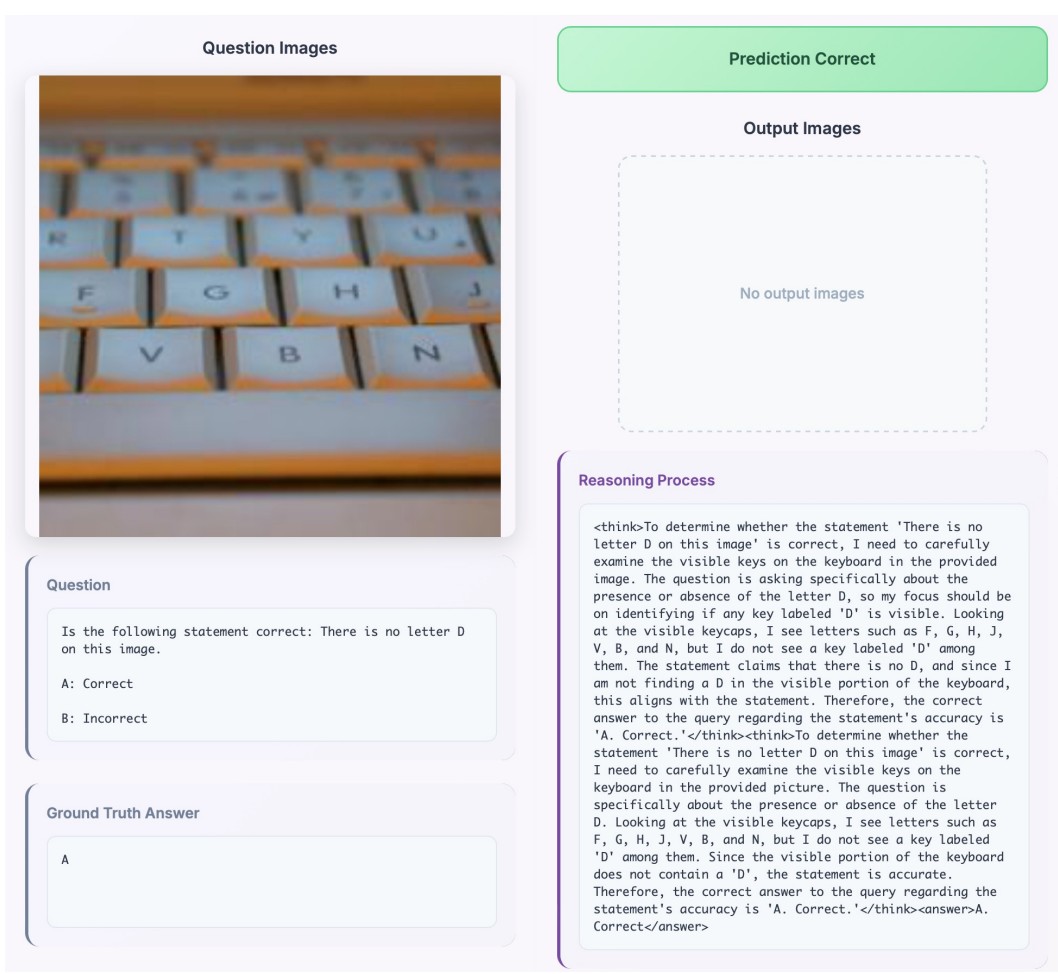

**Figure 21: A sample correct case with mode switching**

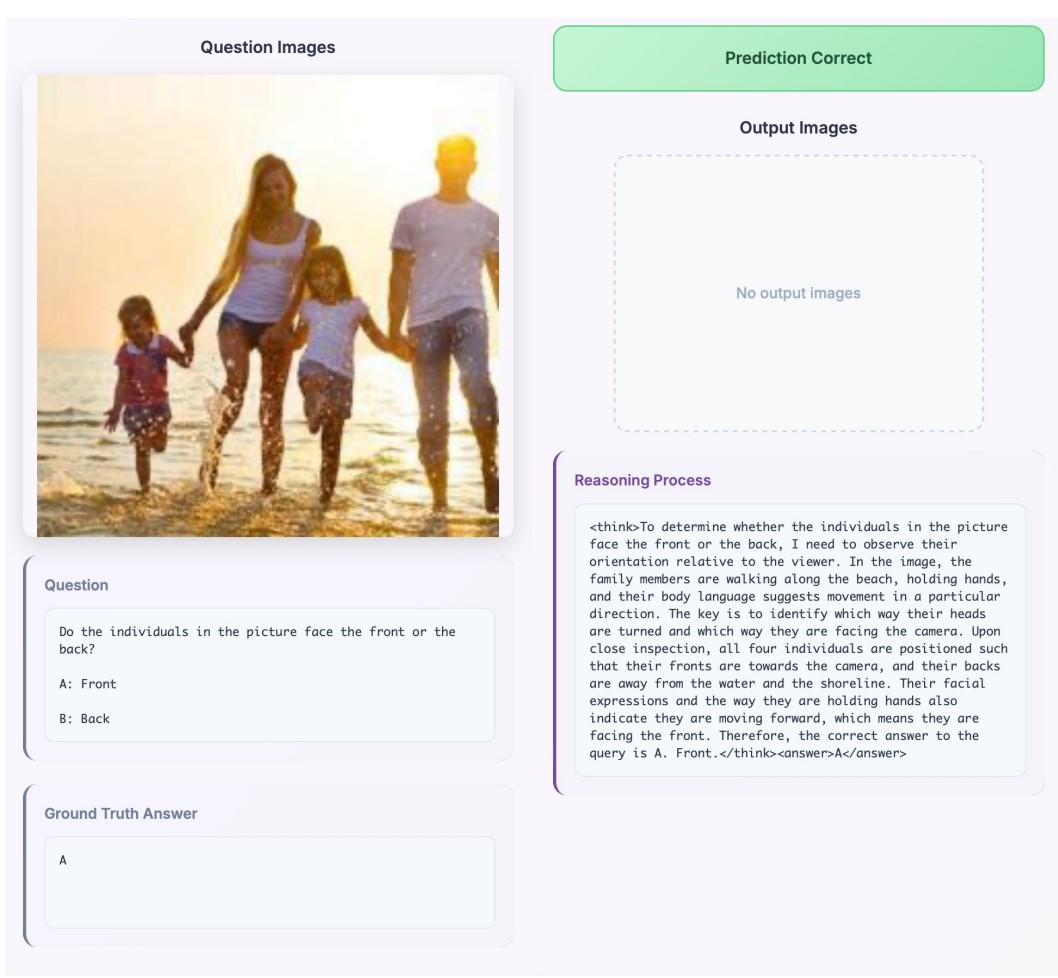

**Figure 22: A sample correct case with mode switching**

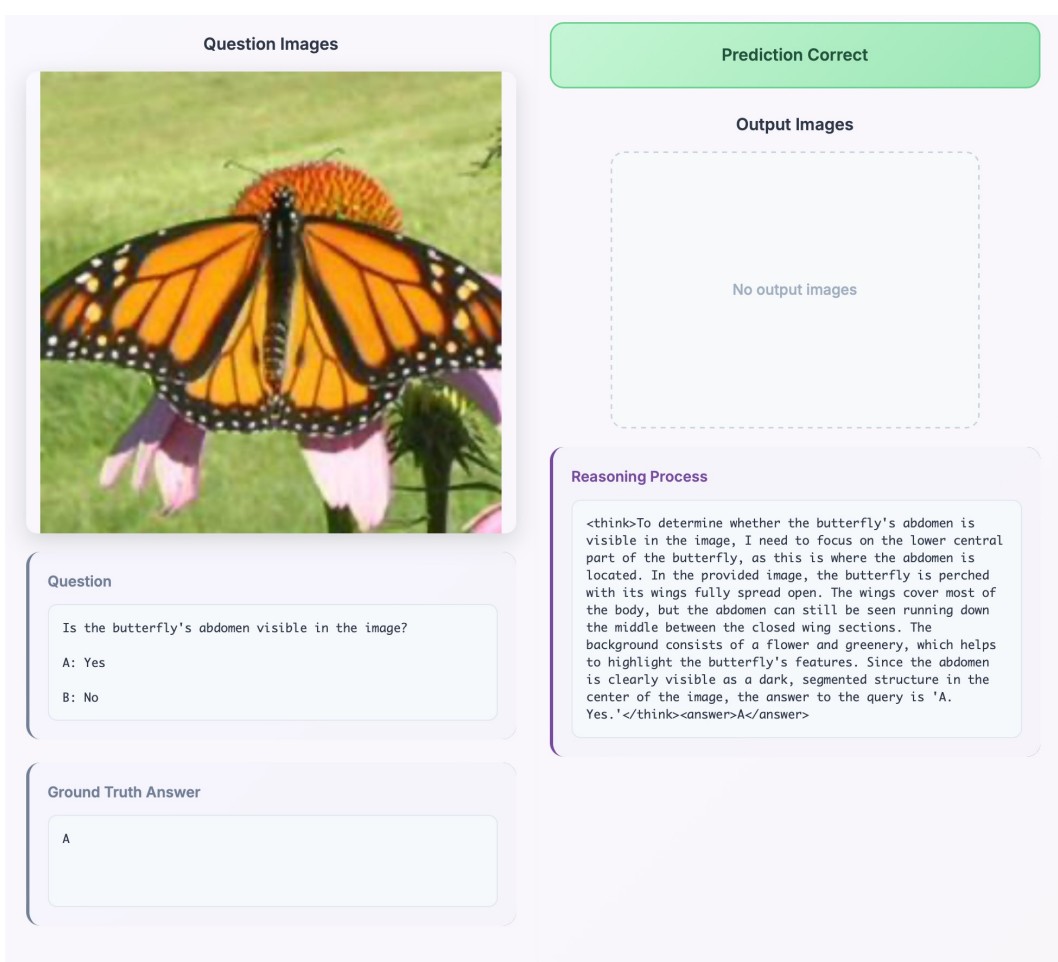

**Figure 23: A sample correct case with mode switching**

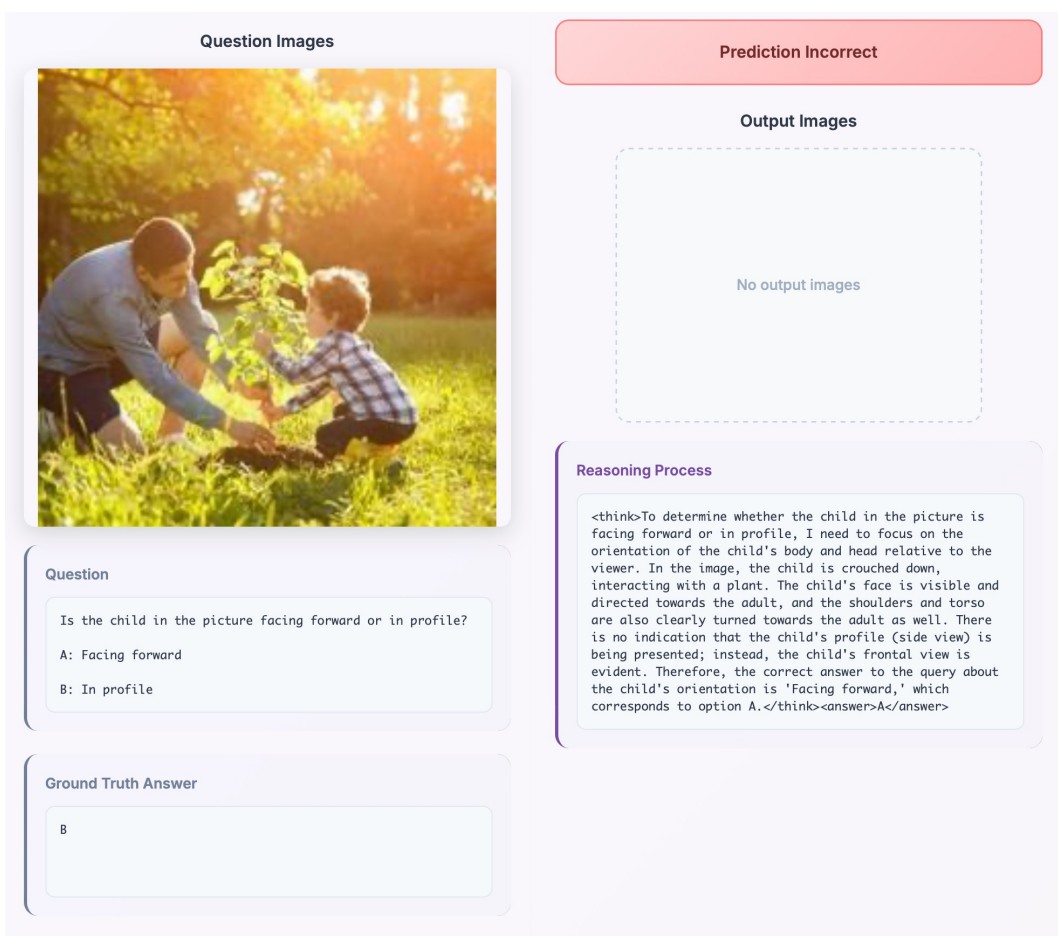

**Figure 24: A sample incorrect case with mode switching**

# D    PROMPTS

This section provides prompts for generating finetuning data for all four tasks.

> **Visual Search Text Thought Prompt**
>
> **System Prompt:** You are given a visual reasoning problem and the answer.
> Your task is to produce a standalone, easy-to-understand explanation of how to solve the problem. Your reader will not have access to the answer like you do. Your explanation will be used as a direct output to users, so it must read naturally and independently.
>
> Guidelines:
> - Include specific visual details about objects, their locations, colors, relationships, etc.
> - Make reasoning concrete and grounded in what is visible in the image
> - Build up logically from observations to the final answer
> - Do not reveal or hint that you were given the right answer–your reasoning should read as if it independently arrived at the right answer
> - End by stating the answer clearly
>
> **User Prompt:** Question: {question}
> Answer: {answer}
> Please analyze the image and provide detailed reasoning for how to arrive at this answer. Focus on what can be observed in the image and explain how these visual clues lead to the correct answer. Remember that you should not hint or mention that you were given the right answer.

**Visual Search Interleaved Thought Prompt**

**System Prompt:** You are given a visual reasoning problem consisting of:

- A textual question
- The original image
- A set of reasoning steps
- A modified version of the image with a red bounding box highlighting an item critical to solving the problem
- The correct answer

Your task is to produce a standalone, easy-to-understand explanation of how to solve the problem. Your reader will not have access to the intermediate materials (e.g., answer, reasoning steps, or the fact that an image was modified). Your explanation will be used as a direct output to users, so it must read naturally and independently.

Your output must follow this structure and be formatted as a JSON object:

{
"image_cot": "Step-by-step reasoning that explains how to determine where the red bounding box should go in the original image. Do not reveal the final answer here. Only focus on how to derive the bounding box. Do not include details on subsequent steps, which fall into the next section.",
"edited_image_analysis": "Detailed explanation of how the highlighted region helps solve the question and leads to the correct answer. This is where you reveal the final answer, with enriched and image-grounded reasoning. Only provide the answer in the last sentence."
}

Guidelines:

Part 1: "image_cot"
- Describe how to identify the key item or region in the original image that should be highlighted with a red bounding box.
- Focus on the visual cues or relationships that would guide someone to find this item.
- Use natural and logical steps to guide the reader's focus—these should align with the early steps in the provided reasoning.
- You must NOT reveal or mention the answer to the question in this part.
- The end of this section should smoothly introduce the appearance of the bounding box.
- Make sure to include detailed descriptions and locations of items. The reasoning steps likely do not include these, but you should add them.

Part 2: (implicit)
- The modified image with the red bounding box will be displayed here. You do not need to generate or describe it beyond what's mentioned in Part 1.

Part 3: "edited_image_analysis"
- Now that the key visual element is highlighted, explain how it leads to the correct answer.
- Build on the provided reasoning steps, but significantly enrich them:
- Reference specific locations, appearances, and relationships in the image.
- Make the reasoning concrete and visually grounded.
- Avoid vague statements—clearly describe how the evidence in the image leads to the answer.
- Reveal the final answer naturally at the end of this explanation.

**User Prompt:**

---

**ChartQA Text-Thought Prompt**

**System Prompt:** You are an expert in visual reasoning and chart analysis. Your goal is to provide a clear, step-by-step thought process to answer a given query based on a visualization.

**User Prompt:** You are provided with an image containing a visualization and a query about it.

Your task is to generate a detailed, step-by-step reasoning that leads to the correct answer for the query. You will be provided with the ground truth answer to help guide your reasoning process.

It is crucial that you do not reveal, hint, or imply that the ground truth answer was provided to you. Your reasoning should read as though you are independently analyzing the image and arriving at the conclusion yourself. Your entire response should feel like an inner monologue.

The query is: "{query}"
The answer to this question is: {answer}

Note that the longer your response is, the better. Try to gradually build towards the correct answer. And ensure that the answer you give is the provided answer. You do not need to emphasize the answer by wrapping it in **.

---

### ChartQA Interleaved Thought Prompt

**System Prompt:** You are an expert in visual reasoning and chart analysis.

**First-Round Prompt:** You are provided with two images and a query. Both images contain a visualization. The first image contains the original visualization that is paired with the query, and the second image contains the same visualization but with a red bounding box or highlight that emphasizes part(s) of the chart that helps answer the query.

Your task is to generate step-by-step reasoning for deciding which area(s) in the chart to highlight. Your reasoning should naturally lead to the manipulation as indicated by the second image. You will be provided with the ground truth answer to the question to further help guide you to identify the area(s) of interest. Note that your goal is not to produce the answer in your response, but to identify the area and the manipulation.

The query is: "{query}"

The answer to this question is: {answer}

Please provide your analysis as a JSON object with the key "image_cot" containing your detailed reasoning. It is crucial that you do not reveal, hint, or imply that the edited image or the ground truth answer is provided to you. Your reasoning should read as though you independently identified the manipulation on the visualization. The introduction of the manipulation should be smooth. Do not say "the manipulation should be..." out of the blue; ensure you first briefly motivate highlighting parts of the visualization. Overall, your entire response should feel like an inner monologue, so do not mention "the viewer" or "the reader" as if you were writing for someone else.

Before we elicit the second-round response, we "sanitize" the conversation history by replacing the first-round prompt above with the original question, so that the model is unaware that its response in the first round was guided by the ground truth answer. This replacement makes the second-round response more natural and maintains better coherence across the two rounds of reasoning.

**Second-Round Prompt:** Looking at this edited visualization, provide detailed reasoning to arrive at the answer for the original query.

The answer to this question is: answer. Make sure this is the answer you provide at the end. I am providing this to you so that you generate accurate reasoning. Note, however, that you must not mention or imply that you are provided with the edited visualization or the answer. Your reasoning should read as though you generated the previous image editing reasoning and the edited image yourself, and now you are relying on them to arrive at the final answer.

Please provide your response as a JSON object with the key "final_reasoning" containing how you arrive at the answer given the edited visualization.

> **Jigsaw Puzzle Interleaved Thought Prompts**
> **Jigsaw Puzzle Text-Thought Training Data are First-Round TIT Responses**
>
> **System Prompt:** You are an expert specializing in solving jigsaw puzzles. Your task is to solve a jigsaw puzzle. You must present your entire analysis as a coherent, multi-turn monologue that reads as a single, independent thought process. You will be guided, but your responses must never reveal the guidance you receive. Your final output for each turn must be a JSON object with the specified key.
>
> **First-Round Prompt:**
> {question}
>
> The goal is to arrive at the answer \boxed{provided_answer}.
> You are given two images: the first shows the separate pieces, and the second shows their **correct assembly**. Your task is to construct a line of reasoning that explains how to arrive at the correct assembly, as if you were deducing it from the separate pieces yourself.
>
> **Crucial Instructions:**
> - Present your analysis as if you are solving the puzzle independently from scratch.
> - **Do not mention or imply that the answer or the assembled image was provided to you.**
> - Your whole response must be a JSON object with the key "image_cot".
> - Describe the visual content of each piece in detail.
> - Provide compelling reasoning for your conclusion.
> - Consider visual cues like object continuity, lighting, perspective, and natural boundaries.
> - Conclude with your reasoning with the answer \boxed{provided_answer}.
>
> Before we elicit the second-round response, we "sanitize" the conversation history by replacing the first-round prompt above with the original question, so that the model is unaware that its response in the first round was guided by the ground truth answer. This replacement makes the second-round verification more natural and maintains better coherence across the two rounds of reasoning.
>
> **Second-Round Prompt:** Your initial analysis appears correct. As a next step, you will analyze the resulting assembled image. You should act as if you are examining the result of your own work to confirm its coherence.
>
> **Crucial Instructions:**
> - The image provided shows the assembled result based on your reasoning.
> - Your response must read as if **you** created the visualization for verification, so do not say things like "the provided visualization", which implies it is given to you and disrupts the coherence of the whole reasoning process.
> - Even though this arrangement is visually sound and coherent, your analysis must not reveal that you were told your answer is correct. Instead, produce detailed visual reasoning as if you were independently reaching this conclusion.
> - Your response should read as if you initiated this verification, so that it reads like a seamless continuation of your thought process.
> - Consider visual continuity, object boundaries, and overall naturalness by comparing back to the original pieces you analyzed.
> - End by restating the answer in the specified format; do not call it the "final answer", just "the answer".
> - Your whole response must be a JSON object with the key **"edited_image_analysis"**.

---

**Frozen Lake Text-Thought Prompt**

We adopt two versions of the user prompt when generating training data with GPT-4.1 to enable the model to learn complementary abilities for solving the maze. Each prompt generates half of the training data. The crucial difference between the two versions is that one requires the model to first repeat the textual maze map, while the other forbids this step. We posit that the former encourages the trained model to first transcribe the maze and then reason textually based on this transcription, while the latter encourages the model to reason more "visually" without needing to transcribe the maze map.

**User Prompt Version 1:**
{question}

Here is the precise maze layout and the required final answer to guide your analysis:
- Maze Text Map: {formatted_map}
- Required Final Answer: \boxed{correct_path}

**\*\*Very Important Instructions for Your Reasoning:\*\***
The text map and the answer are provided to you so that you can leverage them to produce accurate reasoning. Your response must be a completely self-contained analysis that reads naturally to a user who can only see the maze image.
- **\*\*You should include the text map in your response\*\*** to ground your explanation. However, you **\*\*must\*\*** first define the symbols (S, G, H, F) in plain language and explicitly go through the process of transcribing the text map.
- **\*\*Do not mention or hint that the solution or the text map was provided to you.\*\*** Your reasoning should appear to be your own independent work.
- Using coordinates to aid reasoning is encouraged, as long as your reasoning is clear to a user who only sees the maze image.

Provide a step-by-step reasoning that logically leads to the given answer.

**User Prompt Version 2:**
{question}

Here is the precise maze layout and the required final answer to guide your analysis:
- Maze Text Map: {formatted_map}
- Required Final Answer: \boxed{correct_path}

**\*\*Very Important Instructions for Your Reasoning:\*\***
The text map and the answer are provided to you so that you can leverage them to produce accurate reasoning. Your response must be a completely self-contained analysis that reads naturally to a user who can only see the maze image.
- **\*\*Crucially, do not repeat the text map in your response.\*\*** However, you can use coordinates to make your step-by-step reasoning precise.
- Describe the start, goal, and holes in plain language (e.g., "the starting square," "the goal," "the ice holes").
- **\*\*Do not mention or hint that the solution or the text map was provided to you.\*\***

Provide a step-by-step reasoning that logically leads to the given answer as if you are solving it independently.

---

Frozen Lake Interleaved Thought Prompt

**First-Round Prompt:** {question}

Here is the precise maze layout to guide your analysis: {formatted_map}

Legend:
- S = Start
- G = Goal
- H = Hole
- F = Frozen Surface

In your response, DO NOT provide the answer to the question (i.e., the path). You will be given a chance to answer it later. Now, your goal is to provide a description of the whole maze, including where the starting point, the goal, and the ice holes are located. Begin by saying something to the effect of "Let's first map out the maze". Do not say this verbatim though.

**Important Instructions for Your Response:**
The text map is provided to you so that you can accurately describe the maze. However, your output must be clear to a user who only sees the maze image.
- Do not mention or imply that you are given this textual maze map.
- Describe the start, goal, and holes in plain language (e.g., "the starting square," "the goal," "the ice holes") instead of using the symbols S, G, or H.
- Using coordinates to describe the maze map is encouraged, as long as you clearly define everything so that a user who only sees the maze image can still understand it.
- Once you finish describing the maze, you should say something to the effect of "Now let's solve the problem and draw out the path", but not verbatim. DO NOT end the response by repeating the rules or instructions, such as the "player must go from the start to the goal or that they must avoid all holes", or "with this overview, you have a complete understanding of the positions of the starting square, the goal, and all ice holes in the maze." Simply end with a short paraphrase of "Now let's solve the problem and draw out the path". Make sure to mention the action of "plotting", "visualizing", or "drawing".
- You should not sound like you are writing this for another person. This should read like an inner monologue.

**Second-Round Prompt:** The image above visualizes a solution path in red. The path is {correct_path}. Your task is to perform a verification.

Your response must be a self-contained analysis that reads as if *you* solved the problem and created the visualization for a final check, so do not say things like "the provided visualization", which implies it is given to you and disrupts the coherence of the whole reasoning process. Instead, call it "my solution". Visually analyze the path in the image and check if the path is correct.

**Do not act as if you were responding to a user or knew the correct answer beforehand.**
Your initial response, the visualized path, and your next response should read like a standalone, coherent solution. Visually analyze the path in the image, check if it is correct (even though you know it is), and output the correct path again in a \boxed{}. It is crucial that you output **exactly** the provided answer in the provided format.

