# OpenReview forum: "ThinkMorph: Emergent Properties in Multimodal Interleaved Chain-of-Thought Reasoning"
_ICLR.cc/2026/Conference — ICLR 2026 Poster_

### Official Review · Reviewer_qUFF · 2025-10-17

**Soundness:** 2
**Presentation:** 1
**Contribution:** 2
**Rating:** 2
**Confidence:** 5

**Summary:**

This paper proposes ThinkMorph, a unified multimodal reasoning model designed to perform interleaved reasoning across language and vision. The authors construct a high-quality data pipeline to generate interleaved reasoning traces. Through this training strategy, ThinkMorph reportedly achieves improvements on several multimodal reasoning benchmarks.

**Strengths:**

The paper constructs a high-quality interleaved reasoning dataset, which serves as a valuable resource for training and analyzing unified multimodal reasoning models. It also provides relatively complete implementation and reproducibility details, allowing others to replicate the experiments.

**Weaknesses:**

1. **Limited qualitative showcases.**
   The paper provides too few qualitative examples to substantiate its claims—particularly for the highlighted *emergent properties*. Demonstrations of abilities such as multi-bounding box generation, zoom-in, and crop operations are only shown in a single example (Figure 1), which is insufficient to convincingly establish the claimed emergent behaviors.

2. **Lack of detailed experimental settings.**
   The paper lacks clarity regarding the differences between experimental settings, especially between *visual reasoning* and *interleaved reasoning* as reported in Table 2.
3. **Unfair baseline comparison.**
   In line 344, the authors claim that “interleaved training consistently pushes unified models far beyond their previous limitations.” However, the comparison baseline appears to be the model that was **not** fine-tuned. A fairer comparison would involve a baseline fine-tuned on the same task using *text-only reasoning data*, so that the contribution of interleaved training can be more precisely isolated and validated.

4. **Ambiguity in adaptive modality selection.**
   The paper claims that ThinkMorph exhibits “adaptive modality selection,” where the model dynamically chooses between text-only, image-only, or interleaved reasoning depending on the task. However, there is no detailed discussion of image-only reasoning in the experiments, and the relationship between *image-only reasoning* and *visual reasoning* (as shown in Table 2) remains unclear, making this claim confusing and insufficiently supported.

5. **Questionable baseline results.**
   The reported GPT-4o result on SAT (28) in Table 3 is significantly lower than in the original paper, which raises concerns about the experimental setup or evaluation protocol. The discrepancy should be discussed and justified.

6. **Typos.**
There are several minor typographical and formatting errors in the paper. For example, in Table 5, “Reaonsing” should be corrected to “Reasoning.”

**Questions:**

1. Could the authors clarify how the *text reasoning*, *visual reasoning*, and *interleaved reasoning* models in Table 2 were obtained? Specifically, how were these variants fine-tuned within the Bagel framework, especially the details of the training data. And how do they relate to *ThinkMorph* as reported in Table 3?

2. In line 397, the authors mention several cases where the model “shifts to text-only reasoning.” Could the authors provide concrete examples or visualizations of these cases, and elaborate on why the model makes this reasoning shift?

---

> ### Author Response · Authors · 2025-11-23
> **[1/7] Manuscript Revision**
>
> We sincerely thank the reviewer for the thoughtful and constructive feedback.
>
> To address the concerns raised, we have carefully revised **[the manuscript](https://openreview.net/pdf?id=mB3vxfrQZM)** and enhanced the presentation for better clarity. These revisions include new evaluation results (**Tab.2 — Tab.5**), enhanced figures (**Fig.1 — Fig. 9**).  We also provide an [anonymous repository](https://anonymous.4open.science/r/ThinkMorph-rebuttal-71B2) with the data, training details, and rebuttal images for further transparency.
>
> We believe these improvements significantly address the core concerns and **invite the reviewer to examine [the manuscript](https://openreview.net/pdf?id=mB3vxfrQZM)**. Below, we provide detailed responses to each question.
>
> > **Note:** If the anonymous repository does not load, please try opening the link in an incognito browser window or clear your browser cache.
>
> [1/7]

---

> ### Author Response · Authors · 2025-11-23
> **[2/7] W1&Q2 Limited Qualitative Showcases and Analysis-1**
>
> [2/7]
>
> ### **W1&Q2 Limited Qualitative Showcases and Analysis**
>
> We thank the reviewer for this feedback. We have substantially expanded qualitative examples and deeper analysis of emergent properties in the revised manuscript.
>
> **Key revisions**:
> - Property 1: Fig.4, Sec 3.4 (L372—L401) and Appendix C.2 for more unseen cases
> - Property 2: Fig.5, Sec 3.4 (L402—L433) and Appendix C.2 for more swith cases
>
> ---
>
> **Property 1: Unseen Visual Manipulations** (W1)
>
> We now provide extensive qualitative examples beyond the original Fig.1. Fig.4 (more_unseen) shows **eight types** of unseen manipulations with concrete examples, including zoom-in, inpainting, multi-box generation, motion forecasting, perspective transformation, and region cropping. For instance, when asked "Is the bell pepper red or yellow?", the model automatically generates a zoomed-in view to distinguish subtle color differences, mirroring human visual inspection without explicit prompting. Appendix C.2 adds six full cases. Each case shows the complete reasoning chain: question, step-by-step text thoughts, generated images, and final answer.
>
> **Quantitatively**, these operations appear in up to 10% of all visual operations on some benchmarks. Importantly, these are not arbitrary artifacts but precise and task-effective visual actions that contribute directly to problem solving. A deeper analysis reveals systematic patterns: **specific textual cues reliably trigger corresponding manipulations.** Phrases like "examine closely" or "focus on" consistently elicit zoom-in operations, while terms like "restore" and "reconstruct" prompt inpainting. These correlations are both consistent and contextually appropriate, suggesting principled rather than random generation.
>
> ---
>
> **Property 2: Autonomous Mode Switching** (Q2)
>
> We demonstrate this property through **contrasting qualitative cases** in Fig.5 (case_mode_switch). The model adapts its reasoning behavior based on visual complexity:
>
> | Question | Visual Complexity | Model Behavior |
> |----------|------------------|----------------|
> | "Can you see stems of bananas?" | High (fine details) | Stays interleaved; generates zoom-in |
> | "Cannot see window on bus?" | Low (describable) | Switches to text-only |
>
> For the first question, the model maintains interleaved reasoning, generating a zoomed-in view of the upper region where the stem would appear. The close-up enables clear stem identification, showing that continuous visual engagement remains essential when fine-grained details are critical. For the second question, the model switches to pure textual reasoning, describing visible features like yellow paint and lettering to infer the absence of windows. This reflects **front-loaded visual engagement**: after processing the image and question, the model decides whether text alone can complete the reasoning. Appendix C.2 includes five more cases showing when switching saves computation and when staying interleaved is necessary.
>
> **Quantitatively**, despite being trained exclusively on interleaved data, the model autonomously switches to text-only on about 5.3% of cases on MMVP. These switched instances reach 81.25% accuracy, while staying interleaved on the same samples gives 73.96%, a gain of 7.29 points. This shows interleaved training improves multimodal coordination and enables models to dynamically allocate reasoning effort based on task demands, recognizing when each modality is essential versus supplementary.
>
> The mode switching property appears **consistently** across different training settings. We observe it in both single-task interleaved training (Tab.2, 6K samples) and ThinkMorph training on 24K mixed task samples. The property also emerges **robustly** across multiple OOD benchmarks, including MMVP, BLINK, and CV-Bench, even in test-time scaling.
>
> ---

---

> ### Author Response · Authors · 2025-11-23
> **[3/7] W1&Q2 Additional Analysis, W2 Differences Between Visual Reasoning and Interleaved Reasoning**
>
> [3/7]
>
> **W1&Q2: Additional analysis: switching also brings efficiency gains.**
> We also tested whether switching is driven by computational efficiency by forcing samples that switched to text-only to continue with interleaved reasoning instead. The final answers did not change, but computational cost increased significantly.
>
> > **Example:** For "Is the duck's entire beak visible?" the model performs early visual analysis and determines the answer is clear. It then switches to text-only.
>
> In the natural text-only path, the model answers in one round using 89 tokens and outputs “A. Yes.”:
>
> ```
> <think>To determine whether the duck's entire beak is visible in the picture, I need to closely examine the beak's position and orientation relative to the rest of the duck's head and the frame of the image. First, I observe the duck's beak in the provided photograph. The beak is prominently visible and extends outwards from the side of the duck's head. There is no portion of the beak that is cut off or cropped by the edges of the image. The beak appears to start at the base of the duck's face and continue smoothly into the foreground, with no breaks or obstructions. Given this assessment, it is clear that the entire beak is within the visible area of the photograph. Therefore, the answer to the question 'Is the duck's entire beak visible in the picture?' is 'A. Yes.'</think>
> <answer>A. Yes</answer>
> ```
>
> In the forced interleaved path, the model needs two rounds and 156 tokens to give the same answer “A. Yes,” which costs about 75% more tokens:
>
> > Round_1 of the forced interleaved path **repeats the same visual conclusion** reached in Round_0 ("beak extends directly out", "no visible cuts", "entire beak is within the visible area"). The extra image token (repeated original image) adds **no new information**.
>
> ```
> Round_0:
> <think>To determine whether the duck's entire beak is visible in the picture, I need to closely examine the beak's position and orientation relative to the rest of the duck's head and the frame of the image. First, I observe the duck's beak in the provided photograph. The beak is prominently visible and extends outwards from the side of the duck's head. There is no portion of the beak that is cut off or cropped by the edges of the image. The beak appears to start at the base of the duck's face and continue smoothly towards the right side of the photo, without any breaks or obstructions. Since the beak is fully within the visible area and there is no indication that it extends beyond the frame, I can conclude that the duck's entire beak is indeed visible. Therefore, the correct answer to the question is 'A. Yes.'</think>
> <image_start>
> [repeated original image]
> <image_end><think>To determine whether the duck's entire beak is visible in the picture, I closely examine the provided image. The beak extends directly out from the side of the duck's head and there are no visible cuts or crop marks at the beak's end. The beak appears to start at the base of the duck's face and continue smoothly towards the right edge of the frame, indicating that the entire beak is within the visible area. Since the beak is fully within the photo and there are no portions of it obscured or cut off, the correct answer is: A. Yes.</think>
> <answer>A</answer>
> ```
>
> **This shows two benefits of switching.** First, it is task-adaptive: the model uses vision only when extra visual analysis is really needed. Second, it is efficient: it skips unnecessary image generation when the first look is enough, using text-only for easy cases and interleaved reasoning for harder visual cases.
>
> ---
>
>
>
> ### **W2 Differences Between Visual Reasoning and Interleaved Reasoning**
>
> We have added Fig.2 to clarify the differences between experimental settings, especially between visual reasoning and interleaved reasoning in Table 2.
>
> The main difference between visual reasoning and interleaved reasoning is **whether text thoughts are included in the training data.**
>
> Interleaved reasoning:
> ```
> <think>Text thought</think>
> <image_start>image thought<image_end>
> <think>Text thought</think>
> <answer>xx</answer>
> ```
>
> Visual reasoning:
> ```
> <image_start>image thought<image_end>
> <answer>xx</answer>
> ```
>
> Text Reasoning:
> ```
> <think>Text thought</think><answer>xx</answer>
> ```
>
> **Visual reasoning contains only image generation and the final answer**. It has no text thoughts between images. Interleaved reasoning alternates between text thoughts and image thoughts, allowing text and vision to inform each other throughout the reasoning process. In data construction, the image thought of both is the same picture.

---

> ### Author Response · Authors · 2025-11-23
> **[4/7] W4 Clarification of "Image-Only Reasoning" and "Visual Reasoning"**
>
> [4/7]
>
> ### **W4 Clarification of "Image-Only Reasoning" and "Visual Reasoning"**
>
> We thank the reviewer for pointing out this terminology confusion. We clarify that **"image-only reasoning"** and **"visual reasoning"** refer to the same concept. The paper evaluates three reasoning modes, each producing different types of thought chains. We have now unified the terminology to consistently use "Visual Reasoning" throughout the manuscript to refer to the image-only reasoning mode.
>
> ---
>
> **(1) Three Reasoning Modes and Their Thought Chains**
>
> The three modes are:
>
> - **Text Reasoning**: produces text-only thought chains (contains only text tokens, no images)
>
> - **Visual Reasoning**: produces image-only thought chains (contains only image generation, no intermediate text thoughts)
>
> - **Interleaved Reasoning**: produces interleaved thought chains (alternates between text thoughts and image thoughts)
>
>
> As discussed in W2 (Differences Between Visual Reasoning and Interleaved Reasoning): Visual reasoning contains only image generation and the final answer. It has no text thoughts. Interleaved reasoning alternates between text thoughts and image thoughts. This lets text and vision work together during reasoning.
>
>
>
> ---
>
> **(2) Clarification of Experiments in Table 2**
>
> **Table 2 shows controlled comparison of all three modes.**  All modes are fine-tuned on the same tasks with the same amount of data. The only difference is the data format (text-only vs. image-only vs. interleaved). Visual reasoning represents the image-only baseline the reviewer expected. The results show interleaved reasoning consistently outperforms both text-only and image-only baselines on vision-centric tasks, with an average margin of **5.33%** over the next-best mode.
>
> ---
>
> **(3) Why Less Discussion of Visual Reasoning**
>
> The paper focuses less on visual reasoning (producing image-only thought) for two reasons.
> - First, Table 2 shows visual reasoning does not outperform interleaved reasoning on any of the four tasks.
> - Second, in autonomous mode switching behaviour, the model never switches to image-only reasoning. It only switches between text-only and interleaved modes.
>
> **From an effectiveness and efficiency perspective, this makes sense.** Switching to text-only saves computation when visual details are no longer needed. Switching to image-only would discard the efficiency of text while losing the flexibility of interleaved reasoning. Image-only reasoning is thus not adaptive in either direction.
>
> **To further study visual reasoning, we designed the hybrid model training (Sec 3.6)** that explicitly trains on all three modes. This allows us to examine whether the model learns to use visual reasoning in specific contexts. The results show that the model selects visual reasoning much less frequently than the other two modes. On some tasks, visual reasoning is not chosen at all, while on others it appears in small amounts (Sec 3.6 Mode Switching within Task ).
>
> ---

---

> ### Author Response · Authors · 2025-11-23
> **[5/7] W3 Unfair Baseline Comparison**
>
> [5/7]
>
> ### **W3 Unfair Baseline Comparison**
>
> We thank the reviewer for this clarification opportunity. We respectfully point out that we provide the exact fair comparison the reviewer requests. Table 2 provides controlled mode comparison and Table 3 shows generalization of ThinkMorph.
>
> **Key revisions**: Marked OOD benchmarks with ⭐ for better presentation in Table 2 and Table 3.
>
> ---
>
> **(1) Fair Comparison in Table 2**
>
> Table 2 provides exactly what the reviewer asks for: all three reasoning modes are fine-tuned on the same tasks with the same amount of data. The only difference is the data format.
>
> **Experimental setup**:
>
> - Text reasoning: Fine-tuned on ~6K text-only reasoning data per task
> - Visual reasoning: Fine-tuned on ~6K visual-only reasoning data per task
> - Interleaved reasoning: Fine-tuned on ~6K interleaved reasoning data per task
>
> This is the fair controlled comparison that validates the contribution of interleaved training. All three use the same base model (Bagel-7B) and same evaluation protocol. This isolates the contribution of interleaved training.
>
> **Results show interleaved consistently wins**:
>
> | Model                 | VSP   | VStar⭐ | VisPuzzle | BLINK-J⭐ | ChartQA | MMVP⭐ |
> | --------------------- | ----- | ------ | --------- | -------- | ------- | ----- |
> | Text Reasoning        | 49.17 | 56.02  | 63.50     | 68.67    | 81.66   | 76.33 |
> | Visual Reasoning      | 85.50 | 58.63  | 61.25     | 47.33    | 73.08   | 73.00 |
> | Interleaved Reasoning | 86.67 | 63.87  | 73.75     | 73.33    | 79.78   | 82.66 |
>
>  On vision-centric tasks (VSP, VisPuzzle, VStar, BLINK-J), interleaved reasoning yields a 34.74% improvement over the base model and surpasses the next-best mode by 5.33%.
>
>
>
> ---
>
> **(2) Line 344 Refers to Unified Model Baselines, Not Mode Comparison**
>
> The claim "interleaved training consistently pushes unified models far beyond their previous limitations"  refers to Table 3, where we compare ThinkMorph against other unified models like Bagel-7B (non-fine-tuned baseline), Janus-pro-7B, and Chameleon-7B.
>
> > Training setting: we train ThinkMorph on 24K interleaved data from all 4 tasks
>
> **This is a different comparison**:
>
> - Table 2: Fair mode comparison (all fine-tuned on same tasks)
> - Table 3: Generalization comparison (ThinkMorph vs. other unified models)
>
> The statement shows that interleaved training pushes unified models (like Bagel) far beyond their baseline capabilities. The fair mode comparison that isolates interleaved training's contribution is in Table 2.
>
> ---
>
> **(3) Strong OOD Generalization in Table 3**
>
> When , we see strong generalization to 6 OOD benchmarks (marked with ⭐):
>
> **OOD benchmarks test different capabilities**:
>
> - VStar: Compositional visual search with multi-hop reasoning
> - BLINK-J: Jigsaw puzzles in different visual domains
> - MMVP: Fine-grained visual perception
> - SAT: Abstract spatial reasoning
> - BLINK: General perception across 14 tasks
> - CV-Bench: 3D spatial understanding
>
> **Average improvement: +8.58% across 6 OOD benchmarks**. This shows interleaved training not only wins in controlled comparisons but also generalizes broadly.
>
>
> | Benchmark | Bagel-7B | ThinkMorph | Gain   |
> | --------- | -------- | ---------- | ------ |
> | VStar⭐    | 55.49    | 67.02      | +11.53 |
> | BLINK-J⭐  | 67.33    | 72.00      | +4.67  |
> | MMVP⭐     | 70.33    | 80.33      | +10.00 |
> | SAT⭐      | 44.67    | 52.67      | +8.00  |
> | BLINK⭐    | 47.66    | 60.07      | +12.41 |
> | CV-Bench⭐ | 76.03    | 80.82      | +4.79  |

---

> ### Author Response · Authors · 2025-11-23
> **[6/7] W5 SAT Baseline Results, W6 Typos**
>
> [6/7]
>
> ### **W5 SAT Baseline Results**
>
> The difference comes from different evaluation methods, not experimental problems.
> We follow the circular evaluation protocol described in the SAT paper ([Ray et al., 2024](https://arxiv.org/abs/2412.07755)), but in practice we adopt the **stricter implementation** used in `vlmevalkit`.
>
> ---
>
> **(1) Evaluation Protocol Difference**
>
> Specifically, in the SAT paper’s circular evaluation, each question is asked twice with both answer orderings (A–B and B–A), and the final score is the average of the two results.
> In contrast, in vlmevalkit’s circular evaluation, a question is counted as correct only if the model answers correctly under both orderings.
> For example, if the model answers correctly once and incorrectly once, the SAT paper would assign a score of 0.5, whereas our evaluation assigns 0.
>
> ---
>
>
> **(2) Re-evaluated Results**
>
> Your comment is absolutely valid, and we appreciate the careful observation. To ensure fair and consistent comparison, we re-evaluate all baseline models using the SAT paper’s protocol. The results are as follows:
>
> | Model | stricter_circular | looser_circular |
> |-------|--------------|-------------|
> | GPT-4o | 0.28 | 0.5533 |
> | GPT-5 | 0.733 | 0.8833 |
> | Gemini 2.5 Flash | 0.56 | 0.72 |
> | InternVL3_5-8B | 0.4533 | 0.6167 |
> | InternVL3_5-38B | 0.4933 | 0.6367 |
> | Qwen-2.5-VL-7B | 0.5133 | 0.6467 |
> | Qwen2.5-VL-72B | 0.6466 | 0.7367 |
> | Janus-pro-7B | 0.22 | 0.4867 |
> | Chameleon-7b | 0.1067 | 0.3167 |
> | Bagel (understanding) | 0.4466 | 0.6333 |
> | **ThinkMorph** | **0.5267** | **0.65** |
>
> **ThinkMorph reaches 52.67% under stricter-circular evaluation and 65% under looser-circular evaluation.** Under the looser setting, it still surpasses the Bagel baseline (63.33%) and remains competitive with larger models. ThinkMorph’s spatial reasoning generalizes well under **both** evaluation protocols, and its **relative gain is even larger** under the stricter circular evaluation.
>
>
>
> ---
> ### **W6 Typos in the table of training setttings**
>
> We thank the reviewer for catching these typos. We have corrected all typographical and formatting errors in the revised manuscript. The "Reaonsing" error has been fixed to "Reasoning." The original Table 5 is now Table 6, which includes more detailed hyperparameters for different training settings.
>
> ---

---

> ### Author Response · Authors · 2025-11-23
> **[7/7] Q1: Training and data Details**
>
> [7/7]
>
> ### **Q1: Training Details and Relationship Between Table 2 and Table 3**
>
> This response also addresses W2's concern about "Lack of detailed experimental settings." We have revised the manuscript to provide clearer explanations of training data curation (Sec 3.1, Appendix B.2, B.4, and D) and training settings. We also provide an [anonymous repository](https://anonymous.4open.science/r/ThinkMorph-rebuttal-71B2) with the data, training details for further transparency.
>
>
> ---
>
> **(1) Data Format Differences**
>
> **We have explained the data format differences between the three modes in W2 and W4**. In summary, text reasoning uses only text thoughts, visual reasoning uses only image thoughts, and interleaved reasoning alternates between both. In data construction, both visual reasoning and interleaved reasoning use the same images. The key difference is whether text thoughts are included.
>
> ---
>
> **(2) How Training Data Was Obtained**
>
> > **Table 2 shows controlled comparison of three reasoning modes.** All models start from Bagel-7B and use approximately 6K examples per task. The data comes from the same task sources but differs in format.
>
> **For text reasoning**, we prompt GPT-4.1 with ground truth guidance to generate step-by-step textual reasoning. The specific prompts are in Appendix D.
>
> **For visual reasoning**, training data consists of images containing visual manipulations paired with answers, without any textual reasoning. These images come from two sources. For ChartQA and Visual Search, we use images from the original benchmarks after filtering with GPT-5 and Gemini 2.5 Flash to ensure quality. For Frozen Lake and Jigsaw Puzzle, we generate images algorithmically.
>
> **For interleaved reasoning**, training data combines both modalities. Visual thoughts use the same images as visual reasoning. For textual thoughts, we design prompts to elicit reasoning that complements rather than duplicates visual thoughts, and query GPT-4.1. The philosophy behind prompt design is in Appendix B.2, and actual prompts are in Appendix D.
>
> ---
>
> **(3) How Models Were Fine-Tuned**
>
>
>
> All three modes are fine-tuned from Bagel-7B. Text reasoning uses only text generation loss (next-token negative log-likelihood). **Interleaved reasoning (Fig.2) uses dual-objective loss** that jointly minimizes both text generation loss and image generation loss (MSE between ground-truth and generated images). Visual reasoning uses image generation loss plus cross-entropy loss only for the follow-up `<answer>xx</answer>` generation. All the training hyperparameters used during training are presented in Appendix B.4, Table 6.
>
>
> ---
>
> **(4) Relationship Between Table 2 and Table 3**
>
> > **The models in Table 2 and Table 3 differ in training scope.** Table 2 trains 12 separate models, one for each combination of task (4 tasks) and reasoning mode (3 modes). Each model is trained on a single task using a single reasoning mode with approximately 6K examples. This isolates and compares reasoning mode performance within individual tasks.
>
> > **Table 3 trains a single ThinkMorph model** on interleaved reasoning traces across all four tasks, totaling approximately 24K examples. This multi-task model demonstrates generalization capabilities on out-of-domain benchmarks.
>
> **The key difference is scope.** Table 2 provides controlled comparison of three modes within individual tasks. Table 3 shows that when interleaved reasoning is scaled to multi-task training, the resulting ThinkMorph model generalizes strongly to diverse out-of-domain benchmarks, achieving an average gain of 20.74% over the base model across nine tasks.**The interleaved reasoning models in Table 2 can be viewed as single-task versions of ThinkMorph.** Table 3's ThinkMorph is trained on the combined data from all four tasks using only the interleaved format.
>
> ---
>
> **(5) Summary of Three Training Settings**
>
> For clarity, our work uses three training settings. **Setting 1 (Table 2)** uses single-task, single-mode training with 12 models (4 tasks × 3 modes) and approximately 6K examples each for controlled mode comparison. **Setting 2 (Table 3)** uses multi-task interleaved training with 1 model trained on approximately 24K interleaved examples from all four tasks to evaluate generalization.**Setting 3 (Sec 3.6)** uses hybrid mode training with 1 model trained on approximately 24K examples mixing all three modes across tasks to study autonomous mode switching behavior.

---

### Official Review · Reviewer_vxwn · 2025-10-26

**Soundness:** 2
**Presentation:** 3
**Contribution:** 2
**Rating:** 6
**Confidence:** 4

**Summary:**

This paper introduces ThinkMorph, a unified multimodal model designed to perform synergistic, interleaved Chain-of-Thought (CoT) reasoning that alternates between text and image generation. The core premise is that for vision-centric tasks (e.g., spatial reasoning, visual search), pure textual CoT is insufficient, and a dynamic interplay between language and vision can more effectively mimic human "think-and-sketch" reasoning.

**Strengths:**

- **Novel finding on interleaved reasoning**: A key contribution of the paper is the demonstration that training on interleaved text–image reasoning data enables the emergence of visual manipulation capabilities. This suggests that the model acquires a general-purpose mechanism for using visual representations as intermediate reasoning steps, rather than merely memorizing task-specific mappings.

- **Synthetic data curation methodology**: The paper’s effectiveness largely stems from its carefully designed data curation pipeline. A central insight is the construction of reasoning traces in which textual and visual components are complementary and non-isomorphic—an approach that offers a reproducible and valuable strategy for future work in multimodal reasoning.

- **Good empirical performance**: ThinkMorph achieves significant improvements over its base model and exhibits robust out-of-domain generalization. It attains competitive or superior performance compared to substantially larger models on several challenging multimodal benchmarks, including SAT and MMVP.

**Weaknesses:**

- **"Emergent Property 1" claim is overstated.** In my personal experience, the original Bagel model was already capable of zooming in/out and inpainting images when explicitly prompted—for example, given a prompt like "find the banana in the image," it could crop out the banana region from the input image. This capability appears to pre-exist in Bagel and is not necessarily a result of ThinkMorph’s training. Therefore, the claim of Emergent Property 1 lacks sufficient support, as the described behavior is already present in the base model.

- **"Autonomous Mode Switching" claim is conflated:** The most interesting claim is that a model trained exclusively on interleaved data can emerge a text-only reasoning mode. However, this is based on a "rare (2-5%)" observation. To analyze this "mode switching" further, the paper then pivots in Section 3.6 to a hybrid-trained model, which was explicitly trained on different modes. This is a much less interesting result and conflates the "emergence" claim with a simple "multi-task" training setup.

- **Scope of evaluated tasks:** While the four chosen tasks are well-justified, they represent a specific class of vision-centric, puzzle-like problems. The generalizability of this approach to more open-ended, creative, or dialog-based multimodal tasks remains an open question.

**Questions:**

- **Q1:** In my understanding, interleaved reasoning is more costly than single-modality reasoning. In Tab. 2, why are VSP-Main and VisPuzzle reported only in non-think mode for Bagel, whereas all think modes are reported for ThinkMorph?
- **Q2:** In Sec. 3.6, it says:
> This property extends our observation of autonomous mode switching (Emergent Property 2) to a
more dynamic setting. When generating multiple reasoning chains (e.g., N=8), a subset of samples
switch between text-only and interleaved modes as illustrated in Figure 2. Specifically, the modality
distribution changes as the number of candidate solutions increases. The proportion of pure textbased reasoning chains decreases from 18.8% at N=1 and N=2 to 15.2% at N=8. In parallel, the
contribution from interleaved chains—the dominant modality—increases from 81.2% to 84.8% over
the same range. This dynamic shift in modality distribution is directly correlated with an increase
in overall task accuracy, which improves from 51.3% at N=1 to 58.6% at N=8, as the modeling in
§ 2.3 with modality diversity.

Although the proportion of text-based reasoning decreases while the proportion of interleaved reasoning increases with more rollouts, where do the final accuracy gains primarily originate—from text-based or interleaved reasoning?
- **Q3:** The paper states that the interleaved-only model switches to text-only mode in 2-5% of samples, yielding "strikingly high accuracy, up to 90%". This is ambiguous. Does this mean 90% of that 2-5% subset of samples were correct? Or that 90% was the highest accuracy achieved on any sample in that subset? More importantly, why does the analysis in Sec 3.6 pivot to a hybrid-trained model? This feels like it sidesteps the more interesting emergence claim. Can you clarify the "emergence" of mode switching in the interleaved-only model, as this is a key claim?
- **Q4:** The Test-Time Scaling results are impressive. However, generating N interleaved chains, each containing images, is computationally expensive. Have you analyzed the performance vs. cost curve?

---

> ### Author Response · Authors · 2025-11-22
> **[1/8] Manuscript Revision**
>
> We sincerely thank the reviewer for the thoughtful and constructive feedback.
>
> To address the concerns raised, we have carefully revised **[the manuscript](https://openreview.net/pdf?id=mB3vxfrQZM)** and enhanced the presentation for better clarity. These revisions include new evaluation results (**Tab.2 — Tab.5**), enhanced figures (**Fig.1 — Fig. 9**).  We also provide an [anonymous repository](https://anonymous.4open.science/r/ThinkMorph-rebuttal-71B2) with the data, training details, and rebuttal images for further transparency.
>
> We believe these improvements significantly address the core concerns and **invite the reviewer to examine [the manuscript](https://openreview.net/pdf?id=mB3vxfrQZM)**. Below, we provide detailed responses to each question.
>
> > **Note:** If the anonymous repository does not load, please try opening the link in an incognito browser window or clear your browser cache.
>
> [1/8]

---

> ### Author Response · Authors · 2025-11-22
> **[2/8] W1 Clarification of Emergent Property 1- (1) The necessity of ThinkMorph’s training**
>
> [2/8]
>
> ### **W1 Clarification of Emergent Property 1**
>
> We thank the reviewer for this important clarification opportunity. We agree that **Bagel possesses raw manipulation capabilities from pretraining** (as we explicitly state in revision). However, we respectfully clarify that Emergent Property 1 refers to a fundamentally different phenomenon by **(1) The necessity of ThinkMorph’s training**, **(2) Two-Stage Mechanism** and **(3) Evidence of Systematic Patterns** .
> - **Key revisions**: Fig.4, Sec 3.4 (L372—L401) and Appendix C.2 for more unseen cases
>
> ---
>
> **(1) The necessity of ThinkMorph’s training**
> | Model | Perform manipulation when prompted?| Uses in reasoning? | OOD performance |
> |-------|----------------------|-------------------|-----------------|
> | Bagel | ✓ Yes | ✗ No | -|
> | ThinkMorph | ✓ Yes | ✓ Yes | +20.79% avg on 9 benchmarks |
>
>
> >  **(1.1) Explicit Prompting vs. Autonomous Reasoning**
>
> - **The reviewer correctly observes**: Bagel can perform zoom-in/cropping when explicitly prompted
>
> - **Emergent Property 1 claims**: ThinkMorph deploys these manipulations as reasoning aids within multi-step problem-solving, without explicit instructions.
>
> Side-by-Side Comparison: We provide direct empirical comparison using the bell pepper example:
>
> **Scenario A** : Bagel with Explicit Manipulation Instruction
> Prompt: *"Find the pepper in the image and crop it"*
> Nature: Direct instruction-following (manipulation as *task goal* for image editing)
>
> Bagel successfully crops the pepper region [[result_img]](https://anonymous.4open.science/r/ThinkMorph-rebuttal-71B2/rebuttal_figures/prompt_zoom_in.jpg) after thinking:
> ```
> <think>
> To find the pepper in the image, I first observe the question_image, which displays a variety of fresh vegetables. The question_text asks to locate the pepper, so I focus on identifying the distinct characteristics of a pepper. Peppers are typically round or slightly elongated with a smooth surface and a stem attached. In this image, there is a bright yellow pepper with a green stem, which stands out among the other vegetables.
>
> The answer_image would need to zoom in on the yellow pepper to clearly highlight its shape, color, and stem. The pepper is positioned near the center-right of the image, surrounded by other vegetables like kale, radishes, and an eggplant. The zoomed-in view should emphasize the pepper's vibrant yellow hue and its green stem, making it easily identifiable. This approach ensures the user can clearly see the pepper in the image.
> </think>
> ```
> #### **Scenario B: ThinkMorph on Reasoning Question**
> Prompt: *"Is the bell pepper in the picture red or yellow? A: Red B: Yellow"* , ThinkMorph's [[result_img]](https://anonymous.4open.science/r/ThinkMorph-rebuttal-71B2/rebuttal_figures/unseen_zoom_in.jpg) and response:
> ```
> Round_0:
> <think>To determine whether the bell pepper in the picture is red or yellow, I need to closely examine the color of the pepper shown. The image displays a variety of fresh vegetables, including leafy greens, radishes, and what appears to be a single bell pepper. Focusing on the pepper, I observe its surface and overall hue. The pepper has a bright, sunny color that stands out among the other produce. This color is consistent with the typical appearance of a yellow bell pepper, which is usually a lighter, more golden-yellow shade rather than the deeper red of a red bell pepper. The green stem area further supports that this is a yellow pepper, as the transition to the rest of the fruit shows no red tones. Given these observations, it's clear that the correct answer to the query is B. Yellow.</think><image_start>
> [Image: ./imgs/bagel_out_ca9f0a02_1.jpg]
> Round_1:
> <image_end><think>To determine whether the bell pepper in the picture is red or yellow, I closely examine the image provided. The bell pepper is the central focus of the close-up shot, and its color is clearly visible. The pepper displays a bright, consistent yellow hue over its entire surface, including the area around the green stem. There are no red or orange tones present on the pepper, which eliminates the possibility of it being red. The surrounding green leafy vegetables and the purple kale in the background further highlight the distinct yellow color of the bell pepper.
>
> ```
> The critical difference is that no one tells ThinkMorph to zoom in. It infers that magnification will help distinguish colors. This is **reasoning-oriented deployment, not instruction-following** .
> | Aspect | Bagel (Explicit Instruction) | ThinkMorph (Autonomous Reasoning) |
> |--------|------------------------------|-----------------------------------|
> | Trigger | User says "crop the pepper" | Question asks "Is it red or yellow?" |
> | Decision | User tells what to do | Model decides zoom helps |
> | Context | Manipulation is the goal | Manipulation is a reasoning step |
> | Integration | Isolated operation | Part of text → visual → text chain |
> | Role | Following instruction | Enables fine-grained color distinction |

---

> ### Author Response · Authors · 2025-11-22
> **[3/8] W1-(1) concrete case comparision**
>
> [3/8]
>
> > **(1.2) From Raw Capability to Reasoning Context: Bagel Baseline Cannot Use Manipulations for Reasoning**
>
>
>
> To test whether Bagel's raw capabilities translate to reasoning utility, we force Bagel baseline to generate interleaved thoughts on the same example:
>
> **Bagel baseline fails to use visual manipulation for reasoning.** When forced to generate interleaved thoughts, it reproduces original image after giving wrong answer then  collapsing into repetitive text with wrong answer.
>
>
>
> - **Bagel Baseline (Forced TIT Format)**
> ```
> <think>
> To determine whether the male in the picture is wearing a shirt or a T-shirt, I need to closely examine the clothing details.
>
> 1. **Observation of the Image**: The male is dressed in a long-sleeved garment that appears to have a collar and buttoned placket, which are typical characteristics of a shirt.
> 2. **Comparison with Options**:
>    - A shirt usually has a collar, buttons, and sleeves that extend to the wrists.
>    - A T-shirt is typically sleeveless or has short sleeves and lacks a collar.
> 3. **Conclusion**: Based on the visible features, the male is wearing a shirt, not a T-shirt.
>
> </think>
> The answer is A. Shirt.
> [Image: ./bagel_out_55a1f8ab_1.jpg]
> <think>
> Okay, let's see. The question is asking whether the male in the picture is wearing a shirt or a T-shirt. The options are A. Shirt or B. T-shirt.
>
> First, I need to look at the image provided. The man is walking with a woman, and he's wearing a long-sleeved shirt. Let me check the details. The shirt has a collar, buttons down the front, and sleeves that go all the way to his wrists. That's definitely not a T-shirt, which usually has short sleeves or is sleeveless.
>
> Wait, sometimes people might wear a button-up shirt as a T-shirt, but technically, a T-shirt is a different type of garment. The shirt in the image has a more formal look with a collar and buttons, so it's a shirt. The woman is wearing a T-shirt, but the question specifically asks about the male. So the answer should be A. Shirt.
>
> I should make sure there's no trick here. Sometimes, people might wear a collared shirt over a T-shirt, but in this case, the man is only wearing the long-sleeved shirt. No other layers visible. So yes, the correct answer is A.
> </think>
> <think>
> The question asks whether the male in the picture is wearing a shirt or a T-shirt. Observing the image, the man is dressed in a long-sleeved garment with a collar and buttons, which are characteristic features of a shirt. A T-shirt typically lacks a collar and has short sleeves, which is not the case here. Therefore, the correct answer is A. Shirt.
> </think>
> <think>
> The question asks whether the male in the picture is wearing a shirt or a T-shirt. Observing the image, the man is dressed in a long-sleeved garment with a collar and buttons, which are characteristic features of a shirt. A T-shirt typically lacks a collar and has short sleeves, which is not the case here. Therefore, the correct answer is A. Shirt.
> </think>
> [Repeats same textual reasoning 20+ times, no meaningful visual contribution]
> <think>
> The question asks whether the male in the picture is wearing a shirt or a T-shirt. Observing the image, the man is dressed in a long-sleeved garment with a collar and buttons, which are characteristic features of a shirt. A T-shirt typically lacks a collar and has short sleeves, which is not the case here. Therefore, the correct answer is A. Shirt.
> </think>
>
> ```

---

> ### Author Response · Authors · 2025-11-22
> **[4/8] W1-(1) concrete case comparision & (2) (3)**
>
> [4/8]
>
> This shows that Bagel's raw capabilities remain dormant for reasoning: it cannot autonomously deploy visual operations to solve problems. ThinkMorph, by contrast, autonomously generates task-relevant visual manipulations (zoom-in case shown in Fig.14, Appendix) toward correct answer.
>
> ```
> Round_0:
> <think>To determine whether the male in the picture is wearing a shirt or a T-shirt, I need to closely observe the upper body attire of the male figure. The image shows two individuals walking together, and the male is on the left. Upon examining his clothing, I notice that his garment has a collar and appears to be a long-sleeved, button-up style, which are characteristic features of a shirt. T-shirts typically do not have collars and are usually made from a different fabric, fitting more closely to the body. In this case, the male's clothing matches the description of a shirt rather than a T-shirt. To emphasize the answer, it makes sense to highlight the area of the image where the shirt is visible, helping to clearly indicate why the correct choice is 'A. Shirt'.</think><image_start>
>
> [Image: ./imgs/bagel_out_2b6b0e48_1.jpg]
>
> Round_1:
> <image_end><think>To determine whether the male in the picture is wearing a shirt or a T-shirt, I closely examine his upper body attire. The edited visualization helps by highlighting the specific area of his clothing. The male is wearing a long-sleeved garment with a collar and buttons, which are distinctive features of a shirt. T-shirts usually do not have collars and are made from different fabric, offering a closer fit. The highlighted area clearly shows the structured, collared design typical of a shirt. Therefore, based on these observations, the correct answer is: A. Shirt.</think><answer>A</answer>
> ```
>
> ---
>
> > **(1.3) ThinkMorph outperforms Bagel**
>
> If Bagel's raw capabilities were sufficient for reasoning, we should see comparable OOD performance. Instead:
>
> | Model | VSP | VisPuzzle | ChartQA | VStar⭐ | BLINK-J⭐ | MMVP⭐ | SAT⭐ | BLINK⭐ | CV-Bench⭐ |
> |-------|-----|-----------|---------|--------|-----------|--------|-------|---------|------------|
> | Bagel-7B | 0.83* | 35.00* | 61.82 | 55.49 | 67.33 | 70.33 | 44.67 | 47.66 | 76.03* |
> | ThinkMorph | 75.83 | 79.00 | 78.10 | 67.02 | 72.00 | 80.33 | 52.67 | 60.07 | 80.82 |
> | Δ | +75.00 | +44.00 | +16.28 | +11.53 | +4.67 | +10.00 | +8.00 | +12.41 | +4.79 |
>
> ⭐ = OOD
>
> ---
>
> **(2) Two-Stage Mechanism**
>
> > Stage 1 - Pretraining builds raw abilities
>
> Bagel's pretraining exposes the model to diverse visual-text patterns. This creates latent capabilities for visual operations (zoom, crop, inpaint, etc.). But pretraining doesn't teach using these for reasoning. Bagel baseline cannot produce useful visual thoughts for problem-solving.
>
> > Stage 2 - Fine-tuning activates for reasoning
>
> ThinkMorph fine-tuning teaches when and how to use these latent abilities within reasoning chains. The model learns:
> - Text cues trigger visual operations
> - Visual operations help reasoning steps
> - Coordination between text description and visual action
>
> Result: Operations never explicitly trained (zoom on MMVP, inpainting on BLINK-J) emerge when contextually useful.
>
> ---
>
> **(3) Evidence of Systematic Patterns**
>
> The emergence shows consistent patterns, not randomness:
>
> - **Text-visual correlations**: Specific phrases trigger operations consistently. "Examine closely" leads to zoom-in. "Restore" leads to inpainting. "Focus on" leads to highlighting (see Appendix C.1).
>
> - **Task-effectiveness**: Manipulations advance problem-solving (Fig 4).
>
> - **Frequency**: ThinkMorph shows up to 10% unseen manipulations on OOD tasks. Bagel baseline shows ~0%. High task-effectiveness proves these are principled, not random.
>
> This follows transfer learning principles: pretraining provides latent capabilities, fine-tuning activates them for reasoning, and text cues trigger operations when useful.
>
> ---

---

> ### Author Response · Authors · 2025-11-22
> **[5/8] W2/Q3: Clarification of Autonomous Mode Switching-1**
>
> [5/8]
>
> ### **W2/Q3: Clarification of Autonomous Mode Switching**
>
> We thank the reviewer for these important questions about mode switching emergence and the role of hybrid training. We address concerns about **(1)Quantitative analysis shows Robustness Across Different Training Settings and Benchmarks**, **(2) qualitative analysis** and **(3) Hybrid Analysis to avoid conflation**.
>
> ---
> **Key revisions:**
>
> - Sec 3.4: Added switching rate (5.3%), accuracy data, and case studies
>
> - Fig 5: Shows automatic mode switching with examples
>
> - Fig 8,9: Shows mode distribution during test-time scaling
>
> - Appendix C.2: Includes 5 case studies on adaptive vs. non-adaptive switching
>
>
> >  **Data Clarity and Emergence Evidence**: Clarifying "90% accuracy": We apologize for the ambiguous phrasing. The accurate statement is: *Switched instances reach 81.25% accuracy overall, compared to 73.96% for interleaved reasoning on the same samples (+7.29%)* in Fig.5.
> ---
> **(1) Quantitative analysis shows Robustness Across Different Training Settings and Benchmarks**
>
>
> The mode switching property appears **consistently across multiple training paradigms**.
>
> > **Training Setting 1: Single-Task Interleaved** (Chart Refocus or Jigsaw Assembly)
>
> Sec. 3.3 reports results for a model trained exclusively on interleaved data and tested on OOD MMVP. The switching rate is 5.3%. Switched samples achieve 81.25% accuracy while interleaved samples achieve 73.96%, showing a gain of **7.29%**.
>
> > **Training Setting 2: Generalized ThinkMorph** (24K Mixed Tasks, All Interleaved)
>
> Sec 3.3 reports results for a model trained on interleaved data from all 4 tasks (approximately 24K examples) and tested on broader OOD benchmarks:
>
> ```
> "The first two properties remain consistent: unseen visual manipulations continue to emerge on out-of-domain tasks, while autonomous mode switching remains adaptive, with the model transitioning between text-only and interleaved reasoning based on task complexity on BLINK and CV-Bench."
> ```
>
> We observe that the property emerges robustly across different training scales (6K versus 24K) and different OOD benchmarks (MMVP, BLINK, CV-Bench).
>
>
> ---
>
>
>
>
> **(2) qualitative analysis reveals mechanism of effectiveness and efficiency**
>
> >**Effectiveness**: Fig.5 includes detailed case analysis demonstrating systematic patterns based on visual complexity.
>
>
>  Contrasting Examples :
>
> | **Question** | **Visual Complexity** | **Model Behavior** | **Why** |
> |--------------|----------------------|-------------------|---------|
> | *"Can you see stems of bananas?"* | **High** (fine details) | Stays interleaved; generates zoom-in | Needs visual details to see small stems |
> | *"We cannot see the window on the school bus?"* | **Low** (describable) | Switches to text-only; describes yellow paint | Initial visual info is enough; text finishes reasoning |
>
> The mechanism works through **early visual checks** . After seeing the image and question, the model decides if text alone is enough. If the first visual pass gives clear information, it switches to text-only. If details are still unclear, it keeps using interleaved steps.
>
> >**Efficiency**: We also experiment whether switching is driven by computational efficiency by forcing samples that switched to text-only to continue with interleaved reasoning instead.  Forced interleaving adds heavy compute with no accuracy gain.
>
> Example: For "Is the duck's entire beak visible?" the model performs early visual analysis and determines the answer is clear. It then switches to text-only:
>
> In the natural text-only path, the model answers in one round using 89 tokens and outputs “A. Yes.”:
>
> ```
> <think>To determine whether the duck's entire beak is visible in the picture, I need to closely examine the beak's position and orientation relative to the rest of the duck's head and the frame of the image. First, I observe the duck's beak in the provided photograph. The beak is prominently visible and extends outwards from the side of the duck's head. There is no portion of the beak that is cut off or cropped by the edges of the image. The beak appears to start at the base of the duck's face and continue smoothly into the foreground, with no breaks or obstructions. Given this assessment, it is clear that the entire beak is within the visible area of the photograph. Therefore, the answer to the question 'Is the duck's entire beak visible in the picture?' is 'A. Yes.'</think>
> <answer>A. Yes</answer>
> ```

---

> ### Author Response · Authors · 2025-11-22
> **[6/8] W2&Q3-2**
>
> [6/8]
>
> In the forced interleaved path, the model needs two rounds and 156 tokens to give the same answer “A. Yes,” which costs about 75% more tokens:
>
> > Round_1 of the forced interleaved path **repeats the same visual conclusion** reached in Round_0 ("beak extends directly out", "no visible cuts", "entire beak is within the visible area"). The extra image token (repeated original image) adds **no new information**.
> ```
> Round_0:
> <think>To determine whether the duck's entire beak is visible in the picture, I need to closely examine the beak's position and orientation relative to the rest of the duck's head and the frame of the image. First, I observe the duck's beak in the provided photograph. The beak is prominently visible and extends outwards from the side of the duck's head. There is no portion of the beak that is cut off or cropped by the edges of the image. The beak appears to start at the base of the duck's face and continue smoothly towards the right side of the photo, without any breaks or obstructions. Since the beak is fully within the visible area and there is no indication that it extends beyond the frame, I can conclude that the duck's entire beak is indeed visible. Therefore, the correct answer to the question is 'A. Yes.'</think>
> <image_start>
> [repeated original image]
> <image_end><think>To determine whether the duck's entire beak is visible in the picture, I closely examine the provided image. The beak extends directly out from the side of the duck's head and there are no visible cuts or crop marks at the beak's end. The beak appears to start at the base of the duck's face and continue smoothly towards the right edge of the frame, indicating that the entire beak is within the visible area. Since the beak is fully within the photo and there are no portions of it obscured or cut off, the correct answer is: A. Yes.</think>
> <answer>A</answer>
> ```
>
>
>
> - **This reveals dual benefits of switching:** (1) **Task-adaptive**: Routes based on whether continued visual analysis is essential. (2) **Efficiency-aware**: Avoids redundant visual computation when initial perception suffices. The model not only recognizes when vision is non-essential but also allocates computation accordingly—text-only for perception-resolved cases, interleaved for cases requiring iterative visual refinement.
>
>
>
> ---
>
> **(3) Hybrid Analysis**
> As we mentioned in Training Setting 2, ThinkMorph trained on 24K mixed tasks but all interleaved samples still remains adaptive mode switching based on task complexity on BLINK and CV-Bench, even in test-time scaling. To investigate the more about mode switch during test time, we conducted a dedicated experiment as follow:
>
> > **Training Setting 3: Hybrid Mode Training**, ~24K mixed data covering text, visual, and interleaved modes:
>
> Sec 3.6 details the setup and reports the results on OOD benchmarks:
>
> | Benchmark | Switching Rate | Text vs. Interleaved Accuracy | Pattern |
> |---------------|-------------------|-----------------------------------|-------------|
> | MMVP | 6.38% | Text: +9.75% higher | Strong text preference for perception |
> | VStar | 8.64% | Text: +1.84% higher | Moderate text advantage |
> | BLINK-J | 1.25% | Text: -2.98% lower | Interleaved superior for jigsaw |
>
> The same training data (hybrid 24K) produces very different switching rates (1.25% to 8.64%) across benchmarks. This shows that switching **responds to the task** at test time, not the training mix. This rules out training bias. The different patterns across tasks confirm that switching is driven by what the task needs, not how the model was trained.

---

> ### Author Response · Authors · 2025-11-22
> **[7/8]  W3, Q1,Q2**
>
> [7/8]
>
> ### **W3: Generalizability Beyond Vision-Centric Puzzle Tasks**
>
> We address this through **(1) Extensive OOD Evaluation Beyond Puzzle Tasks** and **(2) Future Directions**.
>
> **Key revisions:**
> - **Tab 2**: Added MMVP as OOD benchmark; marked OOD benchmarks (VStar, BLINK-J, MMVP) with ⭐
> - **Tab 3**: Added BLINK and CV-Bench as OOD benchmarks; marked all 6 OOD benchmarks with ⭐
> - **Sec 3.3 (L337-362)**: Added discussion on when interleaved reasoning is essential vs. supplementary
>
> ---
>
> **(1) Extensive OOD Evaluation Beyond Puzzle Tasks**
>
> We evaluate on 6 OOD benchmarks testing diverse capabilities beyond puzzle-like tasks:
>
> | Benchmark | Capability Tested | Type |
> |-----------|------------------|------|
> | MMVP | Fine-grained visual perception and detail disambiguation | Perception |
> | SAT | Abstract spatial reasoning and pattern completion | Abstract reasoning |
> | BLINK (complete) | General visual perception across 14 diverse tasks | General perception |
> | CV-Bench | 3D spatial understanding and scene reasoning | 3D reasoning |
> | VStar | Compositional visual search with multi-hop reasoning | Compositional |
> | BLINK-J | Jigsaw puzzles across different visual domains | Domain transfer |
>
> > These differ fundamentally from training tasks:
>
> **For training** , VSP uses frozen lake grids. VisPuzzle uses 2 to 4 pieces in simple grids (1×2, 2×2). Visual Search uses GQA/VSR with scene graphs (approximately 7K examples, bounding box covering 1% to 30%). Chart Refocus uses a ChartQA subset.
>
> **For OOD evaluation** , VStar requires multi-hop reasoning without scene graphs. BLINK-J uses a different jigsaw setup where the model selects one piece to fill a black box. MMVP tests fine-grained perception. SAT tests abstract patterns. CV-Bench tests 3D understanding. BLINK tests 14 different perception tasks.
>
> > Results show consistent improvements across diverse capabilities: **Average OOD improvement: +8.58%**.
>
> | Model | VStar⭐ | BLINK-J⭐ | MMVP⭐ | SAT⭐ | BLINK⭐ | CV-Bench⭐ |
> |-------|--------|-----------|--------|-------|---------|-----------|
> | Bagel-7B | 55.49 | 67.33 | 70.33 | 44.67 | 47.66 | 76.03 |
> | ThinkMorph | 67.02 | 72.00 | 80.33 | 52.67 | 60.07 | 80.82 |
> | Δ | +11.53 | +4.67 | +10.00 | +8.00 | +12.41 | +4.79 |
>
>
> > **This demonstrates generalization beyond puzzle tasks to:**
> - Perception tasks (BLINK: +12.41%)
> - Abstract reasoning (SAT: +8.00%)
> - 3D understanding (CV-Bench: +4.79%)
> - Fine-grained visual discrimination (MMVP: +10.00%)
>
> **Generalization beyond puzzles**: The advantage emerges when text and images continuously inform each other. Visual tokens enable steps text cannot: re-arranged pieces reveal mismatches, overlaid arrows validate routes, bounding boxes pinpoint locations. ChartQA vs. MMVP contrast (Sec 4.3) shows: textual reasoning suffices when visual info is redundant, but interleaved reasoning is essential for tasks requiring precise visual grounding—a principle that applies broadly, not just to puzzles.
>
> ---
>
> **(2) Future Directions**
>
>
> We agree that open-ended, creative, or dialog-based tasks are important directions. We welcome the reviewer's suggestions on tasks where interleaved reasoning might prove valuable.
>
> ---
>
> ### **Q1: Why VSP-Main and VisPuzzle Use Non-Think Mode for Bagel?**
>
> The **asterisks (*)** in Tables 2 and 3 mark tasks where Bagel uses no-think mode. This is explained in the updated captions.
>
> The reason is that on VSP-Main, VisPuzzle, and CV-Bench, think mode **prevents Bagel from generating valid outputs** . The `<think>` tags cause generation failures. ThinkMorph works in think mode because it is specifically trained on interleaved reasoning traces with structured `<think>` and `<answer>` tags for these tasks.
>
> **For fairness** , we report Bagel's best achievable performance. We use no-think mode where think mode fails and think mode elsewhere. This ensures fair comparison.
>
> ---
>
> ### **Q2: Where Do Accuracy Gains Primarily Originate—Text-Based or Interleaved Reasoning?**
>
> The answer depends on the task. As shown in **Fig.9** , we experimented and analyzed three benchmarks where both modes appear. On questions with both text and interleaved responses, we observe the following patterns.
>
> **Text performs better on perception tasks.**  On MMVP, text achieves 9.75% higher accuracy than interleaved. On VStar, text achieves 1.84% higher accuracy than interleaved. **Interleaved performs better on compositional tasks.** On BLINK-J, interleaved achieves 2.98% higher accuracy than text.
>
> The conclusion is that gains come from **mode diversity**, not one dominant mode. Text excels at perception, while interleaved excels at composition. The modality shift (text drops from 18.8% to 15.2%, interleaved rises from 81.2% to 84.8%) correlates with accuracy improvement (from 51.3% to 58.6%). This happens because the model explores multiple strategies. This **diversity** ensures the correct mode contributes through majority voting.

---

> ### Author Response · Authors · 2025-11-22
> **[8/8] Q4: Performance vs. Cost Analysis for Test-Time Scaling**
>
> [8/8]
>
> ### **Q4: Performance vs. Cost Analysis for Test-Time Scaling**
>
> We acknowledge interleaved reasoning is computationally expensive and provide performance vs. cost analysis.
>
>
> **Key Revision on Test-Time Scaling**
> - Fig 6&7, Tab 4&5: Proves interleaved scales under distribution shift
> - Sec 3.5 (L448-L478): Detailed analysis on test-time advantage
>
>
>
> ---
>
> **(1) Interleaved Reasoning Is More Expensive**
> We measured average token cost during test-time scaling on BLINK-J. We observe that interleaved reasoning costs approximately 3 times more than text at the same N due to image generation:
>
> | Mode | N=1 | N=2 | N=4 | N=8 |
> |------|-----|-----|-----|-----|
> | Text | 354 | 697 | 1,424 | 2,879 tokens |
> | Visual | 473 | 945 | 1,890 | 3,778 tokens |
> | Interleaved | 1,073 | 2,145 | 4,269 | 8,517 tokens |
>
>
>
> ---
>
> **(2) Interleaved Reasoning Offers Better Performance Despite Higher Cost**
>
> Despite higher cost, interleaved reasoning achieves **better performance-cost efficiency** on challenging tasks. On BLINK-J (Jigsaw Assembly), interleaved N=4 outperforms text N=8 by 2.0% at 48% higher cost. In contrast, scaling text to N=8 plateaus without further improvement.
>
> | Strategy | Accuracy | Token Cost | Cost Efficiency |
> |----------|----------|------------|-----------------|
> | Text N=8 | 68.0% | 2,879 | Baseline |
> | Interleaved N=2 | 64.0% | 2,145 | -4.0% accuracy, -25% cost |
> | Interleaved N=4 | 70.0% | 4,269 | +2.0% accuracy, +48% cost |
>
>
>
> **Text-only fails to scale even with matched compute within chain**: We give text-only 4× more samples to match compute cost within one chain. It still cannot match interleaved. More samples search the same narrow space without sufficient diversity.
>
>
> | Task | Interleaved | Cost | Text | Cost | Text (4× cost) | Cost |
> |------|-------------|------|------|------|----------------|------|
> | **VStar** | **63.87** | 938 | 56.02 | 236 | 55.50 | 3752 |
> | **VSP** | **86.67** | 611 | 49.17 | 595 | 47.17 | 2444 |
> | **VisPuzzle** | **73.75** | 1183 | 63.50 | 475 | 63.50 | 4732 |
>
>
> >  **Explanation**: This proves the scaling advantage comes from **multimodal search space** showed in **new Fig.1**, not more compute. Visual tokens open new solution paths that text-only chains cannot reach.

---

### Official Review · Reviewer_MNkK · 2025-10-30

**Soundness:** 3
**Presentation:** 3
**Contribution:** 4
**Rating:** 6
**Confidence:** 3

**Summary:**

ThinkMorph introduces a unified multimodal reasoning framework that enables dynamic interleaving of textual and visual generation, allowing language and vision to mutually advance reasoning rather than operate in parallel. It eliminates reliance on external tools by treating image generation as predictable tokens within an autoregressive sequence, using a VAE and diffusion model to internally construct visual edits. Training is based on 24,991 carefully curated interleaved reasoning traces across four vision-centric tasks—jigsaw assembly, spatial navigation, visual search, and chart refocus—forcing the model to alternately reason with language and produce visual interventions. The approach achieves substantial performance gains on vision-centered benchmarks, including a 38.75 percent improvement on jigsaw assembly and competitive results on MMVP and SAT, matching or exceeding large proprietary models. Crucially, the model exhibits emergent capabilities not present in training data, such as unseen visual manipulation—including zoom-in and image inpainting—and autonomous switching between text-only, image-only, and interleaved reasoning modes. These abilities arise from the richer exploration space enabled by interleaved training, allowing test-time scaling through heterogeneous sampling across reasoning modalities. Unlike prior methods relying on tool augmentation or implicit visual tokens, ThinkMorph establishes a unified representation where vision is not a cue but an active component of reasoning. The work shifts the paradigm from passive perception to active visual thinking, demonstrating that integrated generation and understanding can cultivate higher-level multimodal intelligence. Although computationally intensive, it establishes a foundational pathway for building unified models capable of adaptive, human-like multimodal reasoning.

**Strengths:**

1. ThinkMorph pioneeringly introduces the "interleaved chain-of-thought" multimodal paradigm, enabling deep, dynamic, and complementary synergy between language and vision during reasoning. It moves beyond treating vision as passive input or an external tool, allowing the model to autonomously generate and modify visual content within its reasoning sequence, achieving intrinsic visual manipulation. This unique mechanism also fosters the emergence of novel visual operations and intelligent modality adaptation behaviors not previously observed.

2. This work achieves unified representation and seamless switching between language and vision within a single autoregressive Transformer architecture. By incorporating special tokens like <image start> and <image end>, the model can autonomously trigger internal VAE and diffusion models for image generation and re-insertion. Coupled with a negative log-likelihood loss for text and a mean squared error loss for images, ThinkMorph ensures both the accuracy of linguistic reasoning and the quality of visual generation, thereby achieving effective integration of multimodal signals.

3. This paper excels in its presentation, clearly articulating the limitations of existing models and grounding its core motivation in human "think-and-sketch" cognitive patterns. Through rich illustrations and reasoning examples, it provides intuitive demonstrations of how interleaved reasoning functions and the visual capabilities that emerge from the model. Furthermore, the paper systematically compares its work with tool-augmented, preliminary interleaving, and implicit visual token approaches, highlighting its distinct advantages.

4. ThinkMorph's efficacy is thoroughly validated through experiments on four categories of custom-built, vision-centric tasks. These datasets were specifically designed to embody the characteristics of interleaved reasoning. The experimental results not only demonstrate significant performance improvements across multiple tasks via quantitative metrics but also present qualitative analyses of generated interleaved reasoning chains, visual operation instances, and adaptive modality selection, providing strong evidence for the method's effectiveness.

**Weaknesses:**

1. The emergent visual manipulation capabilities, such as image inpainting and zoom-in, are presented as indicators of higher-level intelligence, yet their geometric and semantic consistency with the original image structure remains unvalidated. It is unclear whether the generated regions are aligned with the true physical structure or whether the model may produce visually plausible but physically incorrect details in the absence of explicit scale or spatial constraints.

2. Autonomous mode switching occurs in only a small fraction of samples, approximately 2 to 5 percent, and is interpreted as an adaptive behavioral trait. However, the underlying triggering mechanisms are not quantitatively analyzed. It remains unresolved whether this behavior is driven by semantic properties of the task, such as the presence of color, quantity, or relational queries, or whether it arises from latent biases in the training data distribution.

3. The computational overhead of interleaved reasoning during inference is not addressed. Each instance of image generation requires execution of VAE encoding, diffusion-based denoising, and re-encoding into clean visual tokens, resulting in substantial latency and limiting scalability.

**Questions:**

This paper highlights ThinkMorph's interleaved textual and visual reasoning, demonstrating significant performance gains on vision-centric tasks. However, generating and re-integrating image content into the LLM's token sequence incurs substantial computational overhead due to VAE encoding, diffusion denoising, and increased sequence length. Are these visual reasoning scenarios fundamentally intractable for pure text-based models, thus strictly necessitating this high-resource graphical reasoning for effective problem-solving?

---

> ### Author Response · Authors · 2025-11-22
> **[1/5] Manuscript Revision**
>
> We sincerely thank the reviewer for the thoughtful and constructive feedback.
>
> To address the concerns raised, we have carefully revised **[the manuscript](https://openreview.net/pdf?id=mB3vxfrQZM)** and enhanced the presentation for better clarity. These revisions include new evaluation results (**Tab.2 — Tab.5**), enhanced figures (**Fig.1 — Fig. 9**).  We also provide an [anonymous repository](https://anonymous.4open.science/r/ThinkMorph-rebuttal-71B2) with the data, training details, and rebuttal images for further transparency.
>
> We believe these improvements significantly address the core concerns and **invite the reviewer to examine [the manuscript](https://openreview.net/pdf?id=mB3vxfrQZM)**. Below, we provide detailed responses to each question.
>
> > **Note:** If the anonymous repository does not load, please try opening the link in an incognito browser window or clear your browser cache.
>
> [1/5]

---

> ### Author Response · Authors · 2025-11-22
> **[2/5] W1 Geometric and Semantic Consistency of Emergent Manipulations**
>
> [2/5]
>
> ### **W1 Geometric and Semantic Consistency of Emergent Manipulations**
>
> We address this through **(1) Quantitative Assessment**, **(2) Nature of Inconsistencies**, and **(3) Impact on Reasoning Accuracy**.
>
> **Key revisions:** Fig 14 as Example of valid zoom-in manipulation and Fig 17 as Example of inconsistent but reasoning-correct case
>
>
> ---
>
> **(1) Most Emergent Manipulations Are Physically Valid**
>
> We manually assessed ThinkMorph responses on MMVP (300 questions). We identified 28 responses showing unseen visual manipulations. Of these, **21 images (75%) showed consistency** with the original image structure. **Seven images (25%) showed inconsistencies** such as hallucinated object details or incorrect spatial relations.
>
> This indicates the vast majority of emergent visual manipulations respect physical structure. For example (Fig 14), ThinkMorph correctly zooms in on the man wearing a shirt, preserving geometric and semantic fidelity.
>
> ---
>
> **(2) Inconsistencies Arise from Over-Emphasis Rather Than Misunderstanding**
>
> We examined the seven inconsistent cases. All seven produced **correct final answers**. The pattern: ThinkMorph identified the correct answer but was **overly aggressive in editing the image to emphasize it**.
>
> Example (Fig 17): Given scattered vegetables, ThinkMorph correctly identified red bell peppers. It then generated an image showing a cluster of red bell peppers, making "red" very clear. However, this did not respect the original scattered layout. The model prioritized answer clarity over spatial fidelity.
>
> This suggests inconsistencies arise from the model's eagerness to visualize its conclusion, not from reasoning errors.
>
> ---
>
> **(3) Visual Hallucinations Do Not Lead to Wrong Answers**
>
> Critically, **hallucination in visual manipulation does not lead to incorrect final answers**. All seven physically inconsistent cases answered correctly. The model's reasoning remained sound—it simply visualized conclusions in ways that sacrificed physical fidelity for conceptual clarity.
>
> This reveals an interesting trade-off: the model sometimes chooses to emphasize the answer visually rather than maintain exact geometric consistency. While this affects visual realism, it does not compromise reasoning utility.
>
> > **Summary**: Most emergent manipulations (75%) are physically valid. When inconsistencies occur, they reflect over-emphasis rather than misunderstanding, and do not affect answer correctness. We leave training models that balance reasoning accuracy with visual consistency to future work.

---

> ### Author Response · Authors · 2025-11-22
> **[3/5] W2 Autonomous Mode Switching Mechanisms- (1) (2)**
>
> ### **W2 Autonomous Mode Switching Mechanisms**
>
> We directly address the concerns by **(1) quantitative analysis**, **(2) qualitative analysis** , **(3) Switching also brings efficiency gains** and **(4) Low Frequency Still Matters** .
>
>
>
> **Key revisions:**
>
> - Sec 3.4: Added switching rate (5.3%), analysis and case studies
>
> - Fig 5: Shows automatic mode switching with examples
>
> - Fig 8,9: Shows mode distribution during test-time scaling
>
> - Appendix C.2: Includes 5 case studies on adaptive vs. non-adaptive switching
>
> ---
>
>
>
> **(1) Quantitative analysis shows Robustness Across Different Training Settings and Benchmarks**
>
> The mode switching property appears **consistently across multiple training paradigms**.
>
> > **Training Setting 1: Single-Task Interleaved** (Chart Refocus or Jigsaw Assembly)
>
> Sec. 3.3 reports results for a model trained exclusively on interleaved data and tested on OOD MMVP. The switching rate is 5.3%. Switched samples achieve 81.25% accuracy while interleaved samples achieve 73.96%, showing a gain of **7.29%**.
>
> > **Training Setting 2: Generalized ThinkMorph** (24K Mixed Tasks, All Interleaved)
>
> Sec 3.3 reports results for a model trained on interleaved data from all 4 tasks (approximately 24K examples) and tested on broader OOD benchmarks:
>
> ```
> "The first two properties remain consistent: unseen visual manipulations continue to emerge on out-of-domain tasks, while autonomous mode switching remains adaptive, with the model transitioning between text-only and interleaved reasoning based on task complexity on BLINK and CV-Bench."
> ```
>
> We observe that the property emerges robustly across different training scales (6K versus 24K) and different OOD benchmarks (MMVP, BLINK, CV-Bench).
>
> > **Training Setting 3: Hybrid Mode Training**. To investigate the switch mechanisms, we conducted a dedicated experiment with **hybrid training** (~24K mixed data covering text, visual, and interleaved modes):
>
> Sec 3.6 details the setup and reports the results on OOD benchmarks:
>
> | Benchmark | Switching Rate | Text vs. Interleaved Accuracy | Pattern |
> |---------------|-------------------|-----------------------------------|-------------|
> | MMVP | 6.38% | Text: +9.75% higher | Strong text preference for perception |
> | VStar | 8.64% | Text: +1.84% higher | Moderate text advantage |
> | BLINK-J | 1.25% | Text: -2.98% lower | Interleaved superior for jigsaw |
>
> The same training data (hybrid 24K) produces very different switching rates (1.25% to 8.64%) across benchmarks. This shows that switching **responds to the task** at test time, not the training mix. This rules out training bias. The different patterns across tasks confirm that switching is driven by what the task needs, not how the model was trained.
>
> ---
>
>
>
>
>
> **(2) qualitative analysis reveals mechanism**
>
> Fig.5 includes detailed case analysis demonstrating **systematic patterns** based on visual complexity:
>
>  Contrasting Examples :
>
> | **Question** | **Visual Complexity** | **Model Behavior** | **Why** |
> |--------------|----------------------|-------------------|---------|
> | *"Can you see stems of bananas?"* | **High** (fine details) | Stays interleaved; generates zoom-in | Needs visual details to see small stems |
> | *"We cannot see the window on the school bus?"* | **Low** (describable) | Switches to text-only; describes yellow paint | Initial visual info is enough; text finishes reasoning |
>
> The mechanism works through **early visual checks** . After seeing the image and question, the model decides if text alone is enough. If the first visual pass gives clear information, it switches to text-only. If details are still unclear, it keeps using interleaved steps.

---

> ### Author Response · Authors · 2025-11-22
> **[4/5] W2-(3) (4)**
>
> **(3) Beyond correctness: Switching also brings efficiency gains.**
> We also tested whether switching is driven by computational efficiency by forcing samples that switched to text-only to continue with interleaved reasoning instead. Result: Forced interleaving adds heavy compute with no accuracy gain. The final answers did not change, but computational cost increased significantly.
>
> > **Example:** For "Is the duck's entire beak visible?" the model performs early visual analysis and determines the answer is clear. It then switches to text-only.
>
> In the natural text-only path, the model answers in one round using 89 tokens and outputs “A. Yes.”:
>
> ```
> <think>To determine whether the duck's entire beak is visible in the picture, I need to closely examine the beak's position and orientation relative to the rest of the duck's head and the frame of the image. First, I observe the duck's beak in the provided photograph. The beak is prominently visible and extends outwards from the side of the duck's head. There is no portion of the beak that is cut off or cropped by the edges of the image. The beak appears to start at the base of the duck's face and continue smoothly into the foreground, with no breaks or obstructions. Given this assessment, it is clear that the entire beak is within the visible area of the photograph. Therefore, the answer to the question 'Is the duck's entire beak visible in the picture?' is 'A. Yes.'</think>
> <answer>A. Yes</answer>
> ```
>
> In the forced interleaved path, the model needs two rounds and 156 tokens to give the same answer “A. Yes,” which costs about 75% more tokens:
>
> > Round_1 of the forced interleaved path **repeats the same visual conclusion** reached in Round_0 ("beak extends directly out", "no visible cuts", "entire beak is within the visible area"). The extra image token (repeated original image) adds **no new information**.
> ```
> Round_0:
> <think>To determine whether the duck's entire beak is visible in the picture, I need to closely examine the beak's position and orientation relative to the rest of the duck's head and the frame of the image. First, I observe the duck's beak in the provided photograph. The beak is prominently visible and extends outwards from the side of the duck's head. There is no portion of the beak that is cut off or cropped by the edges of the image. The beak appears to start at the base of the duck's face and continue smoothly towards the right side of the photo, without any breaks or obstructions. Since the beak is fully within the visible area and there is no indication that it extends beyond the frame, I can conclude that the duck's entire beak is indeed visible. Therefore, the correct answer to the question is 'A. Yes.'</think>
> <image_start>
> [repeated original image]
> <image_end><think>To determine whether the duck's entire beak is visible in the picture, I closely examine the provided image. The beak extends directly out from the side of the duck's head and there are no visible cuts or crop marks at the beak's end. The beak appears to start at the base of the duck's face and continue smoothly towards the right edge of the frame, indicating that the entire beak is within the visible area. Since the beak is fully within the photo and there are no portions of it obscured or cut off, the correct answer is: A. Yes.</think>
> <answer>A</answer>
> ```
>
>
>
> **This reveals dual benefits of switching:** (1) **Task-adaptive**: Routes based on whether continued visual analysis is essential. (2) **Efficiency-aware**: Avoids redundant visual computation when initial perception suffices. The model not only recognizes when vision is non-essential but also allocates computation accordingly—text-only for perception-resolved cases, interleaved for cases requiring iterative visual refinement.
>
>
>
>
>
> **(4) Low Frequency Still Matters**
>
> The low switching rate is still useful.
>
> 1. **The model can already choose** between text and interleaved modes, and these rare switches give clear accuracy gains (+7.29–9.75%).
> 2. **This gives a good base for RL**, which can train the model to pick the better mode more often—text for simple cases and interleaved for harder ones.
> 3. **Even with low frequency**, the model shows an early sense of when each mode helps, giving a clear direction for improving mode selection in future work.

---

> ### Author Response · Authors · 2025-11-22
> **[5/5] W3&Q1 Computational Overhead and Necessity of Visual Reasoning**
>
> [5/5]
> ### **W3&Q1 Computational Overhead and Necessity of Visual Reasoning**
>
> We acknowledge interleaved reasoning is computationally expensive. We address **(1) When Text Fails**, **(2) Cost vs. Performance**, and **(3) Adaptive Efficiency**.
>
>
>
> ---
>
> **(1) Text-Only Fails on Vision-Centric Tasks**
>
> Text-only reasoning hits a performance ceiling on vision-centric tasks. Even with more compute, text cannot match interleaved.
>
> > **Experiment**: We trained text-only with 4× more samples (matching interleaved compute cost):
>
> | Task | Interleaved | Text-only | Text (4× compute) |
> |------|-------------|-----------|-------------------|
> | VStar | 63.87% | 56.02% | 55.50% |
> | VSP | 86.67% | 49.17% | 47.17% |
> | VisPuzzle | 73.75% | 63.50% | 63.50% |
>
> The result shows that more text samples simply search **the same narrow space**. Visual tokens open solution paths that text cannot reach. The reason is that tasks like VSP need visualized paths, VisPuzzle needs manipulated pieces, and VStar needs bounding boxes. These require visual manipulation as reasoning steps, not just description.
>
> ---
>
> **(2) Cost vs. Performance during test time scaling** : Interleaved reasoning costs approximately 3 times more than text, but it achieves better efficiency on some vision-centric tasks.
>
> **Token cost (BLINK-J):**
>
> Based on the image, here's the complete table:
>
> | Mode | N=1 | N=2 | N=4 | N=8 |
> |------|-----|-----|-----|-----|
> | Text | 354 | 697 | 1,424 | 2,879 |
> | Visual | 473 | 945 | 1,890 | 3,778 |
> | Interleaved | 1,073 | 2,145 | 4,269 | 8,517 |
>
> We see this in several performance examples in Table 4. On BLINK-J, interleaved N=4 (70.0%) beats text N=8 (68.0%), and text plateaus. On VSP, text N=8 reaches 56.83% while interleaved N=8 reaches 92.33%. On MMVP, interleaved N=1 (81.33%) beats text N=8 (80.33%) with lower total cost. These results show that on vision-centric tasks, **text cannot match interleaved regardless of scaling** . The extra cost is justified.
>
> ---
>
> **(3) Adaptive Efficiency**
>
> Not all tasks need expensive interleaved reasoning. ThinkMorph switches to text-only when visual manipulation is **supplementary** (Property 2, Fig.5).
>
> For example, consider the question "Is duck's beak visible?" shown in upper W2. Text-only reasoning uses 89 tokens and produces the correct answer. When forced to use interleaved reasoning, the model uses 156 tokens to reach the same answer. This means **75%** more cost with no benefit.

---

> > ### Comment · Reviewer_MNkK · 2025-11-26
> >
> > Thanks for the rebuttal. I believe this work is a good contribution to joint image-text generative MLLMs for interleaved multimodal reasoning. Very nice response, it addressed most of my concerns. I will raise my rating.

---

> > > ### Author Response · Authors · 2025-11-26
> > >
> > > Thank you for your encouraging response. We are optimistic about unified multimodal reasoning and will continue to be committed to advancing it.
> > >
> > > Please let us know if you have any other questions, we would be more than happy to address them.

---

### Official Review · Reviewer_qnFT · 2025-10-31

**Soundness:** 3
**Presentation:** 3
**Contribution:** 3
**Rating:** 6
**Confidence:** 3

**Summary:**

The authors present ThinkMorph, a model for interleaved multimodal reasoning with text and images. They train a joint image-text generative MLLM on a few particular tasks with data generated for those tasks, such as visual search and jigsaw assembly. They create a data generation pipeline for each of these, which create visual tokens and text tokens specific to the task after defining visual manipulations for each tasks. Training on these results in improved performance for the MLLM compared to just the text or just the visual tokens. They also show some improvements in test-time scaling, and note some interesting emergent properties on out-of-domain data.

**Strengths:**

- The problem is well-motivated, mixing reasoning across modalities and aiming to identify where it performs beyond unimodal reasoning and where it may not.
- I appreciate the qualitative analysis and notes on emergent properties.
- Results are evaluated on meaningful OOD benchmarks and provide some benefit in many settings.
- The exploration of the emergent properties is interesting and sheds light on the method, such as the test-time mode dynamics of when mode-switching occurs in Figure 2.

**Weaknesses:**

- Results seem specific to the tasks specific considered, but are presented as general conclusions about interleaved reasoning more broadly. I appreciate the out-of-distribution results, but they generally target similar tasks -- like V-Star evaluating fine-grained visual search.
- The idea that the multimodal setting has a larger reasoning space is intuitively appealing, but it's unclear if this is actually the origin of the test-time advantage just from the presented results. For instance, the multimodal training could hypothetically lead to broader diversity within just text, leading to higher best-of-N performance. It would be interesting to see the hypothesis empirically borne out with some quantitative understanding of the diversity of output samples and qualitative examples of how trajectories differ.

**Questions:**

- It wasn't fully clear to me how the visual output tokens work in that: is it producing tokens for the full output image, including the parts of the input it wants to reproduce, or just the manipulation it wants to show? This would change how it should be presented.
- How dependent is the method on the pretrained model used?
- How do the authors hypothesize the unseen manipulations emerge despite the lack of training?
- How is it possible for the performance to ever decrease with higher N in Table 4?

---

> ### Author Response · Authors · 2025-11-22
> **[1/8]  Manuscript Revision**
>
> We sincerely thank the reviewer for the thoughtful and constructive feedback.
>
> To address the concerns raised, we have carefully revised **[the manuscript](https://openreview.net/pdf?id=mB3vxfrQZM)** and enhanced the presentation for better clarity. These revisions include new evaluation results (**Tab.2 — Tab.5**), enhanced figures (**Fig.1 — Fig. 9**).  We also provide an [anonymous repository](https://anonymous.4open.science/r/ThinkMorph-rebuttal-71B2) with the data, training details, and rebuttal images for further transparency.
>
> We believe these improvements significantly address the core concerns and **invite the reviewer to examine [the manuscript](https://openreview.net/pdf?id=mB3vxfrQZM)**. Below, we provide detailed responses to each question.
>
> > **Note:** If the anonymous repository does not load, please try opening the link in an incognito browser window or clear your browser cache.
>
> [1/8]

---

> ### Author Response · Authors · 2025-11-22
> **[2/8] W1 Task-Specificity**
>
> [2/8]
>
> ### **W1 Task-Specificity**
>
> We appreciate this point. Our results show broad generalization of ThinkMorph beyond task-specific settings. We illustrated this through **(1) Distinction Between Task-Specific and OOD Evaluation**, **(2) Interleaved Reasoning Excels on both** and **(3) Quantitative Evidence** .
>
> **Key revisions**
>
> - **Tab 2**: Added MMVP as OOD benchmark for Chart Refocus; marked OOD benchmarks (VStar, BLINK-J, MMVP) with ⭐ to show interleaved reasoning generalization
>
> - **Tab 3**: Added BLINK and CV-Bench as OOD benchmarks; marked all 6 OOD benchmarks with ⭐ to demonstrates gains across diverse capabilities (perception, 3D reasoning, fine-grained vision)
>
>
> - **Sec 3.3 (L337-362)**: Added Discussion with ChartQA vs. MMVP contrast to explains when interleaved reasoning is essential (OOD) vs. supplementary (in-domain)
>
>
> ---
>
>
>
>  **(1) Clear Distinction Between Task-Specific and OOD Evaluation**
>
>
> | Evaluation Type | Benchmarks |
> |---------------------|----------------|
> | Task-Specific | VSP, VisPuzzle, ChartQA |
> | OOD ⭐ | VStar, BLINK-J, MMVP, SAT, BLINK (complete), CV-Bench |
>
> > **OOD benchmarks differ fundamentally from training:**
>
> - **VStar**: Training uses GQA/VSR with scene graphs for single-object localization (bbox 1-30%, ~7K examples). VStar uses different images without scene graphs. It requires multi-hop reasoning, attribute matching, and compositional logic not present in training.
>
> - **BLINK-J**: Training uses 2-4 pieces in grid configs (1×2, 2×1, 1×3, 3×1, 2×2) with multiple full arrangement options. BLINK-J requires selecting one piece from options to fill a black box in an incomplete puzzle. Different jigsaw setup.
>
> - **MMVP, SAT, BLINK, CV-Bench**: Test fine-grained perception, abstract reasoning, and 3D understanding—capabilities not in training data.
>
> > **All 6 OOD Benchmarks Test Different Capabilities:**
>
> | **Benchmark** | **Capability Tested** |
> |---------------|----------------------|
> | **VStar** | Compositional visual search with multi-hop reasoning |
> | **BLINK-J** | Jigsaw puzzles across different visual domains |
> | **MMVP** | Fine-grained visual perception and detail disambiguation |
> | **SAT** | Abstract spatial reasoning and pattern completion |
> | **BLINK** | General visual perception across 14 diverse tasks |
> | **CV-Bench** | 3D spatial understanding and scene reasoning |
>
> ---
>
>  **(2) Quantitative Evidence of Broad Generalization**
>
>
> > ThinkMorph fine-tuning on single training task (6k samples per task) yields **34.74\% improvement over baseline** and surpasses the next-best mode by **5.33\%**.
>
> | Model | VSP | VStar⭐ | VisPuzzle | BLINK-J⭐ | ChartQA | MMVP⭐ |
> |-------|-----|---------|-----------|-----------|---------|--------|
> | Bagel-7B | 0.83* | 55.49 | 35.00* | 67.33 | 62.05 | 70.33 |
> | Text Reasoning | 49.17 | 56.02 | 63.50 | 68.67 | 81.66 | 76.33 |
> | Visual Reasoning | 85.50 | 58.63 | 61.25 | 47.33 | 73.08 | 73.00 |
> | **Interleaved Reasoning** | **86.67** | **63.87** | **73.75** | **73.33** | **79.78** | **82.66** |
>
>
> **On task-specific benchmarks** , interleaved reasoning achieves 86.67% on VSP (best, +85.84% over baseline), 73.75% on VisPuzzle (best, +38.75%), and 79.78% on ChartQA (competitive, -1.88% versus text). **On OOD benchmarks** , it achieves 63.87% on VStar (best, +8.38% over baseline), 73.33% on BLINK-J (best, +6.00%), and 82.66% on MMVP (best, +12.33%).
>
>
> > ThinkMorph fine-tuning on 24K high-quality interleaved thought samples drawn from all 4 training tasks and **achieves consistent improvements across diverse benchmarks** :
>
>
> | **Model** | **VSP** | **VisPuzzle** | **ChartQA** | **VStar⭐** | **BLINK-J⭐** | **MMVP⭐** | **SAT⭐** | **BLINK⭐** | **CV-Bench⭐** |
> |-----------|---------|---------------|-------------|------------|--------------|-----------|----------|------------|---------------|
> | Bagel-7B | 0.83* | 35.00* | 61.82 | 55.49 | 67.33 | 70.33 | 44.67 | 47.66 | 76.03* |
> | ThinkMorph | 75.83 | 79.00 | 78.10 | 67.02 | 72.00 | 80.33 | 52.67 | 60.07 | 80.82 |
> | **Δ** | **+75.00** | **+44.00** | **+16.28** | **+11.53** | **+4.67** | **+10.00** | **+8.00** | **+12.41** | **+4.79** |
>
>
>
>
> The average improvement across 6 OOD benchmarks is +8.58%. This demonstrates that interleaved reasoning generalizes far beyond task-specific patterns, improving performance on:
> - Perception tasks (BLINK: +12.41%)
> - Spatial reasoning (SAT: +8.00%)
> - 2D and 3D spatial understanding (CV-Bench: +4.79%)
> - Fine-grained visual discrimination (MMVP: +10.00%)
> ---
>  **(3) Qualitative Analysis**
>
> We added **detailed analysis with examples (Fig.3)** examining **when interleaved reasoning is necessary**. The advantage emerges when text and images continuously inform each other: visual tokens enable steps text cannot. The ChartQA versus MMVP results suggest that textual reasoning suffices when visual information is redundant, but interleaved reasoning becomes essential for tasks requiring precise visual grounding.

---

> ### Author Response · Authors · 2025-11-22
> **[3/8] W2 Analysis on the origin of the test-time advantage-1**
>
> [3/8]
>
>
> ### **W2. Analysis on the origin of the test-time advantage**
>
> We thank the reviewer for raising an insightful hypothesis.
>
> Beyond our claim that **a larger multimodal reasoning space increases trajectory diversity and improves test-time scaling**, the reviewer suggests that: *Perhaps the gains come from improved text diversity induced by multimodal training, even without interleaved chains.*
>
>
>
> We explore this by Testing Both Hypotheses:
> > **H1 (Ours)**: Interleaved CoT creates a *larger multimodal search space* at test-time scaling, enabling diverse visual-grounded trajectories toward the correct solution → better test-time scaling.
>
> > **H2 (Reviewer's)**: Multimodal training also enriches text-space itself, improving text diversity → even text reasoning becomes stronger.
>
> **Conclusion**: Both hypotheses are correct and they contribute *different* parts of the final scaling benefits. Interleaved reasoning brings *unique* diversity that text cannot express, while multimodal training indeed improves text reasoning.
>
>
> ---
>
> **Key revisions:**
>
>
> - **Fig 6&7**, **Tab 5&4**: Proves interleaved scales under distribution shift
> - **Sec 3.5 (L448-L478)**: Detailed analysis on test-time advantage
>
> ---
>
>
>
> > **Evidence for H1: Multimodal Space Enables Superior Scaling**
>
> **(1) Interleaved reasoning scales consistently across benchmarks**
>
> Tab. 3 shows Best-of-N results for three modes. Interleaved scales consistently. Interleaved chains explore the multimodal space and find correct paths. Unimodal chains search narrow spaces and cannot maintain effectiveness.
>
>
> | Task | Mode | N=1 | N=2 | N=4 | N=8 |
> |------|------|-----|-----|-----|-----|
> | **VSP** | Text | 48.67 | 48.33 | 51.33 | 56.83 |
> | (in-domain) | Visual | 83.83 | 83.83 | 88.50 | 91.33 |
> | | **Interleaved** | **87.17** | **87.33** | **90.67** | **92.33** |
> | **VStar** | Text | 61.26 | 60.73 | 63.87 | 63.35 |
> | (moderate shift) | Visual | 56.02 | 56.54 | 58.64 | 61.26 |
> | | **Interleaved** | **65.97** | **67.02** | **67.54** | **67.02** |
> | **MMVP** | Text | 74.67 | 75.33 | 78.67 | 80.33 |
> | (general perception) | Visual | 74.33 | 73.00 | 74.00 | 75.00 |
> | | **Interleaved** | **81.33** | **78.67** | **82.00** | **82.00** |
> | **BLINK-J** | Text | 65.33 | 64.67 | 67.33 | 68.00 |
> | (large shift) | Visual | 51.33 | 51.33 | 52.00 | 49.33 |
> | | **Interleaved** | **65.33** | **64.00** | **70.00** | **73.33** |
>
> ---
>
>
> **(2) Text-only fails to scale even with matched compute**
>
> We further force the text‑reasoning model to generate much longer trajectories, giving text‑only an expanded exploration space (up to 4× the average token count of interleaved thoughts).
> Even under this substantially increased compute budget, text‑only thought still cannot close the gap. More samples search the same narrow space without sufficient diversity.
>
> | Task        | Interleaved (Acc) | Interleaved (Cost) | Text (Acc) | Text (Cost) | Text 4× (Acc) | Text 4× (Cost) |
> |-------------|-------------------|---------------------|------------|-------------|----------------|-----------------|
> | **VStar**   | **63.87**         | 938                 | 56.02      | 236         | 55.50         | 3752            |
> | **VSP**     | **86.67**         | 611                 | 49.17      | 595         | 47.17         | 2444            |
> | **VisPuzzle** | **73.75**       | 1183                | 63.50      | 475         | 63.50         | 4732            |
>
>
>  This proves the scaling advantage comes from **multimodal search space** showed in **new Fig.1**, not more compute. Visual tokens open new solution paths that text-only chains cannot reach.
>
> **(3) Visual tokens generate unseen manipulations (Property 1)**
>
> Interleaved chains produce visual operations never seen during training. They help solve problems effectively.
> This proves visual tokens explore beyond training data. **The multimodal space has richer diversity than what the model was trained on.** Text-only chains cannot generate these novel visual solutions.

---

> ### Author Response · Authors · 2025-11-22
> **[4/8] W2 Analysis on the origin of the test-time advantage-2**
>
> [4/8]
>
> > **Evidence for H2: Multimodal Training Enriches Text**
>
> **(1) Text in interleaved chains shows higher entropy and richer vocabulary**
>
> We analyze text from correct predictions on VStarBench and Jigsaw. Interleaved text has higher entropy and more unique words. This proves multimodal training enriches text itself:
>
> | Benchmark | Mode | Entropy ↑ | Unique Vocab ↑ |
> |-----------|------|-----------|----------------|
> | **VStarBench** | Interleaved | **7.665** | **3186** (+11.4%) |
> | | Text-only | 7.499 | 2860 |
> | **Jigsaw** | Interleaved | **7.615** | **3795** (+24.1%) |
> | | Text-only | 7.353 | 3057 |
>
>
>
> **Top words reveal different vocabularies:** Interleaved uses more visual-relational terms like "bounding" and "box". Text-only focuses on generic words like "answer" and "image".
>
>
> **VStarBench:** [Figure: VStarBench Word Clouds (Interleaved)](https://anonymous.4open.science/r/ThinkMorph-rebuttal-71B2/rebuttal_figures/VStarBench_tit_wordcloud.png) | [Text-only](https://anonymous.4open.science/r/ThinkMorph-rebuttal-71B2/rebuttal_figures/VStarBench_text_wordcloud.png)
>
> Interleaved: *bounding* (1.89%), *box* (2.10%), *left* (1.73%), *right* (1.76%), side, red | Text-only: *answer* (4.95%), *image* (3.79%), color, side, left, right
>
>
>
> **Jigsaw:**  [Figure: Jigsaw Word Clouds (Interleaved)](https://anonymous.4open.science/r/ThinkMorph-rebuttal-71B2/rebuttal_figures/Jigsaw_tit_wordcloud.png) | [Text-only](https://anonymous.4open.science/r/ThinkMorph-rebuttal-71B2/rebuttal_figures/Jigsaw_text_wordcloud.png)
>
> Interleaved: *background* (1.17%), *missing* (1.39%), *lower* (1.52%), right, third | Text-only: *image* (8.03%), first, right, second, lower
>
>
>
>
>
>
> **(2) Text-Only Reasoning Gets Better**
>
> The model switch to pure text and still answers correctly. This shows that **multimodal training makes the text space richer and more expressive.** In these cases, the improved text-only mode performs well—much better than a text-only model trained without interleaving. This strengthened text channel allows the model to solve easy cases without relying on visual steps.
>
> ---
>
> > **H1 & H2 Work Together**
>
> Multimodal training improves both channels: 1. Text becomes richer, enabling pure text reasoning on simple cases (Property 2) ; 2. Multimodal reasoning brings unique paths that are essential for hard cases where text fails
>
>
> | Evidence | H1 (MM Space) | H2 (Text Improves) |
> |----------|-------------------|-------------------|
> | Text-only fails to scale (+2.67% vs +8.00%) | ✓ | |
> | Interleaved scales under distribution shift | ✓ | |
> | Visual-relational vocabulary emerges | ✓ | ✓ |
> | Text entropy increases after training | | ✓ |
> | Text-only mode succeeds on switched cases | | ✓ |
> | Larger scaling gains for interleaved | ✓ | ✓|

---

> ### Author Response · Authors · 2025-11-22
> **[5/8] Q1 and Q2**
>
> [5/8]
>
> ### **Q1 Visual Output Token Mechanism**
>
> We clarify the model generates **complete output images**, not just manipulations through **(1)Clearer Statement**, **(2) Training Design**, and **(3) Concrete Examples**.
>
> **Key revisions:** Fig 2 to clarifie how model do interleaved reasoning on token-level
>
>
>
> ---
>
> **(1) The Model Generates Complete Images**
>
> As stated in Sec 2.1, image thoughts use `<image_start>` and `<image_end>` delimiters. Between these delimiters, the model generates a complete sequence of image tokens. The VAE then decodes these tokens into full visual outputs. This is end-to-end image generation. It is not partial modification like isolated edits or overlays.
>
> ---
>
> **(2) Training Design Ensures Complete Generation Aligned with Bagel**
>
> Fig 2 shows our dual training objectives aligned with Bagel:
>
> - **$\mathcal{L}_{\text{img}}$**: MSE loss between ground-truth and generated images
> - **$\mathcal{L}_{\text{text}}$**: Negative log-likelihood for text tokens
>
> The MSE loss supervises complete image generation. The model learns to produce **full images** where task elements (boxes, arrows, arrangements) are integrated into complete scenes, not separate layers.
>
> ---
>
> **(3) Concrete Examples from Fig 1**
>
> - **Visual Search**: Generates complete scene with bounding box rendered inside (not a box overlay)
> - **Jigsaw**: Generates entire re-assembled puzzle as one image (not just moved pieces)
> - **Navigation**: Generates complete map with arrows integrated (not arrows on top)
>
> Each visual thought is a standalone complete image. The model synthesizes full visual context with task elements as integral parts, not separate additions.
>
> ---
>
>
>
>
> ### **Q2 Dependency on Pretrained Model**
>
> We address this through **(1) New Experiments on Chameleon vs. Bagel**, **(2) Analysis with Examples**, and **(3) Requirements and Future Outlook**.
>
> **(1)  New Experiments on Chameleon vs. Bagel**
>
> We trained identical data (text/visual/interleaved reasoning) on Chameleon-7B and Bagel-7B for VSP. Chameleon fails on all modes. Bagel succeeds and interleaved reasoning achieves best results:
>
> | **Model** | **Mode** | **VSP** |
> |-----------|------------|---------|
> | Bagel-7B | Text  | 49.17 |
> | Bagel-7B | Visual | 85.50 |
> | Bagel-7B | **Interleaved** | **86.67** |
> | Chameleon-7B | Text | 1.33 |
> | Chameleon-7B | Visual| 2.33 |
> | Chameleon-7B | Interleaved | 2.33 |
>
>
>
> > Note: Other unified models like MetaMorph (https://arxiv.org/abs/2412.14164) and EMU3.5 (https://arxiv.org/abs/2510.26583) are either too large or unavailable.

---

> ### Author Response · Authors · 2025-11-22
> **[6/8] Q2 Dependency on Pretrained Model**
>
> [6/8]
>
> **(2) Text Reasoning Is the Core Foundation**
>
> The text reasoning scores show the core problem. **Chameleon achieves only 1.33% while Bagel achieves 49.17%.** This reveals a key insight: visual thoughts only help when the base model can already reason well with text. Chameleon lacks this foundation and also generates low-quality images.
>
>
>
>
> **Qualitative example** (VSP task):
>
>
> Bagel demonstrates both coherent Chain-of-Thought reasoning and high-quality image generation, as shown in [[Bagel_img]](https://anonymous.4open.science/r/ThinkMorph-rebuttal-71B2/rebuttal_figures/Bagel_vsp.png).
>
> ```
> Round_0:
> <think>To get an overview of this frozen lake maze, I'll start by noting the positions of key elements.
> The grid is 7 rows by 7 columns. The starting square is located in the fifth row from the top and the
> third column from the left. The goal, which looks like a gift box, sits in the sixth row from the top
> and the fifth column from the left.
>
> For the ice holes: In the top row, there are holes in the first and fifth columns. The second row has
> holes in the second column. The third row features a hole in the sixth column. The fourth row contains
> holes in the second and seventh columns. In the fifth row, there's a hole in the first column. The
> sixth row has a hole in the third column and the seventh row in the fifth column.
>
> With the locations of the start, the goal, and all the ice holes mapped out, it's time to plot the
> safest path through the maze.</think><image_start>
>
> [Image: ./bagel_out_d56e436e_1.jpg]
>
> Round_1:
> <image_end><think>Examining my solution, I see that the red path starts from the player's position in
> the fifth row and third column. The first move is right, taking the player to the fourth column in the
> same row, which is a safe frozen surface. The next move is down, bringing the player to the sixth row
> and fourth column, again landing on a frozen surface. The final move is right once more, placing the
> player at the sixth row and fifth column—the goal square.
>
> Throughout these moves, the path never crosses a hole or goes off the edge. Each step stays on safe
> ground, and the final position is the goal. This confirms that the path is correct.
>
> So, the verified solution is: \boxed{R,D,R}</think><answer>\boxed{R,D,R}</answer>
> ```
>
> **Answer:** `\boxed{R,D,R}` ✓
>
> ---
>
> Chameleon shows broken reasoning and low-quality image generation, as shown in [[chameleon_img]](https://anonymous.4open.science/r/ThinkMorph-rebuttal-71B2/rebuttal_figures/Chameleon_vsp.png).
>
>
> ```
> I'll begin by carefully laying out the structure of this ice maze. The grid is 3x3, with each square
> representing either a safe frozen ground, a dangerous ice hole, the goal, or the starting point.
>
> Looking at the maze:
> - The goal (represented by a present) is located at the bottom right corner of the grid (third row,
>   third column).
> - The starting position (the elf) is at the center of the maze (second row, second column).
> - Ice holes are present at the top left (first row, first column), and at the bottom left and bottom
>   center (third row, first and second columns).
> - All other squares are safe frozen surfaces.
>
> With this layout in mind, it's time to plot the route from the starting position to the
> goal.</think><image_start>
> [Image: ./9fdfe805-d7ca-4a1d-8294-c15ef7c469f5.png]
> <image_end><think>Time to double-check my solution by looking at the maze from another point of view.
> The route I plotted begins at the center square, moves one step downward (D), placing the player on the
> frozen surface at the bottom center, and then one step to the right (R), reaching the goal at the bottom
> right corner.
>
> There are no ice holes on this path; the player avoids all hazards and successfully reaches the goal.
> The moves are correct.
>
> So, the solution is \boxed{D,R}\boxed{}\boxed{}</think><image_end><answer>\boxed{D,R}\boxed{}</answer>
> ```
>
> **Answer:** `\boxed{D,R}\boxed{}` ✗
>
> > Key Difference: Bagel demonstrates systematic spatial reasoning with accurate grid understanding (7×7) and complete path verification, while Chameleon shows broken reasoning with incorrect grid size (3×3 instead of 7×7) and produces a malformed answer with extraneous `\boxed{}` tags.
>
>
>
> ---
>
> **(3) Requirements & Future Potential**
>
> ThinkMorph requires base models with: 1. Strong text reasoning (long CoT);  2. Good visual understanding ; 3. Good image generation.
>
> > **Future outlook**: Current unified models are still limited, but we observe **emergent properties** as a positive signal. As base models improve, ThinkMorph will bring bigger gains—stronger understanding and generation enable larger improvement. **We are optimistic about unified multimodal reasoning and committed to advancing this field**.
>
> ---

---

> ### Author Response · Authors · 2025-11-22
> **[7/8] Q3 Hypothesis for Emergent Unseen Manipulations**
>
> [7/8]
>
> ### **Q3 Hypothesis for Emergent Unseen Manipulations**
>
>
>  We address this through **(1) Two-Stage Mechanism**, **(2) The Emergence Shows Consistent Patterns**, and **(3) The Necessity of ThinkMorph's Training**.
>
> **Key revisions**: Fig.4, Sec 3.4 (L372—L40) and Appendix C.2 for more unseen cases
>
> ---
>
> **(1) Two-Stage Mechanism**
>
> > Stage 1 - Pretraining builds raw abilities
>
> Bagel's pretraining exposes the model to diverse visual-text patterns. This creates latent capabilities for visual operations (zoom, crop, inpaint, etc.). But pretraining doesn't teach using these for reasoning. Bagel baseline cannot produce useful visual thoughts for problem-solving.
>
> > Stage 2 - Fine-tuning activates for reasoning
>
> ThinkMorph fine-tuning teaches when and how to use these latent abilities within reasoning chains. The model learns:
> - Text cues trigger visual operations
> - Visual operations help reasoning steps
> - Coordination between text description and visual action
>
> Result: Operations never explicitly trained (zoom on MMVP, inpainting on BLINK-J) emerge when contextually useful.
>
> ---
> **(2) The Emergence Shows Consistent Patterns**
>
> We observe three types of evidence showing the emergence is systematic, not random.
>
> **First, text-visual correlations are consistent.** Specific phrases trigger operations reliably. The phrase "examine closely" leads to zoom-in. The phrase "restore" leads to inpainting. The phrase "focus on" leads to highlighting. These patterns are detailed in Appendix C.1.
>
> **Second, the manipulations are task-effective.** They advance problem-solving rather than appearing randomly. For example, when asked "Is the bell pepper red or yellow?", the model zooms in to distinguish colors, mimicking human visual inspection. This is shown in Figure 4.
>
> **Third, the frequency patterns are clear.** ThinkMorph shows up to 10% unseen manipulations on OOD tasks. Bagel baseline shows approximately 0%. The high task-effectiveness proves these manipulations are principled, not random.
>
> ---
>
> **(3) The Necessity of ThinkMorph's Training**
>
> | Model | Perform manipulation when prompted? | Uses in reasoning? | OOD performance |
> |-------|-------------------------------------|-------------------|-----------------|
> | Bagel | ✓ Yes | ✗ No | - |
> | ThinkMorph | ✓ Yes | ✓ Yes | +20.79% avg on 9 benchmarks |
>
> **The key distinction is that Bagel possesses raw capabilities but cannot deploy them for reasoning.**  When forced to generate interleaved thoughts, it produces irrelevant images after giving answers and collapses into repetitive text. ThinkMorph, by contrast, autonomously generates task-relevant visual manipulations. For example, when asked "Is the bell pepper red or yellow?", no instruction tells ThinkMorph to zoom in. It infers that magnification will help distinguish colors. **This is reasoning-oriented deployment, not instruction-following**.
>
> If Bagel's raw capabilities were sufficient, we should see comparable performance. Instead, ThinkMorph achieves +10% on MMVP and +4.67% on BLINK-J. This demonstrates that fine-tuning **activates** latent capabilities as reasoning tools.

---

> ### Author Response · Authors · 2025-11-22
> **[8/8]  Q4 Deeper Analysis of Test-Time Scaling Tendency**
>
> [8/8]
>
> ### **Q4 Deeper Analysis of Test-Time Scaling Tendency**
>
> Tab.4 has been expanded into Tab.5 & 6 (Appendix). We conducted additional experiments to analyze this tendency, summarized in **(1) Single-Task Training**, **(2) Multi-Task Training**, and **(3) Mechanism about Decreases at Intermediate N**.
>
> - Key Revision: Sec 3.5, Fig. 6 & 7. and Tab.5 & 6.
> ---
>
>
>
> ### **(1) Single-Task Training: Interleaved Reasoning Shows Clear Scaling Benefits**
>
> We first study models trained on one task at a time (~6K examples each).
> The test-time scaling results across reasoning modes are provided in **[3/8] W2 Analysis on the Origin of the Test-Time Advantage-1**, so we do not repeat them here.
> Full results for this section are shown in Fig.6 and Tab.5 (Appendix).
>
> As shown in Fig.6, **interleaved reasoning scales more reliably** than text-only or image-only reasoning, especially on harder benchmarks where unimodal methods stop improving or even drop:
> - On VSP, interleaved reasoning rises from 87.17% to 92.33% (+5.16%), beating visual reasoning's final 91.33%.
> - On BLINK-J, interleaved reasoning achieves the largest gain (+8.00%, from 65.33% to 73.33%), while visual reasoning drops (from 51.33% to 49.33%).
> - Overall, interleaved reasoning gives the best or near-best results for all N on all benchmarks.
>
> **These results show that interleaved reasoning scales better than unimodal reasoning**, even if a small dip appears at some N values (e.g., BLINK-J at N=2, MMVP at N=2).
>
>
>
> ---
>
> **(2) Multi-Task Training: Task-Dependent Scaling Patterns**
>
> When ThinkMorph is trained on **24K mixed data** from all four tasks (detailed in Table 6, Appendix), test-time scaling behaviors vary across task types. As shown in Fig.7, we observe **two distinct patterns** on out-of-domain benchmarks:
>
> > **Pattern 1** shows monotonic scaling on reasoning-intensive tasks, with orange and blue lines in Figure 7 showing steady, monotonic improvement as N increases.
>
> | **Benchmark** | **Type** | **N=1** | **N=2** | **N=4** | **N=8** | **Net Gain** |
> |---------------|----------|---------|---------|---------|---------|--------------|
> | **VStar** | Visual search | 62.30% | 62.30% | 64.40% | **65.97%** | **+3.67%** |
> | **CV-Bench** | 3D reasoning | 81.31% | 81.65% | 82.60% | **83.25%** | **+1.94%** |
>
>
>
> > **Pattern 2** shows U-shaped scaling on perception-focused tasks, with green and red lines in Figure 7 showing decline at intermediate N and recovery at larger N.
>
> | **Benchmark** | **Type** | **N=1** | **N=2** | **N=4** | **N=8** | **Net Gain** |
> |---------------|----------|---------|---------|---------|---------|--------------|
> | **MMVP** | Fine-grained perception | 81.67% | 80.00% (**-1.67%**) | 80.33% (**-1.34%**) | **82.67%** | **+1.00%** |
> | **BLINK-J** | Jigsaw perception | 68.67% | 66.67% (**-2.00%**) | 66.67% (**-2.00%**) | **69.33%** | **+0.66%** |
>
>
> ---
>
> **(3) Performance Decrease at Intermediate N**
>
> | N   | Behavior   | Why It Can Drop                                      |
> | ------- | -------------- | -------------------------------------------------------- |
> | 1   | One sample     | No voting noise                                          |
> | 2–4 | Low diversity  | Samples are too similar, easy to agree on a wrong answer |
> | 8+  | High diversity | Wider search makes correct answers more likely           |
>
> Perception tasks drop more easily because several wrong visual guesses may cluster together. **A larger N is needed so at least one sample catches the right visual cue.** Notably, the same U-shape appears in both single-task and multi-task results for BLINK-J and MMVP. This suggests the drop is a fundamental property of perception tasks, not an artifact of our training approach.

---

### Meta-Review · Area_Chair_oWr2 · 2026-01-12

**Summary:**

- Concerned about over-generalization of task-specific results and a lack of empirical proof for the hypothesized origin of test-time advantages. Also, there are somequestions about visual token generation and model dependency.

- Concerns about validating emergent properties. 1. The geometric/semantic consistency of "unseen" visual manipulations, 2. The unclear, low-frequency mechanism behind "autonomous mode switching," and 3. The computational overhead of the method versus its necessity.

- The "emergent" visual manipulations might be pre-existing in the base model, and the analysis of "autonomous mode switching" was conflated by introducing a hybrid-trained model.

- Insufficient qualitative evidence for claims, unclear experimental settings, unfair baseline comparisons, ambiguous terminology, and questionable external baseline results.

**Reviewer Concerns:**

Addressed Concerns:

- Over-generalization & Evaluation: The authors added OOD benchmarks, showing consistent gains.

- Origin of Test-Time Advantage: The authors conducted new experiments to support both the authors' hypothesis and the reviewer's.

- Geometric Consistency of Manipulations: The authors added manual analysis showing 75% of emergent manipulations are physically valid.

- Mechanism of Mode Switching: The authors provided analysis, case studies, and an argument showing that switching avoids redundant computation.

- Clarification Questions: The authors  addressed all specific questions (visual token mechanism, model dependency, training details).

- Unfair Baseline Comparison: The authors clarified that Table 2 provides the exact fair comparison the reviewer requested.

 Unaddressed Concerns:

- Necessity vs. Novelty of "Emergent" Manipulations: While the authors argue that autonomous deployment for reasoning is novel, some skepticism may remain.

- Computational Cost & Scalability: The authors acknowledge the high cost and provide a cost-vs-performance analysis, showing interleaved reasoning can be more efficient per unit of accuracy on some tasks. However, the fundamental scalability limitation for real-time or large-scale deployment remains a valid, unanswered weakness of the method.

- Generalizability Beyond Puzzles: Broader task families remain untested.

**Reviewer Scores:**

Reviewer qnFT and Reviewer MNkK may raise their scores, but it is hard to guess Reviewer vxwn and Reviewer qUFF.

---

### Decision · Program_Chairs · 2026-01-26

Accept (Poster)